



# Comprehensive Assessment of Stress Calculations for Crevasse Depths and Testing with Crevasse Penetration as Damage

Benjamin Reynolds[1], Sophie Nowicki[1,2], Kristin Poinar[1,2]

[1]Department of Geology, University at Buffalo, Buffalo, NY 14260, USA
[2]RENEW Institute, University at Buffalo, Buffalo, NY 14260, USA

*Correspondence to*: Benjamin Reynolds (br77@buffalo.edu)

**Abstract.** Crevasse depth calculations with the Nye formulation or linear elastic fracture mechanics are used in many applications, including calving laws, determination of stable cliff heights, shelf vulnerability to collapse via hydrofracture, and damage evolution in ice. The importance of improving the representation of these processes for reducing sea-level rise

uncertainty makes careful calculation of stresses for crevasse depths critical. The resistive stress calculations used as input for these crevasse predictions have varied across studies, including differences such as the use of flow direction stress versus maximum principal stress, the inclusion of crevasse-parallel deviatoric stress, and calculation of effective strain rate. Some studies even use deviatoric stress in the place of resistive stress for crevasse depth calculations. Many studies do not provide an adequate description of how stress was calculated. We provide a systematic review of how resistive stress calculations

found in literature result in differing crevasse depth predictions and where these differences are pronounced. First, we study differences in crevasse size calculated from idealized representative strain rate states and then from velocity observations of several Antarctic ice shelves. To test whether the patterns of crevasse depths predicted from these stresses have a strong connection to bulk rheology, we use crevasse penetration as damage and compare predicted velocities from an ice sheet model against observed velocity. We find that the selection of stress calculation can change crevasse size predictions by a factor of

two and that differences are pronounced in shear margins and regions of unconfined, spreading flow. The most physically consistent calculation uses the maximum principal stress direction, includes vertical strain rate from continuity in the effective strain rate calculation, and uses three-dimensional resistive stress ($R_{xx} = 2\tau_{xx} + \tau_{yy}$). However, this calculation has rarely been used to date in studies requiring crevasse depth predictions. We find that this most physically consistent stress calculation produces a damage pattern that qualitatively matches surface features and quantitatively reproduces observed velocities better

than other stress calculations; we therefore recommend the use of this stress calculation. This result also suggests that other stress calculations likely overpredict shear margin vulnerability to hydrofracture and would overpredict calving in shear margins and spreading fronts when implemented in the crevasse depth calving law.





## 1 Introduction

Ice damage evolution, ice shelf collapse, and calving are three related processes that add uncertainty to the sea level rise

contributions of the Greenland and Antarctic ice sheets (van de Wal et al., 2022). In ice shelves, damage evolution can reduce the amount of buttressing provided by the shelf to the upstream ice. This causes an increase in speed of the upstream ice and thus a higher rate of sea-level contribution (e.g. Khazendar et al., 2015; Lhermitte et al., 2020). This damage evolution, sometimes aided by high surface meltwater availability, can also lead to collapse of entire ice shelves. Shelf collapse fully removes buttressing, so the corresponding change in speed of upstream ice can be large as seen for the inlet glaciers into the

Larsen B shelf (e.g. Rignot et al., 2004; Rott et al., 2011). While retreat from increased calving is less dramatic than a sudden ice shelf collapse, the result can be the same: increase in glacier velocities because of termini in locations that provide less backstress. This effect was demonstrated via ice sheet modeling of Sermeq Kujalleq (Jakobshavn Isbrae) that found terminus position change caused the majority of the doubling of the glacier's velocity (Bondzio et al., 2017). Finally, in the case of ice shelf collapse, initialization of retreat through the rapid, brittle mechanism of Marine Ice Cliff Instability (MICI) has been

proposed if cliffs are exposed that ice strength cannot support (Bassis and Walker, 2011; Pollard et al., 2015). A commonality between these processes is the importance of the presence and size of crevasses. For damage evolution, Sun et al. (2017) used crevasse penetration directly as damage. Albrecht and Levermann (2014) and Borstad et al. (2016) proposed damage laws that do not directly consider crevasse depth but use threshold stresses for damage initiation that can be linked crevasse initiation by linear elastic fracture mechanics (LEFM). For ice shelf vulnerability to hydrofracture, Lai et al. (2020) demonstrated that

crevasse presence predictions with LEFM aligns with locations where crevasses are observed. They then identify regions that both provide buttressing and are expected to be crevassed to assess where hydrofracture could cause shelf collapse that will yield increased upstream velocity. Calving has been modeled directly based on the predicted crevasse depths from local stresses in the crevasse depth calving law (Benn et al., 2007; Nick et al., 2010), which has been used by many subsequent studies (Amaral et al., 2020; Benn et al., 2023; Choi et al., 2018; Todd et al., 2018). Berg and Bassis (2022) also showed the importance

of crevasse advection from upstream for modeling calving. Finally, the limit on cliff height for stability under the MICI theory used crevasse penetration with the Nye crevasse formulation (Bassis and Walker, 2011). On top of these phenomena that are linked to and often modeled with crevasse depths, crevasses also affect surface energy balance (Cathles et al., 2011; Colgan et al., 2016).

There are two primary methods for calculating crevasse depths from stress. The Nye crevasse formulation (Nye,

1957) assumes ice has no tensile strength and that the presence of a crevasse does not modify the surrounding stress field. Linear elastic fracture mechanics (LEFM), which was applied to crevasses by van der Veen (1998a, 1998b), recognizes ice strength and considers the stress-amplifying effect of crevasse geometry. LEFM changes the predicted depths relative to Nye crevasse theory and allows for the determination of threshold stress for crevasse formation based on ice's fracture toughness and the size of the initial flaw (a small defect in the ice surface). Studies have assessed the differences in crevasse depth

predictions between these calculations and compared them to observations (Mottram and Benn, 2009; Enderlin and



Bartholomaus, 2020). A key input to these calculations is the full stress as a function of ice depth, although not all studies have used the full stress.

The crux of the problem is what component or components of ice stress control the propagation of a crevasse. The calculation steps to go from observed strain rate and temperature to the full stress have varied significantly across crevasse-depth studies. For example, five different stress calculations can be found in six studies that use crevasse depths for damage evolution, shelf collapse vulnerability, calving, and comparison to observed crevasses (Amaral et al., 2020; Choi et al., 2018; Enderlin and Bartholomaus, 2020; Lai et al., 2020; Mottram and Benn, 2009; Sun et al., 2017). The differences in stress calculations come from the stress orientation (flow direction or maximum principal), use and calculation of effective strain rate, and inclusion of the deviatoric stress running parallel to the crevasse in the resistive stress calculation. Two studies (Choi et al., 2018; Lai et al., 2020) have tested their methods across two different stress calculations, but to date no study has comprehensively surveyed the range of stress calculations used for crevasse depths.

We seek to determine more broadly whether and where selection of stress calculation is significant in determining crevasse depths. In the background, we will present the differences in stress calculations across studies in more detail. Then, in the methods, we show crevasse sizes for simple idealized strain rate states with each calculation before plotting crevasse penetration on real ice shelves. We compare predicted crevasse penetration against observed surface features and velocity to assess whether the results with each stress calculation are realistic. With stress calculation versions that seem plausible, we then study the connection between calculated crevasse depths and bulk ice rheology. We do this by testing the ability of crevasse penetration as damage to yield observed velocity fields with an ice sheet model.

## 2 Background

### 2.1 Stress contributions for crevasse calculations

While the viscous flow of ice is driven by stress differences (deviatoric stresses), brittle failure comes from the full stress. If ice is pulled on in triaxial tension, it will not flow but it may fracture. For this reason, the lithostatic pressure that increases with depth does not affect viscous deformation but does suppress crevasse extension. Calculating the full stress as a function of depth in the ice column to determine crevasse sizes is done by combining the resistive stress, lithostatic pressure, and water pressure (Fig. 1).



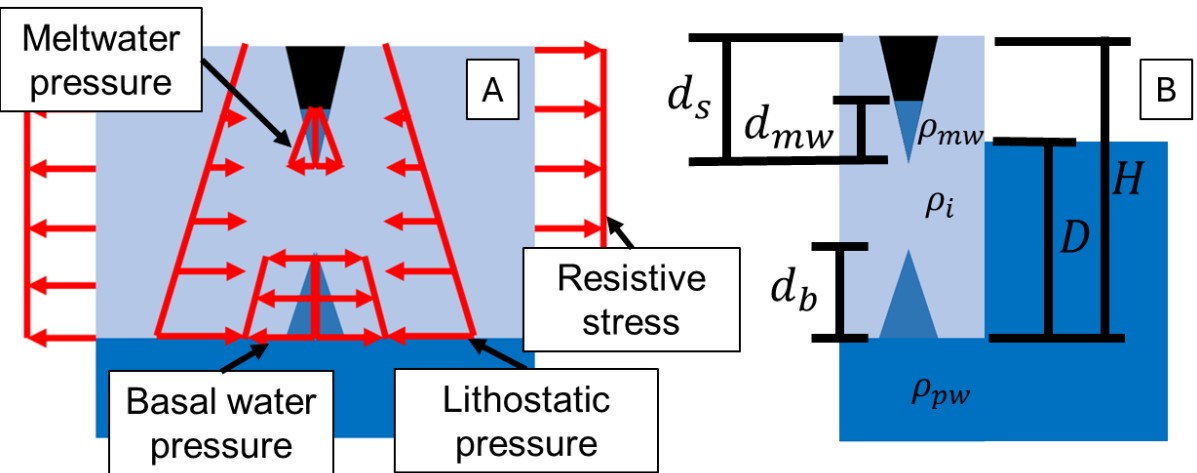

**Figure 1: Diagrams of (A) stress and pressures that control crevasse sizes and (B) variables used in crevasse size calculations. Physical property values and variable descriptions are provided in Table A1 and A2 in the appendix.**

The resistive stress is the stress pulling a crevasse open because of local ice extension. In areas with simple stress states, such as on an ice shelf or near an ice cliff, the average resistive stress may be known directly through force balance calculations. More often, however, stress states are complex; there, deviatoric stresses are calculated from strain rates using Glen's flow law, from which resistive stress may then be calculated. Within Glen's flow law, the selection of depth-averaged temperature, depth-averaged rigidity, a vertical temperature profile, or surface and basal temperatures will have large impacts but is not considered here. Change in ice rigidity with firn density is also important but not considered. The differences in stress calculations we consider are in the stress direction selected, the calculation of effective strain rate, and the consideration of deviatoric stress in the crevasse-parallel direction when calculating resistive stress.

Lithostatic pressure comes from the weight of ice above the vertical position in the ice column and counters the resistive stress to prevent crevasse extension. For surface crevasses, firn density is an important consideration as it can significantly change the lithostatic pressure near the surface (van der Veen, 1998a). In the case of pooled surface melt, water pressure acts against the lithostatic pressure allowing for more crevasse extension. Water in crevasses has been included as fixed depth (e.g., Benn et al., 2007) or as maintaining a water table height (e.g., van der Veen, 1998a). The water table height approach leads to hydrofracture once a crevasse reaches the water table (or potentially a little beyond for LEFM) due to water's density exceeding that of ice. Water in basal crevasses again acts against lithostatic pressure. In floating ice in hydrostatic equilibrium, the lithostatic pressure and water pressure at the bottom surface of the shelf are equal. In the case of a crevasse allowing water to reach higher in the ice column, the increase in pressure with elevation is proportional to the difference between ocean (or lake) water density and ice density. As the corresponding increase in compressive stress with depth for a dry surface crevasse is proportional to ice density, basal crevasses are predicted to be much larger than surface crevasses. This remains so even when the relative softness of basal ice near melting temperature is considered in the calculation, as shown later in the Results section (Table 3).



## 2.2 Nye crevasse formulation

The Nye crevasse formulation (Nye, 1957) assumes that ice has no tensile strength such that a crevasse extends as far into ice as tension is present. When the resistive stress is tensile, then, surface crevasses will extend down until the lithostatic pressure of ice equals the resistive stress. Weertman (1980) and Jezek (1984) used this idea to calculate basal crevasse heights on ice shelves by including the pressure from ocean water. The crevasse depth calving law (Benn et al., 2007; Nick et al., 2010) applies the Nye crevasse formulation for prediction of calving by limiting the terminus position to where surface crevasses do not reach waterline and the combined surface and basal crevasse sizes do not penetrate the full thickness of ice. The water in surface crevasses can thus be used as a tuning variable (e.g., Choi et al., 2018; Amaral et al., 2020). Subsequent studies based on the Nye crevasse formulation (Bassis and Walker, 2011; Sun et al., 2017) have applied the equations in Nick et al. (2010) for surface and basal crevasses, which we reprint here for easy reference. Surface crevasse depth, $d_s$, is given by

$$d_s = \frac{R_{xx}}{\rho_i g} + \frac{\rho_{mw}}{\rho_i} d_{mw} \, , \tag{1}$$

where $R_{xx}$ is the resistive stress perpendicular to the crevasse, $\rho_i$ is ice density, $\rho_{mw}$ is meltwater density, $d_{mw}$ is the depth of meltwater in the crevasse, and $g$ is gravitational acceleration (Fig. 1). Basal crevasse height, $d_b$, is given by

$$d_b = \frac{\rho_i}{\rho_{pw} - \rho_i} \left( \frac{R_{xx}}{\rho_i g} - H_{ab} \right) \, , \tag{2}$$

where $\rho_{pw}$ is the density of the proglacial water (lake or ocean) and $H_{ab}$ is the height above buoyancy. Height above buoyancy is

$$H_{ab} = H - D \frac{\rho_{pw}}{\rho_i} \, , \tag{3}$$

where $H$ is ice thickness and $D$ is the depth of ocean or lake water in contact with the ice cliff. Height above buoyancy is zero for ice shelves assumed to be in hydrostatic equilibrium. The resistive stress, $R_{xx}$, in Nick et al. (2010) is given as

$$R_{xx} = 2 \left( \frac{\dot{\varepsilon}_{xx}}{A} \right)^{1/n} \, , \tag{4}$$

where $\dot{\varepsilon}_{xx}$ is the crevasse-perpendicular strain rate, $A$ is the flow factor in Glen's flow law, and $n$ is the flow exponent. This is the crevasse-perpendicular deviatoric stress multiplied by a factor of two. More generally, the resistive stress is defined as the full stress minus the lithostatic pressure (van der Veen, 2017).

## 2.3 Linear elastic fracture mechanics

The application of linear elastic fracture mechanics (LEFM) to crevasses was formulated by van der Veen (1998a, 1998b) and has been updated by Jiménez and Duddu (2018) to better include the effects of boundary conditions (e.g., grounded versus floating ice). LEFM uses the same stress inputs (resistive, lithostatic, and hydrostatic) but considers the stress concentration at the crevasse tip caused by the presence of the crevasse itself. A crevasse will extend to the depth where the stress intensity factor becomes smaller than the fracture toughness of ice. For a single basal crevasse for example, the stress intensity factor, $K_1$, as a function of the crevasse height, $h$, is given as





$K_1 = \int_0^h \frac{2\sigma_n(z)}{\sqrt{\pi h}} G(\gamma, \lambda) dz$ , (5)

where $\sigma_n$ is the far field full stress normal to the crevasse (which comes from resistive stress, lithostatic pressure, and water pressure), $z$ is the vertical distance from the shelf base, and $G(\gamma, \lambda)$ is a function that accounts for geometry and edge effects (van der Veen, 1998b). As the resistive stress inputs are the same between the Nye crevasse formulation and LEFM, we will work with the Nye crevasse formulation because it is simpler. Our findings will be applicable to LEFM with the recognition

that differences in crevasse sizes from stress calculations may be amplified with LEFM.

**2.4 Stress calculations**

Studies using crevasse depth calculations have selected one or more methods of calculating stress from strain rate and temperature. The choice breaks down into the categories of consideration and calculation of effective strain rate, stress direction, and inclusion of the deviatoric stress parallel to the crevasse. We will consider six combinations of these options to

150 explore where each choice is significant. We use the following notation throughout the study except when noted otherwise. Capital $X$, $Y$, and $Z$ subscripts indicate directions aligned to a global, arbitrary coordinate system with $X$ and $Y$ being the surface planar directions. Lower case $x$, $y$, and $z$ subscripts indicate the local crevasse-perpendicular, crevasse-parallel, and vertical directions, respectively. The local $x$ and $y$ will usually differ from the global $X$ and $Y$, as shown in Fig. 2, whereas $Z$ and z will be equivalent. The flow direction is indicated by the subscript, $flow\ dir$. Fig. 2 shows these coordinate systems.

Also, maximum and minimum principal stresses or strains from the planar tensor receive subscripts of 1 and 2, respectively. (The true minimum principal term would sometimes be the vertical direction.) $\tau_{ij}$ and $\sigma_{ij}$ are used to denote components of the deviatoric stress tensor and full stress tensor, respectively. Finally, $R_{xx}$ is the resistive stress perpendicular to the crevasse, which is the only direction of resistive stress needed.



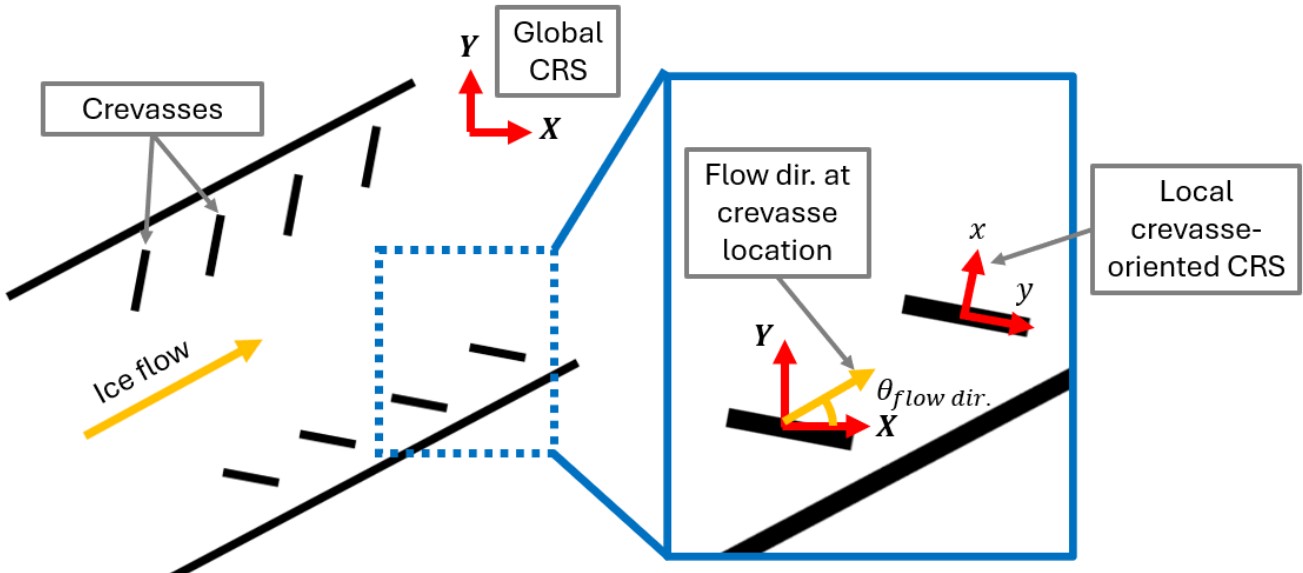

**Figure 2: Diagram showing the global coordinate reference system (CRS), local crevasse-aligned CRS, and local flow direction angle relative to the global CRS.**

### 2.4.1 Effective strain rate

The first determination in calculating stresses we consider is the calculation of the effective strain rate. Some implementations of crevasse depth calculations neglect effective strain rate and apply Glen's flow law in a single dimension to determine the deviatoric stress in the crevasse-perpendicular direction, $\tau_{xx}$, as

$$\tau_{xx} = A^{-1/n} \dot{\varepsilon}_{xx}^{1/n} . \tag{6}$$

Most remote sensing velocity products include only horizontal (or planar) velocities, so studies using these products are only able to directly calculate the planar strain rate components ($\dot{e}_{XX}, \dot{e}_{YY}, \dot{e}_{XY}$). From incompressibility, the divergence of velocity is zero. This means that the vertical strain rate is given by

$$\dot{e}_{ZZ} = -\dot{e}_{XX} - \dot{e}_{YY} . \tag{7}$$

It may be reasonable to neglect the vertical strain rate, as crevasses and low-density firn relax incompressibility. In the testing of the crevasse depth calving law by Amaral et al. (2020), the effective strain rate is calculated without the vertical strain rate. The same resulting value in our notation with the global coordinate systems comes from

$$\dot{e}_{eff,planar} = \sqrt{\tfrac{1}{2}\left(\dot{e}_{XX}^2 + \dot{e}_{YY}^2\right) + \dot{e}_{XY}^2} , \tag{8}$$

which we define as the planar effective strain rate, $\dot{e}_{eff,planar}$. When the vertical strain rate, $\dot{e}_{ZZ}$, from mass conservation is included, the effective strain rate is instead

$$\dot{e}_{eff} = \sqrt{\tfrac{1}{2}\left(\dot{e}_{XX}^2 + \dot{e}_{YY}^2 + \dot{e}_{ZZ}^2\right) + \dot{e}_{XY}^2} . \tag{9}$$



This version neglects only the vertical shear terms, $\dot{e}_{XZ}$ and $\dot{e}_{YZ}$, which is consistent with the shallow shelf approximation (MacAyeal, 1989). Each deviatoric stress component, $\tau_{ij}$, can then be calculated from each corresponding strain rate

component, $\dot{\varepsilon}_{ij}$, (Cuffey and Patterson, 2010) as

$$\tau_{ij} = A^{-1/n}\dot{\varepsilon}_{eff}^{\frac{1}{n}-1}\dot{\varepsilon}_{ij} \ . \tag{10}$$

We will consider crevasse depths calculated without effective strain rate, with the planar effective strain rate, and with the mass-conservation-based effective strain rate.

### 2.4.2 Stress direction

Studies including crevasse depth calculations in three dimensions must select a stress direction. In their tests of the crevasse depth calving law, Amaral et al. (2020) used the maximum principal stress direction while Choi et al. (2018) tested with both the maximum principal and flow direction stresses. Lai et al. (2020) studied the vulnerability of buttressing regions of ice shelves to hydrofracture with both stress directions as well. van der Veen (1999) found that crevasses generally open in or close to the maximum principal stress direction. We will show representative calculations and crevasse depth maps with each

direction. The flow direction, $\theta_{flow\ dir.}$, is given by

$$\theta_{flow\ dir.} = tan^{-1}(V_Y/V_X) \ , \tag{11}$$

where $V_Y$ and $V_X$ are the y and x components of velocity in the global coordinate system. The normal, deviatoric stress in the flow direction can be calculated as

$$\tau_{flow\ dir.} = \frac{\tau_{XX}+\tau_{YY}}{2} + \frac{\tau_{XX}-\tau_{YY}}{2}cos\left(2\theta_{flow\ dir.}\right) + \tau_{XY}sin\left(2\theta_{flow\ dir.}\right) \ . \tag{12}$$

The maximum and minimum principal deviatoric stresses from the planar stress tensor are given by

$$\tau_1, \tau_2 = \frac{\tau_{XX}+\tau_{YY}}{2} \pm \sqrt{\left(\frac{\tau_{XX}-\tau_{YY}}{2}\right)^2 + \tau_{XY}^2} \ , \tag{13}$$

which is the equation for the eigen values of the planar stress tensor (a two-by-two matrix). Note that there is no difference between rotating to a direction before or after calculating effective strain rate and deviatoric stress components, assuming that ice rheology is isotropic (the effective strain rate is an invariant).

### 2.4.3 Crevasse-parallel stress term in the resistive stress

From van der Veen (2017), the resistive stress, $R_{xx}$, is defined as the full stress minus the lithostatic pressure and is given by

$$R_{xx} = 2\tau_{xx} + \tau_{yy} \ , \tag{14}$$

when bridging stress is neglected. For the Nye crevasse formulation, a derivation that yields the use of the resistive stress is as follows. The definitions of the normal, planar, deviatoric stress terms, $\tau_{xx}$ and $\tau_{yy}$, are

$$\tau_{xx} = \sigma_{xx} - \frac{1}{3}\left(\sigma_{xx} + \sigma_{yy} + \sigma_{zz}\right) \ , \tag{15}$$

$$\tau_{yy} = \sigma_{yy} - \frac{1}{3}\left(\sigma_{xx} + \sigma_{yy} + \sigma_{zz}\right) \ , \tag{16}$$



where $\sigma$ terms are full stress components. These equations combine to give

$$\sigma_{xx} = 2\tau_{xx} + \tau_{yy} + \sigma_{zz} . \tag{17}$$

The vertical full stress component is assumed to come only from ice lithostatic pressure and water pressure. For surface and

basal crevasses respectively, this gives

$$\sigma_{zz} = -\rho_i g d_s + \rho_{mw} g d_{mw} , \tag{18}$$

$$\sigma_{zz} = -\rho_i g (H - d_b) + \rho_{pw} g (D - d_b) , \tag{19}$$

where (as before) $\rho_i$ is ice density, $\rho_{mw}$ is meltwater density, $\rho_{pw}$ is proglacial water density, $H$ is ice thickness, $D$ is proglacial water depth, $g$ is the gravitational acceleration, $d_s$ is surface crevasse depth, $d_{mw}$ is the height of meltwater in

surface crevasses, and $d_b$ is basal crevasse height. Equations 18 and 19 can be substituted into Equation 17 to find the full stress as a function of depth ($\sigma_n(z)$ in Equation 5) for surface and basal crevasses. The Nye crevasse formulation predicts crevasse tips where $\sigma_{xx} = 0$, which will give the transition point between tension and compression. This yields the following equations for surface and basal crevasse sizes, respectively:

$$d_s = \frac{2\tau_{xx} + \tau_{yy}}{\rho_i g} + \frac{\rho_w}{\rho_i} d_w , \tag{20}$$

$$d_b = \frac{\rho_i}{\rho_p - \rho_i} \left( \frac{2\tau_{xx} + \tau_{yy}}{\rho_i g} - H_{ab} \right) . \tag{21}$$

Note that when aligned such that $\tau_{xx}$ is the maximum principal deviatoric stress ($\tau_1$), $\tau_{yy}$ will be the minimum principal deviatoric stress from the surface stress tensor, $\tau_2$. The physical explanation is that the full stress in the longitudinal direction must be higher to create the same longitudinal deviatoric stress if there is also a tensile lateral stress.

Studies using two-dimensional flowline models like Nick et al. (2010) inherently do not use the crevasse-parallel,

$\tau_{yy}$, term. However, for studies working with plan view ice sheet models or remote sensing products, this term has been included (Amaral et al., 2020) or neglected (Choi et al., 2018; Lai et al., 2020; Sun et al., 2017). The resulting resistive stress, $R_{xx}$, terms are shown in Table 1. We only consider the crevasse-parallel stress term when using the maximum principal direction stress, as we found no examples in the literature that considered it with flow direction stress.

Finally, several studies have used a deviatoric stress component (either maximum principal stress or flow direction)

rather than the resistive stress in implementations of both the Nye crevasse formulation and LEFM (Sun et al., 2017; Choi et al., 2018; Enderlin and Bartholomaus, 2020). Use of deviatoric stress is not consistent with underlying ice failure assumptions in these crevasse depth theories and will under-predict crevasse sizes by a factor of two for the Nye crevasse formulation (with constant density and temperature) and around two for LEFM, compared to correctly using resistive stresses. The results for such calculations will not be shown.

| 3D (includes crevasse-parallel) | Maximum Principal | Flow Direction |
|---|---|---|
| $R_{xx} = 2\tau_1 + \tau_2$    (22) | $R_{xx} = 2\tau_1$    (23) | $R_{xx} = 2\tau_{flow\ dir.}$    (24) |

**Table 1: Equations for resistive stress assuming crevasse formation in the maximum principal and flow directions with and without the crevasse-parallel deviatoric stress.**



## 3 Methods

We calculate crevasse depths with a subset of all possible combinations of effective strain rate, stress direction, and resistive stress calculations discussed above. The stress calculation versions being tested are given in Table 2 and will be referred to by

the names listed there throughout the rest of this study. Only one flow direction stress calculation is included, as the studies where the selection is significant (plan view) tend to use the maximum principal stress direction. Also, as noted, van der Veen (1999) showed that crevasses tend to align (with some variation) to the maximum principal stress direction. Table 2 does not contain all possible permutations, but instead only several that occur in the literature. For example, there are no cases where the crevasse-parallel stress is considered (low simplification) but the effective strain rate is neglected (high simplification).

The impact of the selection will be shown through idealized deformation state test cases, plots of predicted crevasse penetration on real ice shelves, and modeling ice shelf velocities with crevasse penetration as damage.

| Calculation | Effective Strain Rate | Stress Direction | Crevasse-parallel Stress | Resistive stress |
|:---:|:---:|:---:|:---:|:---:|
| A | None | Flow | No | $R_{xx} = 2\tau_{flow\,dir.}$ |
| B | None | Max Prin | No | $R_{xx} = 2\tau_1$ |
| C | Planar (eq. 8) | Max Prin | No | $R_{xx} = 2\tau_1$ |
| D | Full (eq. 9) | Max Prin | No | $R_{xx} = 2\tau_1$ |
| E | Planar (eq. 8) | Max Prin | Yes | $R_{xx} = 2\tau_1 + \tau_2$ |
| F | Full (eq. 9) | Max Prin | Yes | $R_{xx} = 2\tau_1 + \tau_2$ |

**Table 2: Summary of effective strain rate, stress direction, and crevasse-parallel deviatoric stress used for each calculation considered as well as the corresponding resistive stress equations.**

### 3.1 Idealized deformation state test cases

Before calculating crevasse depths on real shelves, it is useful to review the expected differences between calculations for idealized strain rate states of biaxially spreading flow, uniaxial extension, and pure shear. Biaxial spreading occurs for unconfined ice tongues and, to a lower extent, areas of spreading flow via non-parallel shear margins, such as on the Larsen B remnant ice shefl. Uniaxial extension occurs in the center of glaciers in fjords or shelves with parallel shear margins such as the Pine Island Glacier shelf. Pure shear is approached in shear margins.

### 3.2 Ice shelf crevasse penetration maps

#### 3.2.1 Crevasse penetration workflow

We calculate crevasse penetration maps for several Antarctic ice shelves. Crevasse penetration is the ratio of crevassed ice thickness to the total thickness:

$$crevasse\ penetration = \frac{d_s + d_b}{H}. \tag{25}$$



To do this, we use a workflow described by the flowchart in Fig. 3. The calculation of the stress tensor from strain rates and the calculation of the resistive stress from the stress tensor are varied for each version being tested. The surface topography comes from the reference elevation map of Antarctica (REMA) mosaic product (Howat et al., 2019) as included in BedMachine (Morlighem et al., 2020; Morlighem, 2022). Velocity comes from the ITS_LIVE annual mosaic products (Gardner et al., 2018, 2019) and MEaSUREs multi-year averaged products (Rignot et al., 2022). As the REMA mean year is 2015, velocities from

2015 are used except where a different time period gives better matching ice extents between the topography and velocity data.

Surface crevasse depths are calculated with rigidity corresponding to the surface temperature from Comiso (2000), and basal crevasse heights are calculated with the assumption that the presence of the crevasse allows ocean water to bring the ice temperature to the melting temperature, 0°C. This may be a reasonable assumption in areas where crevasses change in size slowly as they advect to locations with different stresses. If stress increases suddenly, however, such that the crevasse grows

immediately to a larger size without its tip reaching 0°C, the actual crevasse penetration will be larger than what is modeled. The rigidity as a function of temperature comes from Cuffey and Patterson (2010).

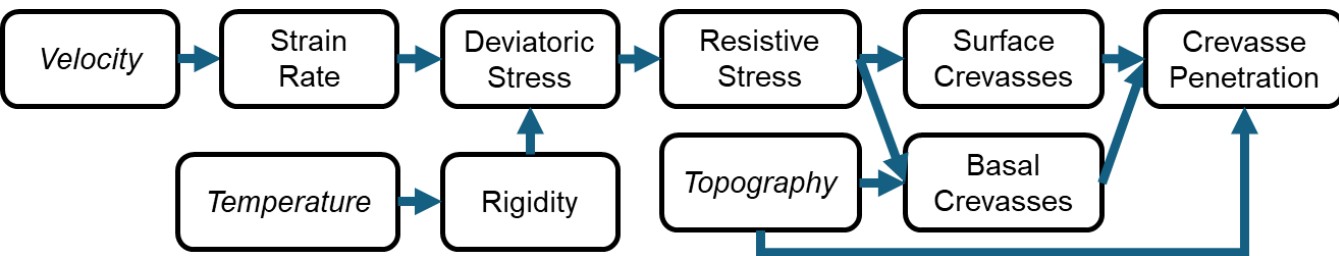

**Figure 3: Flowchart showing the steps of the workflow used to make crevasse penetration maps. Italicized text indicates inputs from remote sensing products.**

**3.2.2 Ice shelf selection**

We compare the predicted crevasse penetration from each calculation on three ice shelves: the Larsen B remnant shelf, the Brunt/Stancomb-Wills shelf, and Pine Island Glacier's shelf. We use floating rather than grounded ice as the predicted large basal crevasses in shelves create a stronger connection between crevasses and bulk rheology. Shelves also remove the confounding effect of basal drag. The Larsen B remnant was selected as it has both shear margins and spreading flow. These

features highlight the difference between calculations in different strain rate states. Shear margins became a focus after finding larger differences in predicted crevasse penetration in those of the Larsen B remnant. This led to studying Brunt/Stancomb-Wills for its fully failed (rift) shear margins and Pine Island Glacier's ice shelf for its southern shear margin which partially broke up around 2018 (Lhermitte et al., 2020).

**3.3 Testing with velocity prediction using crevasse penetration as damage**

Some checking of crevasse depth calculations can be done by assessing whether the results are realistic. For example, some calculations will predict shear margins that have basal crevasses alone penetrating full thickness, which is physically inconsistent with an observation of an intact shear margin with surface crevasses. For calculations that yield plausible results,



validation would require measurement of crevasse depths as well as detailed knowledge of ice temperature, which are rarely (if ever) available. In an attempt to get around this problem, we use crevasse penetration (Equation 25) as damage, as proposed in Sun et al. (2017). The damage field is used to model velocity in the Ice-sheet and Sea-level System Model (ISSM) (Larour et al., 2012) allowing for comparison to the observed velocity field. ISSM is run with the shallow shelf approximation (MacAyeal, 1989).

Because temperature measurements through thickness on shelves are rare, we hold a constant depth averaged temperature across each ice shelf domain. We tune this temperature for the best velocity match, quantified as the lowest average error from each node. While this means we are not directly testing the success of the crevasse depth calculations in predicting the magnitude of damage, the crevasse calculations still provide a pattern of ice rigidity that would define the velocity field. To have reference points, we also calculate velocities with no damage and with inverted damage. For the undamaged velocity predictions, temperature again must be assumed. Without damage, a falsely warm temperature will yield lower error than the real temperature. We intentionally select a temperature that is likely warmer than the fast-flowing portions of the shelves. This is a conservative choice to ensure that the crevasse-based damages are not made to look more successful than they really are by overly high error from the undamaged predictions. We use a quadratic temperature profile where the temperature at one third depth is equal that of the surface and the temperature at the base is near melting. An example of this temperature profile is provided in Fig. S1. For inverting damage, the temperatures tuned for matching velocity with crevasse penetration from calculation F (Table 2) were assumed. Inversions were initialized with 40% damage, as done in Borstad et al. (2016). Like noted in Borstad et al. (2016), we found that the success of the inversion in matching velocity was not very sensitive to this selection of 40% (we tested 30% to 70%). It is not expected that a different temperature assumption would be significant, as the effect is not different than using a different damage percentage initialization.

The ice shelves selected for our analysis are small shelves that show high amounts of crevasse penetration, which makes it more likely that damage, not temperature, drives the pattern of rheology, so that the error in the total rigidity field from assuming constant temperature is lessened. The shelves used to compare predicted crevasse penetration (Larsen B, Brunt/Stancomb-Wills, Pine Island) meet these criteria, as do the Larsen C and Fimbul ice shelves.

## 4 Results

### 4.1 Crevasse depths for representative strain states

As noted in Section 3.1, pure shear, uniaxial extension, and biaxial spreading are simplified strain rate states that are representative of shear margins, centerlines of confined glaciers, and unconfined ice fronts, respectively. To compare the stress calculations in these flow types, the magnitudes of each strain rate component are held constant. The strain rate component magnitude, 0.012 yr$^{-1}$, corresponds approximately to the center of flow near the terminus of the Larsen B remnant ice shelf, which has high and spreading strain rates. Table 3 shows surface and basal crevasse sizes for these three representative strain rate states. For pure shear, the flow direction stress (calculation A) predicts no crevasse depth as the flow direction normal





stress is zero. The differences from effective strain rate and crevasse-parallel stress are better shown graphically and are

discussed with Fig. 4 next. The remaining takeaway from this table is that, even with the warmer ice temperature assumed,

basal crevasses are several times larger than surface crevasses and will make up most of the total crevasse penetration.

| Stress calculation | | | | Surface crevasse depths (m) | | | Basal crevasse heights (m) | | |
|---|---|---|---|---|---|---|---|---|---|
| Calc. | Direction | Effective strain rate | Crevasse-parallel stress | Biaxial spreading | Uniaxial extension | Pure shear | Biaxial spreading | Uniaxial extension | Pure shear |
| A | Flow | None | No | 30.0 | 30.0 | 0.0 | 99.8 | 99.8 | 0.0 |
| B | | None | No | 30.0 | 30.0 | 30.0 | 99.8 | 99.8 | 99.8 |
| C | | Planar | No | 30.0 | 37.7 | 30.0 | 111.2 | 125.7 | 99.8 |
| D | Max principal | Full | No | 20.8 | 30.0 | 30.0 | 77.8 | 99.8 | 99.8 |
| E | | Planar | Yes | 44.9 | 37.7 | 15.0 | 148.3 | 125.7 | 49.9 |
| F | | Full | Yes | 31.2 | 30.0 | 15.0 | 103.7 | 99.8 | 49.9 |

**Table 3: Surface and basal crevasses with each stress calculation for representative strain rates for biaxial spreading ($\dot{\varepsilon}_{xx} = 0.012\ yr^{-1}, \dot{\varepsilon}_{yy} = 0.012\ yr^{-1}, \dot{\varepsilon}_{xy} = 0.0\ yr^{-1}$), uniaxial extension ($\dot{\varepsilon}_{xx} = 0.012\ yr^{-1}, \dot{\varepsilon}_{yy} = 0.0\ yr^{-1}, \dot{\varepsilon}_{xy} = 0.0\ yr^{-1}$), and**

**pure shear ($\dot{\varepsilon}_{xx} = 0.0\ yr^{-1}, \dot{\varepsilon}_{xx} = 0.0\ yr^{-1}, \dot{\varepsilon}_{xy} = 0.012\ yr^{-1}$). The flow direction is $\dot{\varepsilon}_{xx}$ in this example (breaking notation). Ice rigidity corresponds to -18°C for surface crevasses and 0°C for basal crevasses.**

For all crevasse size calculations that assume the crevasse forms perpendicular to the maximum principal stress

direction (calculations B to F), the difference in crevasse depths as a function of strain rate state can be shown by varying the

minimum principal strain rate (from the planar tensor) while holding the maximum principal strain rate constant. Fig. 4 shows

this for the minimum principal strain rate varying from equal in magnitude to the maximum principal strain rate in compression

to tension. This corresponds to a pure shear strain rate state and a biaxially spreading strain rate state, respectively. The values

for basal crevasses are presented, but the ratios of depths between calculations will be identical to those of dry surface crevasse

calculations so long as depth variable temperature and density are neglected.





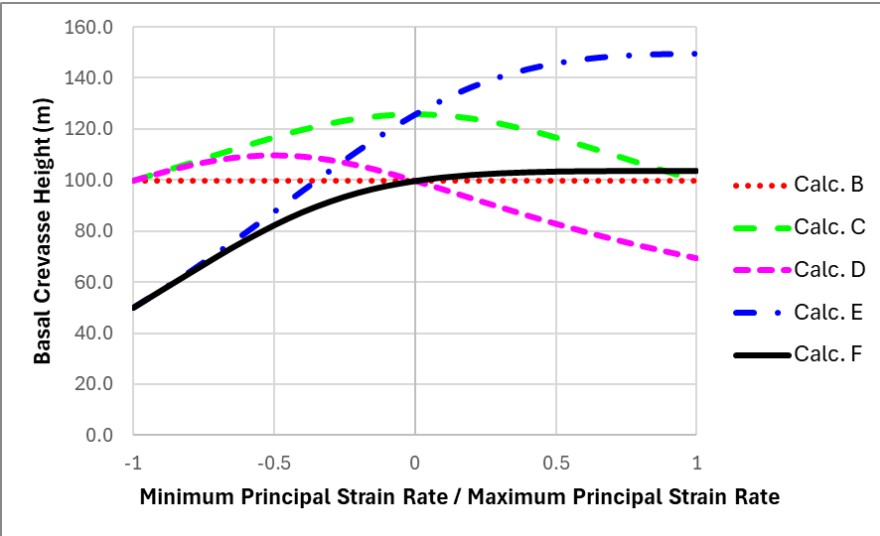

**Figure 4: Basal crevasse heights for each stress calculation using maximum principal directions stress. The maximum principal strain rate, $\dot{e}_1$, is 0.0117 yr$^{-1}$ and the minimum principal strain rate, $\dot{e}_2$, ranges from -0.0117 yr$^{-1}$ to 0.0117 yr$^{-1}$. On the x axis, -1 is a state of pure shear and +1 is biaxial spreading.**

The maximum principal stress direction with no effective strain rate calculation (calculation B) is independent of the minimum principal strain rate. The calculations including crevasse-parallel deviatoric stress (calculation E and F) reduce the crevasse depth in the pure shear strain rate state because the negative minimum principal stress term, $\tau_2$, reduces the total resistive stress. Because the pure shear state has planar strain rates that are equal in magnitude but opposite in sign, mass conservation is met with no vertical strain rate. This causes the pure shear crevasse depths to be independent of the selection of planar or full effective strain rate explaining the equivalency of calculations E and F as well as calculations C and D. As the minimum principal strain rate becomes more positive, the vertical strain rate magnitude grows increasing the effective strain rate when continuity is respected. This reduces the stress from the same value of maximum principal strain rate, explaining the smaller crevasse depths from the full effective strain rate calculations (D and F) compared to their planar effective strain rate counterparts (calculations C and E) when moving towards the right side of the plot. For positive values of minimum principal strain rate, the simple calculation without effective strain rate (calculation B) is nearly equivalent to the most physically based calculation that includes effective strain rate and crevasse-parallel deviatoric stress (calculation F). For calculation F, as the minimum principal strain rate increases, the increase in effective strain rate reduces stress in the maximum principal direction. This effect is apparently cancelled by the growing minimum principal stress term to explain the nearly constant value of calculation F from zero to one on the x axis.

**4.2 Crevasse penetration on the Larsen B remnant ice shelf**

Fig. 5 provides the crevasse penetration ($(d_s + d_b)/H$) for the Larsen B remnant ice shelf with each stress calculation listed in Table 2. The Larsen B remnant was selected as it includes both shear margins (approximately pure shear) and spreading





flow, which, as our idealized test case shows, will highlight where differences between calculations occur (Table 3). The calculations using the maximum principal direction but neglecting crevasse-parallel deviatoric stress (calculations B, C, and D) appear similar and are characterized by wide zones of full crevasse penetration in the shear margins. The flow direction calculation (A) and the two maximum principal direction calculations that use the full resistive stress (calculations E and F)

show similar results to one another. Calculation A, however, does predict more zones of no damage where the flow direction stress components are not tensile, as well as a larger fully failed area in the north shear zone near the terminus.

Next, we compare crevasse penetration predicted by each calculation against that of calculation F. These differences in crevasse penetration are shown in Fig. 6. Significant differences between the calculations occur in certain areas. In the fast-flowing center, particularly near the terminus, calculation E predicts higher crevasse penetration than calculation F. This is

likely because of increased lateral spread between the diverging, non-parallel shear margins causing the crevasse-parallel deviatoric stress to increase relative to the maximum principal (crevasse-perpendicular) stress. This corresponds to moving toward the right side of the Fig. 4 plot. There is also a large difference in crevasse penetration between calculation F and the calculations in the max principal direction that do not include crevasse-parallel stress (calculations B, C, and D) in the region between the two inlets. Calculations B, C, and D predict higher crevasse penetrations because they neglect the compressive

lateral stress in this region, which results from the converging flow. This causes a negative minimum principal stress term and corresponds to the left side of Fig. 4. Calculation A predicts lower crevasse penetration in the glacier inlets themselves. This is because lateral spreading is occurring faster than longitudinal spreading such that the maximum principal stress direction is rotate approximately 90 degrees from the flow direction. In the rest of the fast-flowing center region closer to the front, the flow direction and maximum principal directions more closely align such that calculation A is nearer to calculation F.






**Figure 5: Crevasse penetration at the Larsen B remnant ice shelf with (A) calculation A, (B) calculation B, (C) calculation C, (D) calculation D, (E) calculation E, and (F) calculation F resistive stress versions overlaid on satellite imagery from October 2014 (Landsat-8 image courtesy of the U.S. Geological Survey). The glacier inlets into the Larsen B remnant (Flask and Lepperd glaciers) are shown on (A). Ice flow direction is approximately from image bottom to top, as shown with orange arrows in panel A.**



**Figure 6: Difference from calculation F crevasse penetration for (A) calculation A, (B) calculation B, (C) calculation C, (D) calculation D, and (E) calculation E crevasse penetration at the Larsen B remnant ice shelf.**



### 4.3 Crevasse penetration in ice sheet shear margins

Next, we use cross section plots to examine differences in crevasse penetration across shear margins. Fig. 7 shows the observed surface velocity and surface topography on one of the shear margins for the Larsen B remnant ice shelf. It also shows the velocity, thickness, and crevasse penetration (with each stress calculation) on a cross section through the fast-flowing center of the shelf. The calculations including crevasse-parallel stress (calculations E and F) predict high but not total crevasse penetration in the south shear margin. In the portion of the north shear margin included in the cross section, all calculations

predict full crevasse penetration. The surface topography of the south shear margin shows visible features oriented 45 degrees from flow, but the ice appears continuous except for the regions near the rifts. This suggests the calculations that use the maximum principal direction but do not include the crevasse-parallel deviatoric stress (calculations B, C, and D) overpredict crevasse depths here, and possibly in shear margins in general. Also, the higher predicted crevasse penetrations in the spreading flow area for calculations with the planar effective stress (calculations C and E) are visible at the 5-15 km path distance on the

cross-section plot.



**Figure 7: (A) observed velocity map (MEaSUREs 2014-2017 – Gardner et al., 2019, 2018) with cross section location, (B) 2015 hillshade REMA (Howat et al., 2019) snapshot of the shear margin between rifts, (C) cross section velocity, (D) cross section thickness, and (E) cross section crevasse penetration at the Larsen B remnant ice shelf.**

Focusing on the difference in shear margin crevasse penetration between calculations that do and do not consider crevasse-parallel deviatoric stress, we next consider the Stancomb-Wills shelf, as it has entirely failed shear zones. Both shear



zones bordering the fast flowing center were identified as rifts by Larour et al. (2014). Fig. 8 shows the velocity and topography over one of the shearing rifts as well the cross-section velocity. The velocity cross section shows a sharp drop without rounded
corners. If the entirety of the shear margin of the Larsen B remnant was fully failed, a similar velocity profile might be expected. This provides more evidence then, against the quality of the calculations without crevasse-parallel stress that predict fully failed shear margins (calculations B, C, and D) and for the calculations that include crevasse-parallel stress (calculations E and F). Given that the northern shear margin at the Larsen B is predicted to be fully crevasse-penetrated by all calculations and yet a smooth velocity profile remains, all principal direction calculations may over-predict crevasse penetration, with calculations
E and F overpredicting by less than calculations B, C, and D.

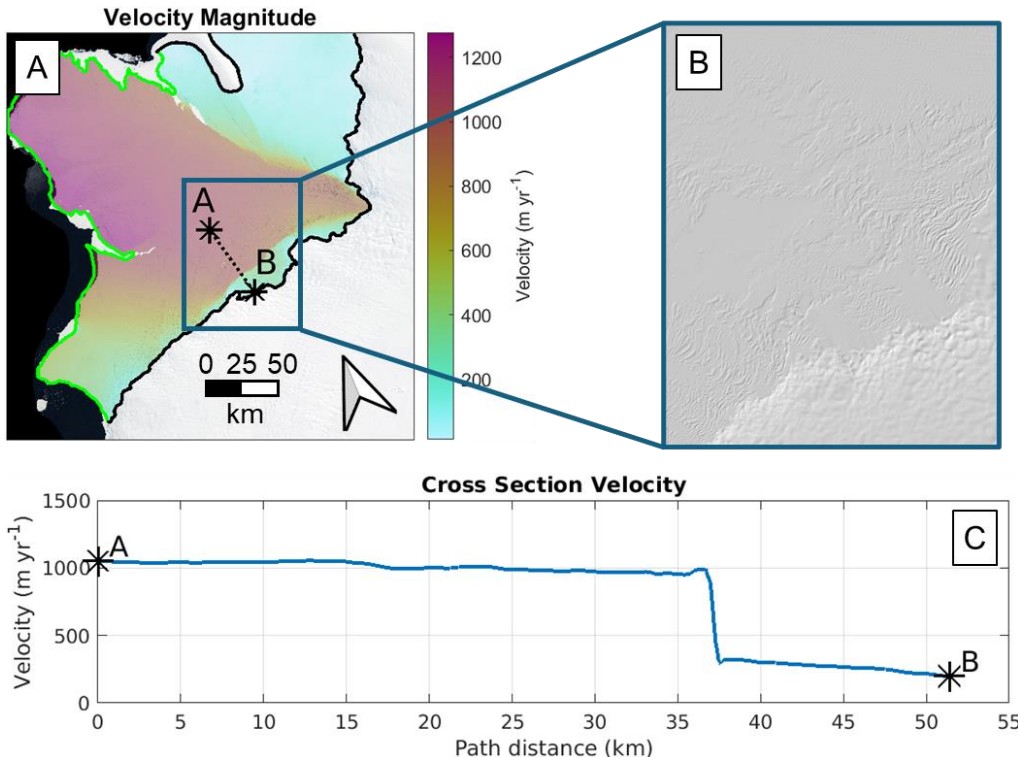

**Figure 8: (A) observed velocity map (ITS_LIVE 2015 – Rignot et al., 2022) with cross section location, (B) 2015 hillshade REMA (Howat et al., 2019) snapshot of the lower rift, and (C) cross section velocity for the Brunt/Stancomb-Wills ice shelf. Plot A is overlaid on the Landsat Image Mosaic of Antarctica (Bindschadler et al., 2008). Ice flow direction is approximately from image right to left.**

The last shelf used to evaluate crevasse penetration is the Pine Island Glacier shelf. It again shows full penetration in its shear margins with all versions of crevasse depth calculations that do not include the effect of the crevasse-parallel deviatoric stress. This can be seen from the map view of crevasse penetration with calculation C (Fig. 9D), which has the same crevasse penetration as calculations B and D for pure shear. Calculation F shows elevated but not complete crevasse penetration in the shear margins (Fig. 9E), which is a similar result to that of the Larsen B remnant. Pine Island's south shear margin failed in
some regions in 2018, which can be seen in the 2014 and 2018 surface topography views provided in Fig. 9B. This indicates that partial crevasse penetration rather than total crevasse penetration prior to 2018 should be predicted, again favoring the use



of the crevasse-parallel deviatoric stress term. In the center of flow, differences between the different stress calculations are small (Fig. S2 and S3). This is because, unlike the Larsen B remnant, Pine Island has parallel shear margins and thus little lateral spreading in the center of flow. Therefore, the strain rate state in the center of flow corresponds to the center portion of

Fig. 4, where there is less range across the calculations.

**Figure 9:** (A) observed velocity (ITS_LIVE 2015 – Rignot et al., 2022) map with cross section location, (B) hillshade REMA (Howat et al., 2019) views from 2014 and 2018 of the south shear margin , (C) cross section velocity, (D) map view crevasse penetration with calculation C, and (E) map view crevasse penetration with calculation F for the Pine Island Glacier ice shelf. The plan view plots are
overlaid on satellite imagery from November 2014 (Landsat-8 image courtesy of the U.S. Geological Survey). Ice flow direction is approximately image top right to bottom left.





## 4.4 Velocity comparison results

Velocity predictions were made with crevasse penetration as damage from the two stress calculations that include crevasse-parallel stresses (calculations E and F). The maximum principal stress calculations that do not use the crevasse-parallel stress
(calculations B, C, and D) are not included, as they all yield fully failed shear margins for Pine Island Glacier's shelf and the Larsen B remnant. In these locations, the modeled velocity would fully depend on the selection of the maximum allowable damage, which is a user-defined parameter that protects against an element having full damage and therefore zero viscosity. The flow direction stress calculation (A) is also not used for similar reasons and because it is not consistent with the observed orientations of crevasses (Section 4.3).

440   The average nodal velocity misfits for damage calculated with calculations E and F as well as with no damage and inverted damage are shown in Fig. 10A. As noted in Section 3.3, the temperature and thus undamaged rigidity is tuned for the best velocity match averaged across the entire model domain. Damage from crevasse penetration will control the relative rigidity between regions of the ice shelves. The assumption is that the stress calculation that gives the best modeled velocity performs best in predicting relative crevasse depths between regions of the ice shelf (e.g., center of flow, shear margins,
unconfined front). Velocity misfits with no damage and from an inversion are included to contextualize the crevasse-penetration-based velocity misfit values.

  We found that the inversion performs best on four of the five domains tested, that modeling damage from crevasse depth calculations and tuning temperature performed best on the Brunt/Stancomb-Wills, and that the crevasse-modeled damage approaches the success of inversions on the other four domains. The full-effective-strain-rate-based calculation (F) outperforms
the planar-strain-rate-based calculation (E) for all shelves, albeit marginally in some cases (Fig. 10B). The largest improvements relative to no damage are at small shelves with high crevasse penetration (Pine Island, Brunt/Stancomb-Wills, Larsen B remnant) as opposed to shelves that are larger (Larsen C) or have less crevasse penetration (Fimbul) as shown by the error reduction percentages in Fig. 10C. Larger shelves may also have more temperature variability that is not being captured. Lower crevasse penetration may make temperature error more significant in predicting the net bulk rheology.



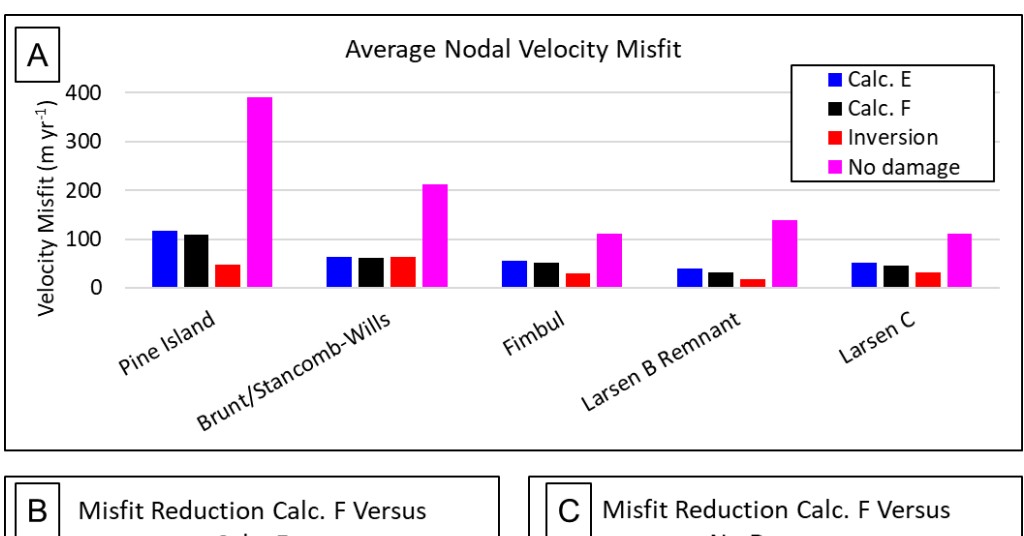

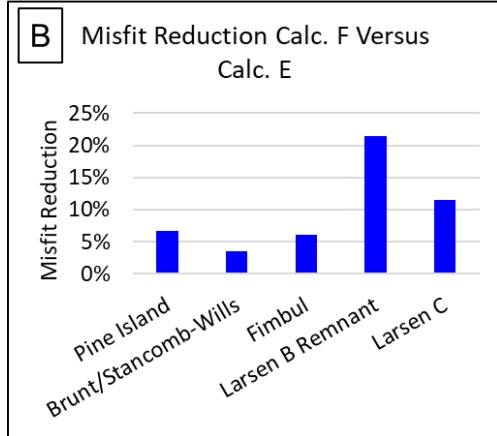
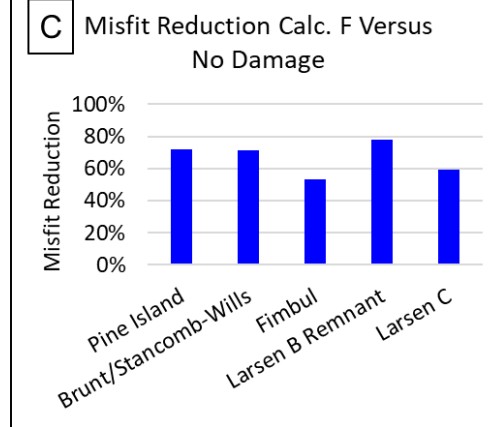

**Figure 10: (A) Average nodal velocity misfit with damage from calculation E crevasse penetration, damage from calculation F crevasse penetration, no damage, and inverted damage and (B) misfit reduction percentages with calculation E relative to calculation F, and (C) misfit reduction percentages with calculation F relative to no damage.**

### 4.4.1 Larsen B remnant velocity predictions

Fig. 11 shows the modeled velocity correlation and mapped misfit for calculations E and F at the Larsen B remnant ice shelf. The calculation E correlation plot (Fig. 11A) shows that it predicts excess velocity for the fastest-moving ice near the terminus. This likely stems from the average temperature tuning, which balanced excess velocity at the terminus with overly slow velocities upstream to minimize the overall misfit. This may indicate that the damage in the spreading flow region approaching the terminus is being over-predicted relative to the shear margins and more confined flow upstream. The full effective strain rate calculation (F) predicts smaller crevasses in regions of spreading flow due to increased ice softening from the vertical strain rate term, as was seen in Fig. 6E. This fixes the problem of the fast front and slow upstream seen in the planar effective strain rate calculation (E) and reduces the average nodal velocity misfit by 21%, from 39 m yr$^{-1}$ (calculation E) to 31 m yr$^{-1}$ (calculation F). While local regions with substantial misfit remain, it is distributed across observed velocities rather than being



concentrated. This provides some evidence that mass conservation should be included in the effective strain calculation even
when crevasses are present (calculation F rather than E). The modeled velocity maps themselves are available as Fig. S4.

The tuned ice rigidity for calculation F damage corresponds to -17°C if temperature were constant through thickness
$(T(\bar{B}) = -17℃)$. This agrees well with the average surface temperature over the shelf of -17.7°C from Comiso (2000); the
cold bias (being close to the surface rather than basal temperature) likely stems from advection of colder ice from upstream.
However, the tuned temperature is colder than thermal-model-derived temperatures in Borstad et al. (2012) which were no
colder than approximately -12°C. This does not significantly alter our findings, however, as we are not testing the ability of
crevasse depth to predict the absolute magnitude of damage, only the pattern of damage.

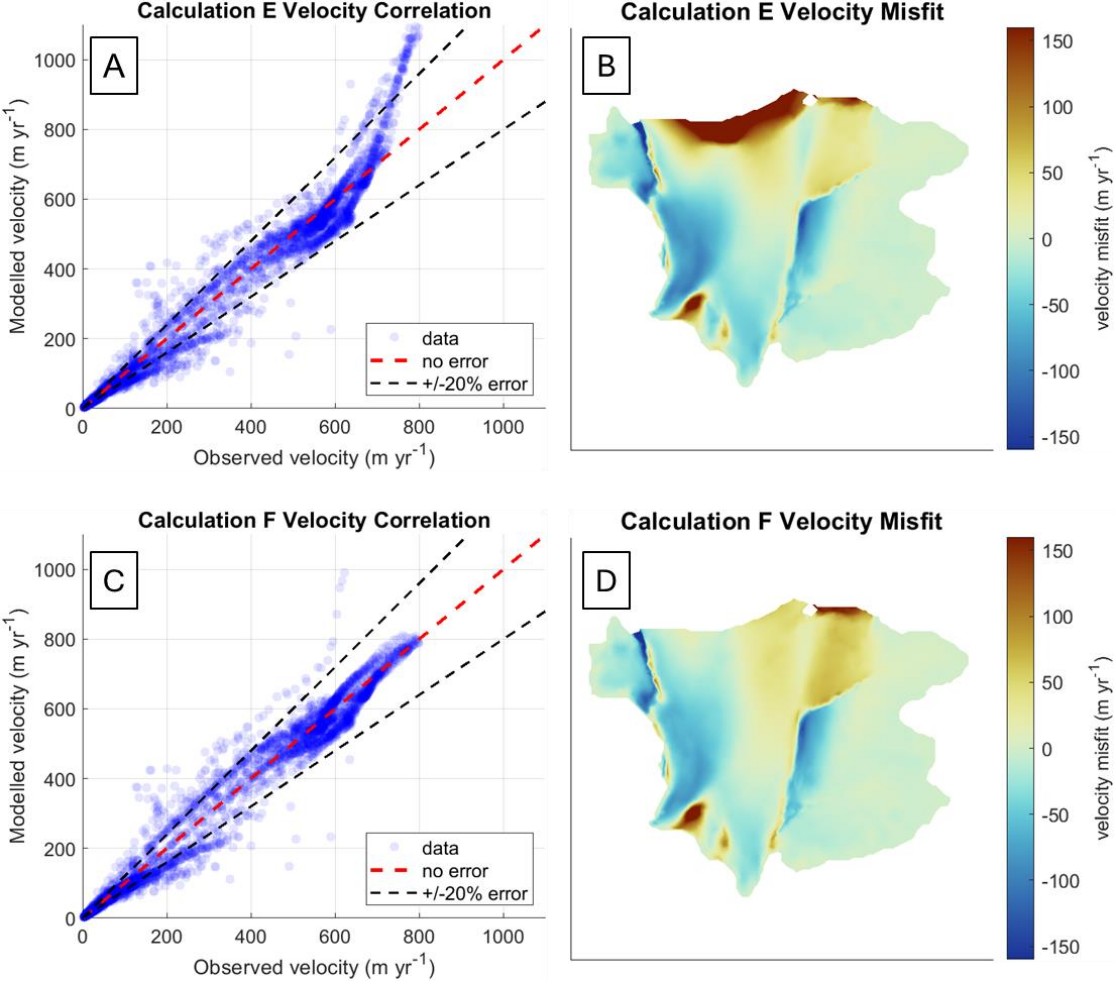

**Figure 11: Plots of (A) velocity correlation and (B) velocity misfit with calculation E as well (C) velocity correlation and (D) velocity misfit with calculation F for the Remnant Larsen B ice shelf. The velocity product used for crevasse penetration calculation and**
**correlation plots is the MEaSUREs 2014-2017 averaged product (Gardner et al., 2018, 2019).**





### 4.4.2 Pine Island Glacier ice shelf velocity predictions

The effect of the full effective strain rate based on mass conservation versus the planar effective strain rate is less significant at the Pine Island Glacier ice shelf as can be seen in Fig. 12. This is likely due to the parallel shear margins and thus lack of spreading flow that would cause increased stress when the vertical strain rate term is not included in the effective strain rate

calculation. Including the full effective strain rate reduces average nodal misfit by 7% from 117 m yr$^{-1}$ (calculation E) to 109 m yr$^{-1}$ (calculation F). Unlike at the Larsen B remnant, there is no region or clear spatial pattern in the difference between calculation E and F; the 7% appears to come from small improvements spread over the whole domain. The tuned rigidity corresponds to a temperature of -16°C, which compares well to the average surface temperature from Comiso (2000) of -17.8°C. The crevasse penetration plots used as damage and their differences are provided in Fig. S2, and the modeled velocity

plots themselves are in Fig. S5.

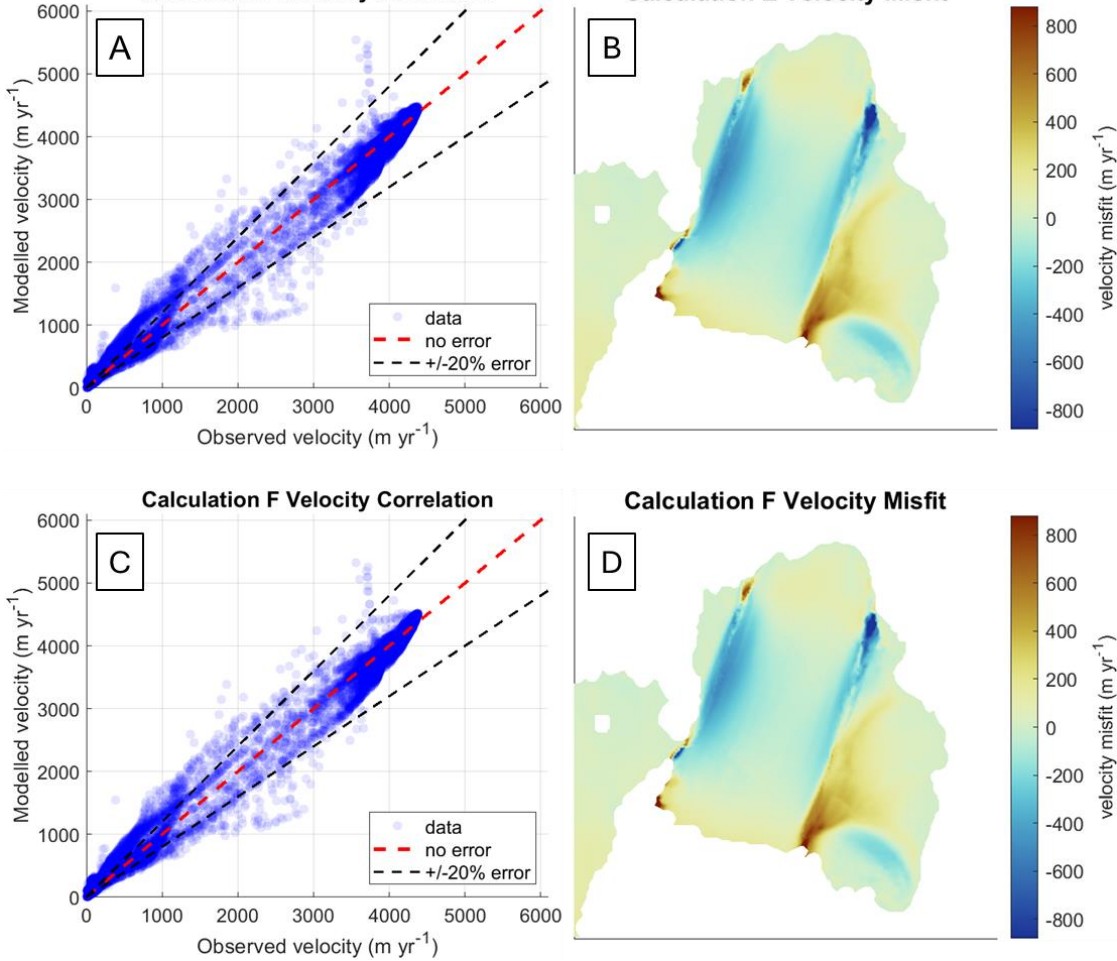

**Figure 12: Plots of (A) velocity correlation and (B) velocity misfit with calculation E as well (C) velocity correlation and (D) velocity misfit with calculation F for the Pine Island Glacier ice shelf. The velocity product used for crevasse penetration calculation and correlation is the ITS_LIVE 2015 annual map (Rignot et al., 2022).**



## 5 Discussion

### 5.1 Recommended resistive stress calculation

Our findings support resistive stress calculation F for use in crevasse depth predictions. Predicted crevasse penetration with calculation F are better aligned with observed surface features and velocity cross sections than all other calculations except calculation E (Sections 4.2 and 4.3). Calculations B, C, and D, which neglect the crevasse-parallel deviatoric stress in the three-dimensional resistive stress equation, overpredict crevasse penetration in shear margins. Calculations E and F yield identical results in shear margins and similar results in uniaxial extension, but calculation E predicts larger crevasses in biaxial spreading through neglecting the ice softening effect of vertical strain rate (Section 4.1). Applying crevasse penetration as damage from these two calculations supports calculation F. Calculation F outperforms calculation E in reducing modeled velocity misfit for all shelves with large improvements at the Larsen B remnant and Larsen C (Section 4.4). At the Larsen B remnant, the improved modeled velocity field of calculation F can be explained by its lower crevasse penetration prediction in biaxially spreading flow. Finally, calculation F is the most physically consistent. It can be derived from deviatoric stress equations with the assumptions of continuity, crevasse formation in the maximum principal stress direction, and vertical stress ($\sigma_{zz}$) coming from only lithostatic pressure and water pressure (Section 2.4.3).

### 5.2 Classification of stress calculation by study type

As noted throughout the introduction and background, the stress calculation used as input for crevasse depth calculations has varied widely across studies. In some cases, the differences are zero or trivial. For example, the maximum principal stress and flow direction stress in the center of a two-dimensional flowline domain will be equivalent. However, for studies that use planar remote sensing data (Amaral et al., 2020; Lai et al., 2020) or ice sheet models (Choi et al., 2018; Sun et al., 2017; Todd et al., 2018), the selections of effective strain rate, stress direction, and resistive stresss equation may differ significantly.

Table 4 shows the stress calculations used by eight studies. It was not always possible to tell with complete certainty which stress calculation was used from the study text. For example, Todd et al. (2018) simply state crevasses extend until the maximum principal full stress is zero. As they used a full Stokes model, it is assumed all effects were considered, but this is not certain. Secondly, not all studies fit a category perfectly. Mottram and Benn (2009), for example, directly used the direction perpendicular to the crevasse by measuring strain rate with stakes on either side, bypassing the stress direction consideration. This method avoids the assumption that the crevasse forms perpendicular to the flow direction or maximum principal stress that must be made when predicting the spatial pattern of crevasse depths. Studies that used multiple stress calculations are listed in each corresponding cell. Choi et al. (2018) and Lai et al. (2020) both evaluated their results with a flow-direction and maximum-principal-direction calculation. Enderlin and Bartholomaus (2020) used different stress versions for the Nye formulation and LEFM components of their workflow.



525    The eight studies tabulated use seven distinct stress calculations with multiple studies selecting Calculations A, B, and D. Three studies (Choi et al., 2018; Enderlin and Bartholomaus, 2020; Sun et al., 2017) use deviatoric rather than resistive stress terms. Only one (Todd et al., 2018) study uses the most physically consistent calculation (F).

| Stress calculations | | | | Study types | | |
|---|---|---|---|---|---|---|
| **Calculation** | **Effective strain rate** | **Stress direction** | **Crevasse-parallel stress** | **Comparison to measured** | **Crevasse depth calving law** | **Other** |
| A | None | Flow | No | Enderlin and Bartholomaus (2020) | Nick et al. (2010) | Lai et al. (2020) |
| *Not tested* | Planar | Flow | No | Enderlin and Bartholomaus (2020) * | | |
| *Not tested* | Full | Flow | No | | Choi et al. (2018) * | |
| *Not tested* | Planar | Flow | Yes | | | |
| *Not tested* | Full | Flow | Yes | | | |
| B | None | Max Prin | No | Mottram and Benn (2009) | | Lai et al. (2020) |
| C | Planar | Max Prin | No | | | |
| D | Full | Max Prin | No | | Choi et al. (2018) * | Sun et al. (2017) * |
| E | Planar | Max Prin | Yes | | Amaral et al. (2020) | |
| F | Full | Max Prin | Yes | | Todd et al. (2018) | |

**Table 4: Classification of studies using crevasse depth calculations by stress calculation method. Asterisks indicates studies that used**
**(for at least one calculation included in the study) a deviatoric stress term rather than the resistive stress. The predicted crevasse depths would correspond to one half the values yielded by the calculation classification.**

    Many more studies reviewed used crevasse depth calculations but did not provide adequate details for classification. As we have shown, these factors can change crevasse size by a factor of two, so future studies be more diligent in describing which stresses and what equations were used. We recommend calculation F based on its physical basis and success in recreating
535 ice sheet velocity patterns when implemented as damage.



**5.3 Effect on studies comparing observed crevasse depths to predictions**

Mottram and Benn (2009), or using calculation B, neglected effective strain rate and crevasse parallel stress in their testing of the Nye formulation and van der Veen (1998) LEFM. So long as the crevasse-parallel stress (minimum principal surface stress) is positive, the effect of this would be negligible. Future studies evaluating crevasse depths against observations could avoid
potential error by confirming this to be the case or by using calculation F if the crevasse-parallel stress is available.

Enderlin and Bartholomaus (2020) used different stress calculations for the Nye formulation and LEFM components of their analysis. Their Nye formulation calculation neglects effective strain rate and crevasse-parallel stress but does use resistive stress (calculation A). The LEFM calculation uses effective strain rate but takes the flow-direction deviatoric stress as the resistive stress. The Nye formulation and LEFM have different assumptions about ice's failure criterion and the local
effects of a crevasse on far-field stress, but do not call for differing calculations of that far-field stress. We encourage the use of calculation F for resistive stress regardless of the subsequent crevasse depth calculations being performed.

**5.4 Effect on the crevasse depth calving law**

An ideal calving law will capture retreat across the terminus and across different glaciers accurately with minimal difference in tuning, and the stress calculation used affects both criteria. In their testing of the crevasse depth law, Choi et al. (2018) use
deviatoric stress. We have shown neglecting the crevasse-parallel stress causes an over-prediction of shear margin crevasse depth. This may cause a single tuning of the crevasse depth law to balance over-retreat of the shear margins with under-retreat of the glacier center. This would correspond to an overly convex shape in the modeled glacier front. Also, using a deviatoric stress term rather than resistive stress will under-predict crevasse depths and require a higher tuned meltwater height. This will lead to calving law that is less sensitive to changes in stress.

We have also shown that neglecting vertical strain in the effective strain rate calculation predicts large crevasse depths in regions of unconfined spreading flow. If the calculation including vertical strain rate best corresponds to crevasse depths, then crevasse depth law implementations that neglect this term will artificially require different tunings between glaciers based on the confinement of their termini. This bias may be present in the calving law testing by Amaral et al. (2020), who used calculation E in their crevasse depth law implementation.

**5.5 Effect on damage laws**

Complex damage laws that are consistent with continuum mechanics, capture water pressure effects, consider both ductile and brittle failure, and avoid overly-general use of LEFM's stress intensity factor functions are in development (e.g., Duddu et al., 2020). Some of the associated challenges and opportunities with these models are discussed by Mobasher et al. (2024). However, where simpler damage implementations tied to crevasse depths are used (e.g., Sun et al., 2017), our results encourage
the use of the calculation F for crevasse depths. The calculation selection will control the ratio of damage applied to shear margins versus the extensional center of glacier and ice shelves. A mechanism of shelf retreat observed at Pine Island and





Petermann is thinning of shear margins via melting in basal channels, increased damage in shear margins from the thinning, and frontal retreat (calving) from reduced buttressing from weakened shear margins (Alley et al., 2019; Lhermitte et al., 2020). The presence of polynyas indicating basal melt channels under other shelves' shear margins suggests widespread vulnerability

to this retreat mechanism (Alley et al., 2019). This observed process of retreat highlights the importance of capturing damage in shear margins accurately which includes using a resistive stress calculation where the crevasse parallel stress impacts the resistive stress.

## 5.6 Effect on ice shelf vulnerability to hydrofracture

Lai et al. (2020) considered the impacts of including or neglecting effective strain rate, firn density and rigidity effects, and

stress direction in their analysis of where ice shelves are simultaneously vulnerable to hydrofracture and provide significant buttressing. For each of these choices, they showed either mathematically or empirically that their findings are minimally affected. The vulnerability of shear margins with calculation F would fall between their maximum principal direction and flow direction calculations and is thus enveloped. However, if future ice sheet modeling efforts use their criterion to locally fail regions of ice shelves, the calculation choice may control whether some shear margins are vulnerable or not. As discussed

above (Section 5.5), shear margins are critical to ice shelf integrity, so overprediction of shear margin vulnerability to hydrofracture may be significant in controlling which and how much of ice shelves are predicted to collapse under increased surface melt. Based on our results showing that the flow direction calculation can miss crevasse penetration in shear margins while neglecting crevasse parallel stress over-predicts crevasse penetration, we suggest that calculation F provides the most accurate mapping of vulnerability in shear zones.

## 6 Conclusions

We reviewed the differences in stress calculations found in literature and calculated the corresponding differences in crevasse depths for representative strain rate states. Next, we showed the spatial patterns of crevasse penetration with each calculation using the Nye crevasse formulation on several real ice shelves. Finally, we tested the ability of damage patterns from crevasse penetration to yield velocity fields that match observations. We found that, among six variations of stress state commonly

found in the literature, the predicted crevasse depths can vary by a factor of two. This difference is most pronounced in shear margins and regions of unconfined spreading flow. The best results, where predicted damage patterns were consistent with observed velocity patterns, came from the most physically based calculation, which uses effective strain rate respecting continuity, maximum principal direction stress, and includes the crevasse-parallel deviatoric stress term in the resistive stress equation. This method (calculation F) outperformed a slightly simpler formulation (calculation E), which uses the planar

effective strain rate instead, on all ice shelves tested, but especially on the Larsen B remnant and Larsen C shelves. All other stress calculations yielded too-deep, unrealistic crevasse penetration in shear margins, except in stress states between uniaxial and biaxial tension (no shear or compressive planar strain rate; most commonly found near the center of ice flow), where the





simplest principal direction stress calculation (calculation B) gives crevasse depths that are nearly identical to those of calculation F.

From these findings, we encourage future studies needing crevasse sizes to carefully choose their resistive stress calculation methodology and explicitly state the equations. Due to the significant changes the stress calculation method makes, this is necessary to assure comparisons can be made across studies. We also encourage studies to use the resistive stress rather than a deviatoric stress term to avoid underpredicting crevasse depths relative to the depths that correspond to the physical bases of the Nye crevasse formulation and LEFM. Finally, we encourage studies to use calculation F based on its performance

in modeling observed velocity and its physical basis. This selection is particularly important for applications where any strain rate state from pure shear to unconfined spreading is possible (e.g., plan view remote sensing based or modeling studies). This includes crevasse depth calving law implementations, which can give the calving front a falsely convex shape when less physically based stress calculations are used. Crucially, our findings also affect calculations of ice shelf vulnerability, which may be overpredicted in shear margins when less physically based stress calculations are used.

**Appendix A: Physical properties and variables**

| Name | Symbol | Units | Value | Justification / Comments |
|---|---|---|---|---|
| Gravitational acceleration | $g$ | m s$^{-2}$ | 9.81 | |
| Ice density | $\rho_i$ | kg m$^{-3}$ | 917 | Cuffey and Patterson (2010) |
| Meltwater density | $\rho_{mw}$ | kg m$^{-3}$ | 1000 | Not used for results in paper as dry surface crevasses assumed. |
| Proglacial water density | $\rho_{pw}$ | kg m$^{-3}$ | 1027 | This is the value of ocean density used in BedMachine (Morlighem et al., 2020) for calculating hydrostatic equilibrium. Temperatures and salinities at 500m from the world ocean atlas (Locarnini et al., 2013; Zweng et al., 2013) converted to density with the Thermodynamic Equations of SeaWater – 2010 (TEOS-10) oceanographic toolbox (McDougall and Barker, 2011) yield densities up to 1030.3 kg/m^3. This causes no more than a 3% difference in predicted basal crevasse height. If density is closer to 1000 kg m$^{-3}$ through fresh meltwater, then the impact could be important (up to 30%). |
| Glen's flow law exponent | $n$ | n/a | 3 | Using rheology from Cuffey and Patterson (2010). |

**Table A1: Values of all physical properties used with justification and comments.**



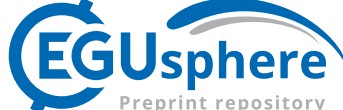

| Name | Symbol | Unit | Comments |
|---|---|---|---|
| Resistive stress | $R_{xx}$ | Pa | Resistive stress in the crevasse perpendicular direction (in this paper's nomenclature). Defined as the full stress minus the lithostatic stress. See van der Veen (2017) for the derivation of Resistive stress in terms of deviatoric stresses. |
| Surface crevasse depth | $d_s$ | m | |
| Basal crevasse height | $d_b$ | m | |
| Meltwater depth | $d_{mw}$ | m | Not used in workflow as surface crevasses assumed to be dry. |
| Height above buoyancy | $H_{ab}$ | m | |
| Ice thickness | $H$ | m | |
| Submerged depth | $D$ | m | |
| Flow factor in Glen's flow law | $A$ | s$^{-1}$ Pa$^{-3}$ | Using rheology from Cuffey and Patterson (2010). |
| Velocity | $V_x, V_y$ | m s$^{-1}$ | |
| Planar deviatoric stresses (crevasse aligned) | $\tau_{xx}, \tau_{yy}, \tau_{xy}$ | Pa | $\tau_{xx}$ is perpendicular to the crevasse and $\tau_{yy}$ runs parallel. |
| Planar maximum principal deviatoric stress | $\tau_1$ | Pa | May also be called the major principal stress or the first eigen value of the planar deviatoric stress tensor. |
| Planar minimum principal deviatoric stress | $\tau_2$ | Pa | May also be called the minor principal stress or the second eigen value of the planar deviatoric stress tensor. |
| Flow direction deviatoric stress | $\tau_{flow\ dir.}$ | Pa | |
| Planar strain rates (crevasse aligned) | $\dot{\varepsilon}_{xx}, \dot{\varepsilon}_{yy}, \dot{\varepsilon}_{xy}$ | s$^{-1}$ | $\dot{\varepsilon}_{xx}$ is perpendicular to the crevasse and $\dot{\varepsilon}_{yy}$ runs parallel. |
| Planar strain rates (global CRS aligned) | $\dot{\varepsilon}_{XX}, \dot{\varepsilon}_{YY}, \dot{\varepsilon}_{XY}$ | s$^{-1}$ | |



| Planar effective strain rate | $\dot{e}_{eff,planar}$ | s⁻¹ | Effective strain rate calculated only from $\dot{\varepsilon}_{xx}$, $\dot{\varepsilon}_{yy}$, $\dot{\varepsilon}_{xy}$ terms neglecting the $\dot{\varepsilon}_{zz}$ term that could be calculated using conservation of mass. |
|---|---|---|---|
| (Full) effective strain rate | $\dot{e}_{eff}$ | s⁻¹ | Effective strain rate including the $\dot{\varepsilon}_{zz}$ term calculated from conservation of mass. |
| Full stress components (crevasse parallel direction) | $\sigma_{xx}$, $\sigma_{yy}$, $\sigma_{zz}$ | Pa | $\sigma_{xx}$ is perpendicular to the crevasse and $\sigma_{yy}$ runs parallel. |
| Damage | $D$ | [unitless] | Defined as the fraction of the ice column that does not carry stress. See Borstad et al. (2016). |

**Table A2: All variables used with their symbols, units, and comments.**

**Code availability**

A Python function that calculates resistive stress and then surface crevasse depths and basal crevasse heights with the Nye
formulation is available at https://doi.org/10.5281/zenodo.12659663 (Reynolds et al., 2024). The repository also includes a
Jupyter notebook and downsampled data files to plot the crevasse penetration results for the Larsen B and Pine Island Glacier
ice shelves. ISSM is available at https://issm.jpl.nasa.gov/.

**Data availability**

The fields used to create any of the figures are available by request. The data used in this work are publicly available. NASA
ITS_LIVE annual velocity mosaics can be found at https://doi:10.5067/6II6VW8LLWJ7. The MEaSUREs 2014-2017 velocity
mosaic (and other multiyear mosaics) are available here: https://doi.org/10.5067/FB851ZIZYX5O. The temperature data from
Comiso,     (2000)     can     be     found     in     the     example     data     sets     for     ISSM     here:
https://issm.jpl.nasa.gov/documentation/tutorials/datasets/. The BedMachine product including the REMA surface elevation
mosaic can be found here: https://doi.org/10.5067/FPSU0V1MWUB6.

**Supplement link**

[PDF attached in submission]



**Author contribution**

BR did conceptualization, methodology, software, visualization, and original draft writing. SN did conceptualization, manuscript review and editing, funding acquisition, and supervision. KP did conceptualization and manuscript review and editing.

**Competing interests**

One of the coauthors is a member of the editorial board of *The Cryosphere*.

**Acknowledgements**

DEMs provided by the Byrd Polar and Climate Research Center and the Polar Geospatial Center under NSF-OPP awards 1043681, 1542736, 1543501, 1559691, 1810976, and 2129685. The scientific color maps used in all plan view plots come from Crameri et al. (2020).



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
