# Peer review of "Comprehensive Assessment of Stress Calculations for Crevasse Depths and Testing with Crevasse Penetration as Damage"

_EGUsphere, 2024_

## Referee Comment (RC1)

**Review of Comprehensive Assessment of Stress Calculations for Crevasse Depths and Testing with Crevasse Penetration as Damage**

The authors make an important point that those in the field should be more careful about their definition of stress. The variety of stresses used in previous literature can lead to pronounced effects on the predicted crevasse depths, and subsequently damage and viscosity, which could potentially modify predictions of ice sheet flow and sea level rise.

That said, the authors miss relevant points in literature, and do not adequately address uncertainties in their modeling. As such, major revisions for this paper are suggested.

As a starting minor point, I will refer to your "Nye's theory" as the "Zero Stress Approximation" throughout this review. First, following Benn et al. 2007's error and subsequently many others, the wrong paper is cited in Line 111 - Nye's back-of-the-envelope fracture depth calculation is equation 2 in his 1955 paper "Comments on Dr. Loewe's Letter and Notes on Crevasses". Second, Nye did not discuss basal crevasses, meltwater, or any other variation that has since been applied to the theory, so the version used in this study is not Nye's conception. Third, the term Zero Stress approximation already exists in the literature - Duddu et al., 2020; Huth et al., 2021; Coffey et al., 2024.

**General Comments**

1. You should not include discussion with LEFM or papers that apply it unless you will do so properly. In its current form, this preprint does not adequately detail LEFM nor its limitations, summarized by the following two points. Overall, since you do not model LEFM, you should remove comparison of previous papers that use LEFM, as well as your description of the theory near the start of the paper. I recommend you focus on comparisons given what you have modeled, which is the Zero Stress Approximation (you call Nye's theory).
    a. Mode I LEFM, as presented in van der Veen, 1998a,b from Tada's Handbook of Stress Intensity Factors, assumes plane strain ($\epsilon_{yy} = 0$, with y the direction into or out of the page). Naturally, as strain rate is

the time derivative of strain, this would make the deviatoric stress $\tau_{yy}=0$. Hence, for Mode I LEFM, you must assume that the flow is 1D, and $R_{xx} = 2 \tau_{xx} + 0$. As such, it's inappropriate to discuss using stress states with 2HD (2 horizontal dimensions) because that goes against the assumptions used to calculate the SIFs (stress intensity factors) in the Tada Handbook.

b. Second, there are other Modes of fracture for LEFM, specifically in-plane and out-of-plane shear, which are referred to as Modes II and III, respectively. You should not discuss Mode I LEFM as the failure mechanism in shear margins, which makes including Mode I LEFM papers confusing in your discussion of over-/under-predicting in shear margins.

2. The Zero Stress Approximation does not uphold horizontal force balance. This has been shown in Buck 2023 and discussed in Coffey et al., 2024. For isothermal ice, incorporating force balance as discussed in those two papers yields deeper crevasses, and reduces the calving stress threshold by a factor of 2. This is a significant omission that would alter the predicted crevasse depths maps and velocity misfits when using damage.

3. There are substantial omissions in addressing uncertainties in your data-model comparison.

   a. From the modeling side, I have the following questions. They all tie to the point of inverse problems allowing for non-unique matches to data.

      i. What is the uncertainty regarding rheology, such as the flow law exponent?

      ii. Is there uncertainty in the dependence of your effective viscosity on temperature?

      iii. What is your uncertainty in the vertical temperature profile?

      iv. Is using a depth-averaged ice hardness equivalent to depth-varying ice hardness when computing fracture depths, or do these give possibly different results as discussed in Coffey et al., 2024?

      v. Is SSA, a long-wavelength or large-scale continuum approximation of the momentum equation, still a good approximation when you have fractures or rifts?

      vi. Why do you use isotropic damage mechanics when Huth et al., 2021 suggest using anisotropic damage?

      vii. For the inversion, what is your cost function?

       viii.   If using damage is worse than inversion (Figure 10a), could this mean that your starting point for rheology with no damage (pink) is unreasonable?

    b.   From the observations side, I have the following questions.

       i.   Isn't the data returned from rifts inappropriate for use in your model because a) mélange may have different material properties, b) you may be computing strain rates from mélange velocity and spreading rather than glacial ice, c) the ice thickness is significantly decreased, greatly decreasing observed thickness H and increasing your crevasse depths e.g. $R_{xx} / \rho_i g H$ to by default predict a full thickness fracture? One attempt to deal with this by Coffey et al., 2024 is masking H with an average of local unbroken ice thickness, but this does not fix the problem with strain rates. For a study explicitly on crevasse depths you should mask out rifts as your prediction of crevasse depths becomes non-causal.

       ii.   How do you compute the derivative of the velocity field to create strain rate maps? For example, the strain rate maps in Wearing et al., 2016 (thesis) vs Furst et al., 2016 are quite different. Wearing discusses the influence of various spatial filters - it would be nice to see maps of strain rate that go into your crevasse depth maps, perhaps in an appendix.

4. Since you discuss cliff failure, your paper is about errors of around a factor of 2 with the Zero Stress approximation, and you are asking authors to be careful about the confusion between resistive stress and deviatoric stress, is there anything more you would like to say about Bassis and Walker, 2011?

5. Please use names for the stress calculations that have physical relevance and meaning. Calculation A-F gives the reader no insight into the differences. Please make this change in the text and especially in the figures.

6. Following Table 2 and Figure 4: Can you compute, for a given ice shelf or idealized rectangular domain ice shelf, maps of the ratio of components of strain rate, e.g. minimum / maximum principal strain rate? This will give your audience a good idea of where these different choices of stress are most at play and would pair very nicely with your Figure 4 if you put them side by side. I think this would greatly strengthen your study.

**Specific Comments and Technical Corrections**

1. There is no need to include surface meltwater in your figure 1 diagram. You do not use it, making it confusing for a fast read of your figures.
2. Lines 12-14: unnecessary 2 sentences. Either put those in the main text with specific citations or leave them out of the abstract. All you need to say in this upper abstract is that stress calculations vary greatly across studies and make cross-study validation challenging.
3. Introduction paragraph 1 is too large. Be more succinct or make 2 paragraphs.
   a. Line 29: be more specific in tying back these fracture processes to grounding line flux, which is a glacial contribution to the rate of sea level rise.
   b. Line 31: define buttressing with citations.
   c. Line 36: I would not equate calving with shelf collapse. "Both can result in" rather than "the result can be the same"
   d. Line 38: New paragraph at Finally, maybe drop that word choice.
   e. Lines 38 to the end of the paragraph: reads as summarizing some previous work with no clear story arc, ending in surface energy balance which is never again mentioned in the paper. Decide if there is a message here or move this to when you discuss individual studies.
4. Line 54: I would add Horizontal Force Balance (see main point 2).
5. Line 56: LEFM "can recognize" ice strength, but it does not have to as you can choose zero fracture toughness e.g. for Mode I, $K_{Ic} = 0$.
6. Paragraph of Line 75: please end with your main result at the end of the introduction so the audience knows where it is going, not just the broad methodology.
7. Line 81: ice deformation is set from the full stress tensor regardless of rheology. Also, if you're talking about ice shelves or SSA to start with, please begin with that instead so readers can follow your logic.
8. Line 82: provide a citation for ice not being able to flow in triaxial tension.
9. Lines 82-3: Lithostatic pressure is essential to all glacier deformation. Lithostatic pressure is what creates the driving stress (e.g. $\rho_i g \partial_x s$) in SSA and is what drives vertical shear ice flow in SIA. Take away gravity as a body force and nothing drives glacier flow. It is often referred to as a viscous gravity current.

10. Lines 89-90: Near an ice cliff (or ice front) there will likely be vertical shear effects. Not so simple.

11. Lines 92-4: Consider citing relevant literature: Gao et al., 2023 (firn), Coffey et al., 2024 (temperature), Meng et al., 2024 (poroelasticity).

12. Lines 99: meltwater in a surface crevasse. A pool of meltwater, or a small lake, will add a vertical force downwards on the ice surface (e.g. MacAyeal et al., 2015)

13. Lines 103-7 sentence: can be much more succinct. This can be visualized in the supplement of Buck and Lai 2021, or the Appendices of Coffey et al., 2024. Also, be more specific - compressive stress vs lithostatic stress? I recommend lithostatic, unless you are talking about sources of buttressing providing compression.

14. Equation 4: Is this resistive stress only along-flow or crevasse-normal? Otherwise, you should include the second invariant of strain rate in your calculation (see Appendix A of Coffey et al., 2024). I realize you discuss more later on about calculating resistive stress, but make a point of what that equation in Nick et al., 2010 is missing early on and what you want to change about it.

15. In case you strongly disagree and want to keep the LEFM portions of this paper,
    a. Line 135: Say what boundary conditions are unphysical and what applications (ice shelves) must be changed.
    b. Line 143: The resistive stress is not the same because of the plane strain assumption of the Mode I LEFM result in Tada.

16. Section 2.4.1: In general, please define the whole Cauchy stress tensor, resistive stress tensor, the relation between deviatoric stress and strain rate, etc. Lead by example in being thorough with your stress definitions.

17. Line 171: provide a citation, or argue that individuals in the field have done this (cite them) and state your opinion on the matter. If you write out the full expression from mass conservation with variable density, what is the relation between strain rates and density?

18. Line 178: Move your chosen approximate momentum equation (SSA) up earlier when defining your stresses.

19. Equation 10: Move this up, and use another equality to show that the product of the first two terms is what you are calling viscosity.

20. Lines 185-8: Choi et al., 2018 and Lai et al., 2020. You don't need a new sentence about the Lai et al., 2020 application.

21. Lines 195: can you provide a citation for where you get the jargon planar stress tensor?
22. Line 203: cite SSA with neglecting vertical shear stresses.
23. Equations 18 and 19: Do you mean at the crack tips? Otherwise \sigma_{zz} should be a function of z.
24. Line 215: You will not get the full stress as a function of depth unless you use \sigma_{zz} (z). This is clear from the (z) component of Stokes flow, removing vertical shear stress terms.
25. Lines 230, 526: Add Bassis and Walker to this list.
26. Line 253: shelf (typo).
27. Section 3.1: Make these plots! It would be so useful! Even if they go in the appendix, they are the basis for how you understand the bizarre geometries of real ice shelves. I know you have Table 3, but following main point 6, it would be helpful.
28. Lines 267-8: Since the ocean is saltwater, the freezing point is roughly -2 C. Why do you use 0 instead of -2?
29. Line 271: Write the rigidity function and say what you have used to interpolate temperature between the surface and the bed. This significantly affects crevasse height, see Lai et al., 2020 and Coffey et al., 2024.
30. Line 277: The theory you chose suggests that surface crevasses alone don't really matter for making a large damage variable and don't drive calving without water. I would be more forthcoming about why ice shelves are a natural environment to study crevasses (removing basal drag), and that basal crevasses are likely the driver of calving, as they have received far less attention in terms of number of papers.
31. Line 279: Isn't the Larsen B remnant multi-year landfast sea ice (Ochowat et al., 2023) instead of glacial ice? Would it have different fracture properties? It also collapsed from surface meltwater in 2022 (Ochowat et al., 2023) - wouldn't this affect your modeling if the surface crevasses had meltwater in them, or if there were surface meltwater ponds again?
32. Paragraph starting with line 285: good logic! Well written.
33. Paragraph 293: It is unclear what exactly you are doing with temperature. In 293, you say it is constant with depth. In 301, it is quadratic, with a 5-degree shift at 1/3rd of the ice thickness discussed in Line S19. There's a lot of discussion about tuning and it's very unclear what the overall effect is - you warm bias

temperature, and you tune temperature to match velocity with calculation F (shouldn't your damage calculations with calculation E by definition be worse?). Place some of this in S2 if you feel it is detailed.

34. Line 321: for isothermal ice and the Zero Stress theory, you should be able to predict just how much larger the basal crevasses are than the surface crevasses by computing the ratio of basal to surface crevasses. I would recommend doing so.

35. Table 3: Please make some of this nondimensional. I don't know how to contextualize these other than relative to each other but not a fraction of the ice thickness.

36. Figure 4: non-dimensionalize y-axis, change labels to be physically relevant.

37. Lines 358 & 390, Figures 5 & 9: as discussed in the main points 3.2.1, every theory will predict rifts if they exist in the data because of the reduced thickness and velocity anomaly. These should either not be included in your analyses or you should treat them carefully.

38. Section 4.3: Is it valid to use the Zero Stress approximation for shear cracks in addition to tensile cracks?

39. Line 442: "tuned … across the domain" for calculation F? This is unclear.

40. Line 447, Figure 10: You should be clear about the point of the inversions. The way it is presented, if I want to match observations, I should just use the inversions, no need for calculation E or F. But I doubt that's what you want to say - presumably, it is that you can't do better than the inversion, and the reader should measure your calculations (E & F) velocity misfit relative to no damage. You should state this more explicitly as it is unclear during a first fast read how to interpret your results.

41. Lines 516-18: You can remove this example and just cite them as being unclear.

42. Lines 522-3: With flow-direction versus maximum principal stress direction, did this alter the conclusions of this study? What is the order of magnitude of this distinction?

43. Line 545: see main point 1.

44. Can you add Wilner et al., 2023 to your table 4?

45. Section 5.6: see main point 1.1 and 1.2.

46. Line 590: the variation by a factor of 2 - is this more or less than confusing the deviatoric and resistive stress? Isn't this something that should be clear from the start of the study?

47. Line 590: The regions of difference between stress calculations on ice shelves is the major new finding and citing a figure to go along and show those differences would be very helpful in your figure 4 - see main point 6.
48. Lines 604-5: re physical basis for the Zero Stress approximation, see main point 2.
49. Lines 607-8: cite someone or provide a supplemental figure for these points about convexity of the ice front.
50. Table A2: might be useful to have equations in this table as well. Specifically for damage, defining resistive stress, etc.

**References**

Benn et al., 2007

Buck, 2023

Buck and Lai, 2021

Coffey et al., 2024

Duddu et al., 2020

Furst et al., 2016

Gao et al., 2023

Huth et al., 2021

MacAyeal et al., 2015

Meng et al., 2024

Nick et al., 2010

Nye, 1955

Ochowat et al., 2022

Tada et al., 1973

Van der Veen, 1998a

Van der Veen, 1998b

Wearing et al., 2016

Wilner et al., 2023

---

## Author Comment (AC1)

**Review of Comprehensive Assessment of Stress Calculations for Crevasse Depths and Testing with Crevasse Penetration as Damage**

The authors make an important point that those in the field should be more careful about their definition of stress. The variety of stresses used in previous literature can lead to pronounced effects on the predicted crevasse depths, and subsequently damage and viscosity, which could potentially modify predictions of ice sheet flow and sea level rise.

That said, the authors miss relevant points in literature, and do not adequately address uncertainties in their modeling. As such, major revisions for this paper are suggested. As a starting minor point, I will refer to your "Nye's theory" as the "Zero Stress Approximation" throughout this review. First, following Benn et al. 2007's error and subsequently many others, the wrong paper is cited in Line 111 - Nye's back-of-the-envelope fracture depth calculation is equation 2 in his 1955 paper "Comments on Dr. Loewe's Letter and Notes on Crevasses". Second, Nye did not discuss basal crevasses, meltwater, or any other variation that has since been applied to the theory, so the version used in this study is not Nye's conception. Third, the term Zero Stress approximation already exists in the literature - Duddu et al., 2020; Huth et al., 2021; Coffey et al., 2024.

We appreciate the correction regarding Nye's 1955 and 1957 papers and have corrected this in the text. We would change to "Zero Stress Approximation" throughout the manuscript as recommended to be consistent with recent literature. We are encouraged that the reviewer concurs with the importance of showing how stress calculations in recent literature lead to different crevasse depths which impact parametrizations of damage and viscosity used in ice sheet modeling. We answer the concerns about additional relevant literature and modeling uncertainty inline below.

**General Comments**

1. You should not include discussion with LEFM or papers that apply it unless you will do so properly. In its current form, this preprint does not adequately detail LEFM nor its limitations, summarized by the following two points. Overall, since you do not model LEFM, you should remove comparison of previous papers that use LEFM, as well as your description of the theory near the start of the paper. I recommend you focus on comparisons given what you have modeled, which is the Zero Stress Approximation (you call Nye's theory).
   a. Mode I LEFM, as presented in van der Veen, 1998a,b from Tada's Handbook of Stress Intensity Factors, assumes plane strain ($\epsilon_{yy} = 0$, with y the direction into or out of the page). Naturally, as strain rate is the time derivative of strain, this would make the deviatoric stress $\tau_{yy}=0$. Hence, for Mode I LEFM, you must assume that the flow is 1D, and $R_{xx} = 2 \tau_{xx} + 0$. As such, it's inappropriate to discuss using stress states with 2HD (2 horizontal dimensions) because that goes against the assumptons used to calculate the SIFs (stress intensity factors) in the Tada Handbook.

b. Second, there are other Modes of fracture for LEFM, specifically in-plane and out-of-plane shear, which are referred to as Modes II and III, respectively. You should not discuss Mode I LEFM as the failure mechanism in shear margins, which makes including Mode I LEFM papers confusing in your discussion of over-/under-predicting in shear margins.

We appreciate the constructive feedback regarding our consideration of LEFM. We believe that that our findings are relevant to the application of LEFM to crevasses based on our responses below to the two individual points. In a revised manuscript, we would provide a more thorough introduction to LEFM including mode I,II,III and mixed mode fracture and studies assessing crevasse orientation.

a. We appreciate the raising of this complexity and agree that it is an important broken assumption when comparing stress intensity factors to fracture toughness from plane strain tests. That said, we question whether, if LEFM is to be applied to all regions across ice shelves as has been done in Lai et al., 2020 and is likely to be done by future studies, applying a crack perpendicular stress that is known to be inaccurate is the best course of action. Put another way, when applying LEFM where tau_yy~=0, neglecting the unknown effect of violating this assumption is unavoidable. Applying a stress perpendicular to the crevasse (s_xx), which is the direction that seems reasonable to assume as most important, that is not the actual stress seems like an additional but more avoidable error. A minor additional point about this: our understanding is that in 2D LEFM derivations with plain strain, it is plain strain in an elastic material such that a stress parallel to the crevasse tip will occur (s_zz in fracture literature, s_yy in crevasse literature) (Anderson, 2005 section 2.10). The point remains that this assumption is violated because any value could occur when applying LEFM across glaciers and ice shelves versus the specific value controlled by Poisson's ratio.

b. Our understanding of Mode I, II, and III crack orientations is that they are based on the loading direction rather than the deformation state. Mode I is defined as "where the principal load is applied normal to the crack plane" (Anderson, 2005 section 2.6). Given the observation of crevasses (admittedly surface crevasses) in shear margins aligning approximately 45 degrees from flow (Colgan et al., 2016; Van Wyk de Vries et al., 2023), which aligns with the maximum principal stress direction, we would consider these to be mode I dominated cracks. Mode III would be a crevasse aligned parallel to flow in a shear margin. There is certainly additional complexity (mixed mode) as crevasses in shear margins reorient with flow, which is less the case for mode I crevasses in the center of flow. But for crevasse initialization in particular, we would argue mode I is an appropriate starting place.

With these responses, we propose to maintain discussion of LEFM and LEFM-based studies in the manuscript. We would be clear about the

violation of the assumption noted (a) and that there is additional complexity in shear margins as discussed in (b) particularly as crevasses evolve with flow. We would also be clear that our recommendation of Calculation F is more caveated in LEFM accordingly. At the same time, the differences in stress calculation feeding into LEFM will still give differing results with the approximate pattern shown with the zero stress approximation and past studies employing LEFM have selected different resistive stress versions, so we think that our results are worth reviewing for future LEFM based studies even if we did not make a recommendation.

2. The Zero Stress Approximation does not uphold horizontal force balance. This has been shown in Buck 2023 and discussed in Coffey et al., 2024. For isothermal ice, incorporating force balance as discussed in those two papers yields deeper crevasses, and reduces the calving stress threshold by a factor of 2. This is a significant omission that would alter the predicted crevasse depths maps and velocity misfits when using damage.

We appreciate the raising of this important point regarding the zero stress approximation and Buck's horizontal force balance model. Reviewing in particular Figure 6 of Buck (2023), we make the following notes:

1. For the surface crevasse depth / basal crevasse height predictions, similar to LEFM, this model would increase the difference in basal crevasse that predicted from each stress calculation, because of the non-linear relationship between stress and crevasse size.
2. For the modeling component of the paper, inclusion of Buck's horizontal force balance model would likely have an impact like going from calc. F to calc. E for the Larsen B remnant. The deviation from the zero stress approximation grows with decreasing buttressing. As buttressing numbers tend to be lowest at the front, this would cause a relative weakening of the ice near the front relative to elsewhere. This is similar to the relative weakening toward the front from calc. E from omitting ice softening from the vertical strain rate.

With this, we propose to leave the crevasse penetration plots (figures 5 and 9) in terms of the zero stress approximation. This is in part because it is not clear how the force balance model should be applied when longitudinal flow cannot be assumed (shear margins). We would note the qualitative impact on table 3 and figure 4 as we did for LEFM. For at least the Larsen B, we would generate a crevasse penetration map with calc. F with the force balance model applied, and show the difference to calc. F with only the zero stress approximation. We would also test velocity misfit from damage with calc. F and force balance and compare to the original calc. F and calc. E results. All new results discussed above would be included in the supplement and noted in the discussion section of the manuscript.

3. There are substantial omissions in addressing uncertainties in your data-model comparison.
   a. From the modeling side, I have the following questions. They all tie to the point of inverse problems allowing for non-unique matches to data.
      i. What is the uncertainty regarding rheology, such as the flow law exponent?
      ii. Is there uncertainty in the dependence of your effective viscosity on temperature?
      iii. What is your uncertainty in the vertical temperature profile?
      iv. Is using a depth-averaged ice hardness equivalent to depth-varying ice hardness when computing fracture depths, or do these give possibly different results as discussed in Coffey et al., 2024?
      v. Is SSA, a long-wavelength or large-scale continuum approximation of the momentum equation, still a good approximation when you have fractures or rifts?
      vi. Why do you use isotropic damage mechanics when Huth et al., 2021 suggest using anisotropic damage?
      vii. For the inversion, what is your cost function?
      viii. If using damage is worse than inversion (Figure 10a), could this mean that your starting point for rheology with no damage (pink) is unreasonable?

   Each of these uncertainties would be briefly discussed in the methods section consistent with the statements below. The purpose of modeling in this paper was primarily to assess whether calc. E or F yields a relative pattern of damage that is more consistent with observed velocity, as opposed to providing an estimate of crevasse sizes with fully assessed error bounds. We respond to these points individually below noting where we would do additional work:

   i. As reviewed in Cuffey & Patterson (2010), rheology is anisotropic and path dependent as crystallographic preferred orientation and grain size change with deformation. Values of n from 2 to 5 have been found with 3 being used most frequently per their recommendation. Millstein et al. (2022) recently recommended n=4 as a better "average" representation, and we would assess n=4 for the Larsen B remnant (Scar inlet) and PIG with the same modeling workflow as supplementary material.
   ii. This uncertainty follows the above uncertainty, as microscale processes are temperature dependent and the function for flow factor will have error accordingly.
   iii. We briefly discuss in the supplement the dearth of constraining observations for through thickness temperatures in shelves. We would note this in the methods section.

iv.   No. We would reference (Coffey et al., 2024) in discussing the difference in predictions caused.

v.   No continuum stress balance equations including full stokes will be a good representation of fractures and rifts.

vi.   This is a limitation of the ice sheet model selected, ISSM, and discussion of this limitation has been added.

vii.   We use the velocity error alone (no regularization) because damage from crevasses may be expected to have sharp gradients. Checking of whether this prevented the algorithm finding the best solution versus using a small regularization coefficient was performed.

viii.   We would not expect damage as forward calculated from crevasse penetration to be better than inversion, because inversion minimizes error with no physical constraints and can thus account for local unknown temperature profile, imperfect rheology, and crevasse effects. Our methodology aimed primarily at assessing Calc. E vs Calc. F only addresses temperature in bulk and crevasses (with its own error). That said, we were surprised out how close a first pass at making a damage field with crevasse penetration and bulk temperature tuning got to matching inversions and think that future work including rheology modifications and crevasses could help understand how much of each is being accounted for by inverted damage.

b.   From the observations side, I have the following questions.

i.   Isn't the data returned from rifts inappropriate for use in your model because a) mélange may have different material properties, b) you may be computing strain rates from mélange velocity and spreading rather than glacial ice, c) the ice thickness is significantly decreased, greatly decreasing observed thickness H and increasing your crevasse depths e.g. $R_{xx} / \rho_i g H$ to by default predict a full thickness fracture? One attempt to deal with this by Coffey et al., 2024 is masking H with an average of local unbroken ice thickness, but this does not fix the problem with strain rates. For a study explicitly on crevasse depths you should mask out rifts as your prediction of crevasse depths becomes non-causal.

ii.   How do you compute the derivative of the velocity field to create strain rate maps? For example, the strain rate maps in Wearing et al., 2016 (thesis) vs Furst et al., 2016 are quite different. Wearing discusses the influence of various spatial filters - it would be nice to see maps of strain rate that go into your crevasse depth maps, perhaps in an appendix.

Responding to each point:

i. We agree on all subpoints and would mask the rifts from figures 5, 6, and 9 (based on a thickness/strain rate criterion) to avoid readers taking information from these zones where the result is self-fulfilling.

ii. We use second order accurate central differences (single-sided at boundaries) and no filtering. Strain rate maps for the Larsen B and PIG shelves would be added to the appendix as you suggest.

4. Since you discuss cliff failure, your paper is about errors of around a factor of 2 with the Zero Stress approximation, and you are asking authors to be careful about the confusion between resistive stress and deviatoric stress, is there anything more you would like to say about Bassis and Walker, 2011?

Reviewing Bassis & Walker (2011), $S_{xx}$ is termed horizontal deviatoric stress when it is instead the horizontal resistive stress. Their Equation 2.2 is consistent with the definition of resistive stress and the resulting equation 2.4 appears to match the depth-averaged boundary condition for a marine ice cliff in terms of resistive stress.

5. Please use names for the stress calculations that have physical relevance and meaning. Calculation A-F gives the reader no insight into the differences. Please make this change in the text and especially in the figures.

We propose the following naming system to add physical relevance while maintaining brevity: E[X]-S[Y]-[Z] where [X] is assigned to 0 (no eff strain rate), P (planar eff strain rate), or F (full effective strain rate); [Y] is assigned to F (flow direction) or M (max prin direction); and [Z] is assigned to 0 (lateral deviatoric stress not used) or 1 (lateral deviatoric stress used).

This would give the following mapping from the old to the new system:

Calc. A = E0-SF-0

Calc. B = E0-SM-0

Calc. C = EP-SM-0

Calc. D = EF-SM-0

Calc. E = EP-SM-1

Calc. F = EF-SM-1

6. Following Table 2 and Figure 4: Can you compute, for a given ice shelf or idealized rectangular domain ice shelf, maps of the ratio of components of strain rate, e.g. minimum / maximum principal strain rate? This will give your audience a good idea of where these different choices of stress are most at play and would pair very nicely

with your Figure 4 if you put them side by side. I think this would greatly strengthen your study.

We appreciate this suggestion and agree it would go a long way towards connecting figure 4 to the later crevasse penetration and crevasse penetration difference plots. We will make the suggested plots for the Larsen B remnant (or Scar inlet) and PIG shelves and include them with figure 4 or as an immediately subsequent figure. In conjunction with the recommendation from reviewer #2 to make code available for reproducing Figure 4, we will also plan to add code to make these plots to the Jupyter notebook.

**Specific Comments and Technical Corrections**

1. There is no need to include surface meltwater in your figure 1 diagram. You do not use it, making it confusing for a fast read of your figures.
   We will remove meltwater from both panels of figure 1.

2. Lines 12-14: unnecessary 2 sentences. Either put those in the main text with specific citations or leave them out of the abstract. All you need to say in this upper abstract is that stress calculations vary greatly across studies and make cross-study validation challenging.
   Sentences removed.

3. Introduction paragraph 1 is too large. Be more succinct or make 2 paragraphs.
   a. Line 29: be more specific in tying back these fracture processes to grounding line flux, which is a glacial contribution to the rate of sea level rise.
      Added.
   b. Line 31: define buttressing with citations.
      We have added the following definition with a sentence before this one.
      NEW: Ice shelves restrain upstream ice flow via buttressing, backstress from shear load transmitted to embayment walls or from compressive load caused by pinning points (Fürst et al., 2016; Gudmundsson, 2013; Schoof, 2007).
   c. Line 36: I would not equate calving with shelf collapse. "Both can result in" rather than "the result can be the same"
      Change implemented verbatim.
   d. Line 38: New paragraph at Finally, maybe drop that word choice.
      We would add a split at line 40 between processes that drive uncertainty in sea-level rise projections and how crevasse depths are used in parametrizations of those processes.

e. Lines 38 to the end of the paragraph: reads as summarizing some previous work with no clear story arc, ending in surface energy balance which is never again mentioned in the paper. Decide if there is a message here or move this to when you discuss individual studies.

We would drop energy balance and end with a stronger statement on importance of crevasse depths for these parametrizations of important processes. We would still cite Colgan et al. (2016) elsewhere as it is the most recent crevasse review paper and summarizes some of the above.

4. Line 54: I would add Horizontal Force Balance (see main point 2).

We have modified and added the start of that paragraph

OLD: There are two primary methods for calculating crevasse depths from stress. The Nye crevasse formulation (Nye, 1957) assumes ice has no tensile strength and that the presence of a crevasse does not modify the surrounding stress field. Linear elastic fracture mechanics (LEFM) …

NEW: There are three primary methods for calculating crevasse depths from stress. The Zero Stress Approximation (Nye, 1955) assumes ice has no tensile strength and that the presence of a crevasse does not modify the surrounding stress field. The Horizontal Force Balance method (Buck, 2023) maintains the assumption that ice has no tensile strength but considers the impact of water pressure in basal crevasses on force balance. As basal crevasse height increases with stress according to the Zero Stress Approximation, so too does the force balance impact creating a crevasse size amplifying effect. Linear elastic fracture mechanics (LEFM) …

5. Line 56: LEFM "can recognize" ice strength, but it does not have to as you can choose zero fracture toughness e.g. for Mode I, $K_{Ic} = 0$.

In an attempt to give a better one sentence opener for LEFM (because another fair point is that a "100KPa stress criterion" would also be recognizing ice strength, we make the following change.

OLD: Linear elastic fracture mechanics (LEFM), which was applied to crevasses by van der Veen (1998a, 1998b), recognizes ice strength and considers the stress-amplifying effect of crevasse geometry.

NEW: Linear elastic fracture mechanics (LEFM), which was applied to crevasses by Rist et al. (1996) and van der Veen (1998a, 1998b), considers the stress-amplifying effect of crevasse geometry and allows laboratory measurements of a material's resistance to fracture to be used for predicting fracture in more complex stress states.

6. Paragraph of Line 75: please end with your main result at the end of the introduction so the audience knows where it is going, not just the broad methodology.

We have added the following final sentence:

NEW: "We find that common assumptions made when calculating resistive stress from strain rates can lead to differing crevasse depths by a factor of two and that the most physically based calculation applied in an ice sheet model as damage best recreates observed velocity."

7. Line 81: ice deformation is set from the full stress tensor regardless of rheology. Also, if you're talking about ice shelves or SSA to start with, please begin with that instead so readers can follow your logic.

We are seeking to provide a distinction between the use of the deviatoric stress and the full or Cauchy stress.

OLD: While the viscous flow of ice is driven by stress differences (deviatoric stresses), brittle failure comes from the full stress.

NEW: While the viscous flow of ice is driven by deviatoric stress, the component of the Cauchy stress that does not cause volume change during deformation, brittle failure is driven by the Cauchy stress itself.

8. Line 82: provide a citation for ice not being able to flow in triaxial tension.

Following a helpful point from review #2 we will clarify this as "equi-triaxial" tension. If sigma_1=sigma_2=sigma_3, all deviatoric stress terms are zero. We wanted to point out that an incoherency that would arise from using deviatoric stress for brittle failure is that ice would not be predicted to fail for that scenario.

9. Lines 82-3: Lithostatic pressure is essential to all glacier deformation. Lithostatic pressure is what creates the driving stress (e.g. \rho_i g \partial_x s) in SSA and is what drives vertical shear ice flow in SIA. Take away gravity as a body force and nothing drives glacier flow. It is often referred to as a viscous gravity current.

Thank you for noting that our wording was confusing, we will clarify the text to include your comment.

10. Lines 89-90: Near an ice cliff (or ice front) there will likely be vertical shear effects. Not so simple.

Consistent with a similar comment from reviewer #2, we have removed this sentence and modified the following one for continuity.

11. Lines 92-4: Consider citing relevant literature: Gao et al., 2023 (firn), Coffey et al., 2024 (temperature), Meng et al., 2024 (poroelasticity).

We will do so.

12. Lines 99: meltwater in a surface crevasse. A pool of meltwater, or a small lake, will add a vertical force downwards on the ice surface (e.g. MacAyeal et al., 2015)

Correction taken verbatim.

13. Lines 103-7 sentence: can be much more succinct. This can be visualized in the supplement of Buck and Lai 2021, or the Appendices of Coffey et al., 2024. Also, be more specific - compressive stress vs lithostatic stress? I recommend lithostatic, unless you are talking about sources of buttressing providing compression.
We will make the sentence more succinct and use "lithostatic" as suggested.

14. Equation 4: Is this resistive stress only along-flow or crevasse-normal? Otherwise, you should include the second invariant of strain rate in your calculation (see Appendix A of Coffey et al., 2024). I realize you discuss more later on about calculating resistive stress, but make a point of what that equation in Nick et al., 2010 is missing early on and what you want to change about it.
We have noted that the resistive stress here is the 1D form

OLD: The resistive stress, $R_{xx}$, in Nick et al. (2010) is given as...

NEW: The resistive stress, $R_{xx}$, in Nick et al. (2010) is the one-dimensional form and is given as...

We will foreshadow by adding the following to the end of the paragraph:

NEW: Assessing three-dimensional implementations of equation 4 that consider effective strain rate (ice softening from multiple directions of deformation) and lateral stress is the primary focus of our work.

15. In case you strongly disagree and want to keep the LEFM portions of this paper,
    a. Line 135: Say what boundary conditions are unphysical and what applications (ice shelves) must be changed.
    b. Line 143: The resistive stress is not the same because of the plane strain assumption of the Mode I LEFM result in Tada.
    We will address these consistent with our discussion of LEFM following main point 1 and will revise L135 and L143 to include your suggestions.

16. Section 2.4.1: In general, please define the whole Cauchy stress tensor, resistive stress tensor, the relation between deviatoric stress and strain rate, etc. Lead by example in being thorough with your stress definitions.
We will do so with an additional section prior to what is presently 2.4.1. This section will cover the Cauchy stress tensor, deviatoric stress tensor, and resistive stress tensor. It will also introduce why this calculation path is necessary, namely we can measure strain rate directly in the field and from velocity products which gives deviatoric stresses through the flow law. And then full stress versus thickness is recovered by calculating resistive stress from deviatoric stress and adding lithostatic pressure (and sometimes water pressure) back in.

17. Line 171: provide a citation, or argue that individuals in the field have done this (cite them) and state your opinion on the matter. If you write out the full expression from mass conservation with variable density, what is the relation between strain rates and density?

    We will cite at least Amaral et al. (2020) here, who neglected vertical strain rate. We will note that neglected vertical strain rate (assuming it to be zero) will cause a density decrease and write the lagrangian form of continuity (to avoid confusion with advected density change) in terms of strain rates d rho / d t = - rho(e1 + e2 + e2). For crevasse formation in the absence of firn, we will recommend upholding incompressibility as likely the most physically consistent.

18. Line 178: Move your chosen approximate momentum equation (SSA) up earlier when defining your stresses.

    We will discuss the SSA assumption and its necessity when working from surface velocity in the new section 2.4.1 where we introduce the stress tensor.

19. Equation 10: Move this up, and use another equality to show that the product of the first two terms is what you are calling viscosity.

    We will move it to after equation six and define effective viscosity immediately after.

20. Lines 185-8: Choi et al., 2018 and Lai et al., 2020. You don't need a new sentence about the Lai et al., 2020 application.

    Sentence about application removed.

21. Lines 195: can you provide a citation for where you get the jargon planar stress tensor?

    We are trying to balance consistency with glaciological literature and accuracy with continuum mechanics definitions. Glaciological literature (e.g. van der Veen, 1999) will define tau_1 and tau_2 from the upper-left 2x2 matrix (x, y) as the maximum and minimum principal stresses. Recognizing the 3d state, tau_zz could end up being the maximum or minimum principal stress in some cases. Maximum / minimum principal stress from the planar tensor was our idea for efficiently balancing these targets. Instead, we will just explain this difference between notation in glaciological vs continuum mechanics literature and then use tau_1 and tau_2 rather than introducing terminology that could cause confusion with the formal meaning of plane strain.

22. Line 203: cite SSA with neglecting vertical shear stresses.

    We have noted SSA as the reason for neglecting bridging stress with citation (MacAyeal, 1989).

23. Equations 18 and 19: Do you mean at the crack tips? Otherwise \sigma_{zz} should be a function of z.

    Added "at the crack tip".

24. Line 215: You will not get the full stress as a function of depth unless you use \sigma_{zz} (z). This is clear from the (z) component of Stokes flow, removing vertical shear stress terms.
We rewrote this to clarify we are solving for the surface crevasse depth and basal crevasse height where the full stress is zero.
OLD: Equations 18 and 19 can be substituted into Equation 17 to find the full stress as a function of depth ($\sigma_n$ (z) in Equation 5) for surface and basal crevasses.
NEW: Equations 18 and 19 can be substituted into Equation 17 to find the full stress at the crack tip as a function of surface crevasse depth and basal crevasse height.

25. Lines 230, 526: Add Bassis and Walker to this list.
We originally chose not to include Bassis and Walker because they use the depth-averaged boundary condition rather than calculations from strain rates. As noted in main point 4, we do not think they used deviatoric stress rather than the resistive stress.

26. Line 253: shelf (typo).
Fixed! Thank you.

27. Section 3.1: Make these plots! It would be so useful! Even if they go in the appendix, they are the basis for how you understand the bizarre geometries of real ice shelves. I know you have Table 3, but following main point 6, it would be helpful.
Thank you for this suggestion, we plan to make these plots and add them to the main text as noted below main point 6.

28. Lines 267-8: Since the ocean is saltwater, the freezing point is roughly -2 C. Why do you use 0 instead of -2?
This was a carryover from old analysis; we would update to -2 C.

29. Line 271: Write the rigidity function and say what you have used to interpolate temperature between the surface and the bed. This significantly affects crevasse height, see Lai et al., 2020 and Coffey et al., 2024.
We will add clarification here (constant temperature is assumed), define rigidity (B = A^(-1/n)) following equation 6. We also plan to add a more thorough description of temperature assumptions in the supplement, as we note in response to a comment further down.

30. Line 277: The theory you chose suggests that surface crevasses alone don't really matter for making a large damage variable and don't drive calving without water. I would be more forthcoming about why ice shelves are a natural environment to study crevasses (removing basal drag), and that basal crevasses are likely the driver of calving, as they have received far less attention in terms of number of papers.
We will foreshadow why shelves are selected at the start of 3.2.1 as this would not yet be obvious; we could have shown the surface crevasse penetration patterns on

grounded ice for example. We will add emphasis to why shelves are studied and that basal crevasses drive in 3.2.2 and 3.3.

31. Line 279: Isn't the Larsen B remnant multi-year landfast sea ice (Ochowat et al., 2023) instead of glacial ice? Would it have different fracture properties? It also collapsed from surface meltwater in 2022 (Ochowat et al., 2023) - wouldn't this affect your modeling if the surface crevasses had meltwater in them, or if there were surface meltwater ponds again?

We are working with the southern end of the Larsen B that never collapsed. This remaining continuous shelf has been called both the Larsen B remnant (Borstad et al., 2016) and the Scar Inlet (Ochwat et al., 2024) . In hindsight, Scar inlet is less likely to cause confusion with the landfast ice in the embayment, and we would switch to Scar inlet as the primary title throughout the manuscript.

32. Paragraph starting with line 285: good logic! Well written.

Thank you.

33. Paragraph 293: It is unclear what exactly you are doing with temperature. In 293, you say it is constant with depth. In 301, it is quadratic, with a 5-degree shin at 1/3rd of the ice thickness discussed in Line S19. There's a lot of discussion about tuning and it's very unclear what the overall effect is - you warm bias temperature, and you tune temperature to match velocity with calculation F (shouldn't your damage calculations with calculation E by definition be worse?). Place some of this in S2 if you feel it is detailed.

We will provide a short qualitative description in the main text and move a detailed explanation of how temperature is treated for the crevasse penetration calculations, for tuning with a given damage field from calculatoins E / F, for inversions, and for the no damage case.

34. Line 321: for isothermal ice and the Zero Stress theory, you should be able to predict just how much larger the basal crevasses are than the surface crevasses by computing the ratio of basal to surface crevasses. I would recommend doing so.

We will note the theoretical ratio.

35. Table 3: Please make some of this nondimensional. I don't know how to contextualize these other than relative to each other but not a fraction of the ice thickness.

We agree fraction of ice thickness does not make sense as zero stress approximation crevasse depths / heights are independent of ice thickness for a given resistive stress and that producing arbitrary crevasse sizes is unfortunate. The only option we see is normalizing against calc. B. Normalizing against a calculation that is not the most "true" seems misleading. Normalizing against calc. F however would recast the plot and obscure direction of crevasse size change across flow

states. For this reason, we would prefer to maintain as is with a reasonable value of strain rate so that predicted sizes are not unrealistic.

36. Figure 4: non-dimensionalize y-axis, change labels to be physically relevant.

Same as above. We would add additional labels to the x-axis (-1: pure shear, 0: longitudinal spreading, 1: equi-biaxial spreading).

37. Lines 358 & 390, Figures 5 & 9: as discussed in the main points 3.2.1, every theory will predict rifts if they exist in the data because of the reduced thickness and velocity anomaly. These should either not be included in your analyses or you should treat them carefully.

As described under main point 3.2.1, we will mask out the rifts to better avoid drawing attention to non-meaningful results. The regions corresponding to lines 358 and 390 are shear margins not the rifts adjacent to them. We will clarify this point in the text. The northern shear margin is partially failed like PIG 2018, and we will double check that the cross section is not going through the more fracture regions.

38. Section 4.3: Is it valid to use the Zero Stress approximation for shear cracks in addition to tensile cracks?

Following onto our comments regarding the application of LEFM in shear areas, we would argue that cracks aligned ~45 degrees from the flow direction are tensile cracks in areas of shear deformation. Additional complexity would again apply as deformation reorients crevasses, but as a first pass we think zero stress is reasonable.

39. Line 442: "tuned … across the domain" for calculation F? This is unclear.

A single depth-averaged temperature is assumed across the domain and is tuned for minimum velocity misfit with the pattern of damage applied on top. We will improve clarity in line 442.

40. Line 447, Figure 10: You should be clear about the point of the inversions. The way it is presented, if I want to match observations, I should just use the inversions, no need for calculation E or F. But I doubt that's what you want to say - presumably, it is that you can't do better than the inversion, and the reader should measure your calculations (E & F) velocity misfit relative to no damage. You should state this more explicitly as it is unclear during a first fast read how to interpret your results.

The takeaways we want are:

1. Calculation F outperforms calculation E giving some evidence that calculation F has a stronger connection to real basal crevasse height patterns
2. Like you point out: Given the bookends of no damage and the best possible damage pattern for matching velocity (inversion), damage from crevasses far outperforms no damage and gets fairly close to inversions.

We would reorder the third paragraph in 4.4 to better cover these two points in this order / priority.

41. Lines 516-18: You can remove this example and just cite them as being unclear.

We will do so.

42. Lines 522-3: With flow-direction versus maximum principal stress direction, did this alter the conclusions of this study? What is the order of magnitude of this distinction?

We did not discuss the flow direction calculation after noting the obvious, no crevasses in shear margins, in the results section. We would reiterate that finding here.

43. Line 545: see main point 1.

We will reiterate the broken LEFM assumption here.

44. Can you add Wilner et al., 2023 to your table 4?

Yes, however, because Wilner uses the divergence of velocity rather than resistive stress in 3D or its 2D version, it will get its own row.

45. Section 5.6: see main point 1.1 and 1.2.

Consistent with our discussion after main points 1.1 and 1.2, we would raise the violated assumption regarding lateral stress as well as potential for mixed mode in shear margins. The differences pointed out between calculations will still apply. We would soften our suggestion to use calculation F as applying the most accurate crevasse perpendicular stress but being subject to these sources of uncertainty.

46. Line 590: the variation by a factor of 2 - is this more or less than confusing the deviatoric and resistive stress? Isn't this something that should be clear from the start of the study?

We will clarify early on that using a deviatoric stress rather than resistive would be a factor of 2 error (depending on strain rate state). And that this study's finding is that even when using a form of resistive stress, assumptions in the calculation steps can still lead to a factor of 2 error for the zero stress approximation.

47. Line 590: The regions of difference between stress calculations on ice shelves is the major new finding and citing a figure to go along and show those differences would be very helpful in your figure 4 - see main point 6.

We will point back to the updated figure 4 / new figure with the new plots.

48. Lines 604-5: re physical basis for the Zero Stress approximation, see main point 2.

We will include in the conclusion reiteration of how Buck's Force Balance Method, particularly in longitudinal flow, is truer to the criterion being no tensile strength.

49. Lines 607-8: cite someone or provide a supplemental figure for these points about convexity of the ice front.

We would discuss terminus shapes from (Choi et al., 2018), plots of the maximum and minimum principal stresses in (Benn et al., 2023), and potentially make plots showing distance from surface crevasse tip to waterline and/or crevasse penetration near the terminus for some of the stress calculations.

50. Table A2: might be useful to have equations in this table as well. Specifically for damage, defining resistive stress, etc.

We would review Table A2 and add at least the suggested equations.

Amaral, T., Bartholomaus, T. C., & Enderlin, E. M. (2020). Evaluation of Iceberg Calving

    Models Against Observations From Greenland Outlet Glaciers. *JOURNAL OF*

    *GEOPHYSICAL RESEARCH-EARTH SURFACE*, *125*(6), e2019JF005444.

    https://doi.org/10.1029/2019JF005444

Anderson, T. L. (2005). *Fracture mechanics: fundamentals and applications* (3rd ed.). Boca

    Raton, FL: Taylor & Francis.

Bassis, J. N., & Walker, C. C. (2011). Upper and lower limits on the stability of calving

    glaciers from the yield strength envelope of ice. *Proceedings of the Royal Society A:*

    *Mathematical, Physical and Engineering Sciences*, *468*(2140), 913–931.

    https://doi.org/10.1098/rspa.2011.0422

Benn, D. I., Todd, J., Luckman, A., Bevan, S., Chudley, T. R., Astroem, J., et al. (2023).

    Controls on calving at a large Greenland tidewater glacier: stress regime, self-

    organised criticality and the crevasse-depth calving law. *JOURNAL OF*

    *GLACIOLOGY*. https://doi.org/10.1017/jog.2023.81

Borstad, C., Khazendar, A., Scheuchl, B., Morlighem, M., Larour, E., & Rignot, E. (2016). A

    constitutive framework for predicting weakening and reduced buttressing of ice

    shelves based on observations of the progressive deterioration of the remnant

    Larsen B Ice Shelf. *Geophysical Research Letters*, *43*(5), 2027–2035.

    https://doi.org/10.1002/2015GL067365

Choi, Y., Morlighem, M., Wood, M., & Bondzio, J. H. (2018). Comparison of four calving laws

    to model Greenland outlet glaciers. *The Cryosphere*, *12*(12), 3735–3746.

    https://doi.org/10.5194/tc-12-3735-2018

Coffey, N. B., Lai, C.-Y., Wang, Y., Buck, W. R., Surawy-Stepney, T., & Hogg, A. E. (2024). Theoretical stability of ice shelf basal crevasses with a vertical temperature profile. *Journal of Glaciology*, *70*, e64. https://doi.org/10.1017/jog.2024.52

Colgan, W., Rajaram, H., Abdalati, W., McCutchan, C., Mottram, R., Moussavi, M. S., & Grigsby, S. (2016). Glacier crevasses; observations, models, and mass balance implications. *Reviews of Geophysics (1985)*, *54*(1), 119–161. https://doi.org/10.1002/2015RG000504

Cuffey, K. M., & Patterson, W. S. B. (2010). *The Physics of Glaciers* (4th ed.). Burlington, MA: Elsevier.

Fürst, J. J., Durand, G., Gillet-Chaulet, F., Tavard, L., Rankl, M., Braun, M., & Gagliardini, O. (2016). The safety band of Antarctic ice shelves. *Nature Climate Change*, *6*(5), 479–482. https://doi.org/10.1038/nclimate2912

Gudmundsson, G. H. (2013). Ice-shelf buttressing and the stability of marine ice sheets. *The Cryosphere*, *7*(2), 647–655. https://doi.org/10.5194/tc-7-647-2013

MacAyeal, D. R. (1989). Large-scale ice flow over a viscous basal sediment: Theory and application to ice stream B, Antarctica. *Journal of Geophysical Research: Solid Earth*, *94*(B4), 4071–4087. https://doi.org/10.1029/JB094iB04p04071

Millstein, J. D., Minchew, B. M., & Pegler, S. S. (2022). Ice viscosity is more sensitive to stress than commonly assumed. *Communications Earth & Environment*, *3*(1), 1–7. https://doi.org/10.1038/s43247-022-00385-x

Nick, F., Veen, C. J., Vieli, A., & Benn, D. (2010). A physically based calving model applied to marine outlet glaciers and implications for the glacier dynamics. *Journal of Glaciology*, *56*. https://doi.org/10.3189/002214310794457344

Nye, J. F. (1955). Comments on Dr. Loewe's Letter and Notes on Crevasses. *Journal of Glaciology*, *2*(17), 512–514. https://doi.org/10.3189/S0022143000032652

Nye, J. F. (1957). The Distribution of Stress and Velocity in Glaciers and Ice-Sheets. *Proceedings of the Royal Society of London. Series A, Mathematical and Physical Sciences*, *239*(1216), 113–133.

Ochwat, N. E., Scambos, T. A., Banwell, A. F., Anderson, R. S., Maclennan, M. L., Picard, G., et al. (2024). Triggers of the 2022 Larsen B multi-year landfast sea ice breakout and initial glacier response. *The Cryosphere*, *18*(4), 1709–1731. https://doi.org/10.5194/tc-18-1709-2024

Rist, M. A., Sammonds, P. R., Murrell, S. a. F., Meredith, P. G., Oerter, H., & Doake, C. S. M. (1996). Experimental fracture and mechanical properties of Antarctic ice: preliminary results. *Annals of Glaciology*, *23*, 284–292. https://doi.org/10.3189/S0260305500013550

Roger Buck, W. (2023). The role of fresh water in driving ice shelf crevassing, rifting and calving. *Earth and Planetary Science Letters*, *624*, 118444. https://doi.org/10.1016/j.epsl.2023.118444

Schoof, C. (2007). Ice sheet grounding line dynamics: Steady states, stability, and hysteresis. *Journal of Geophysical Research: Earth Surface*, *112*(F3). https://doi.org/10.1029/2006JF000664

Van Wyk de Vries, M., Lea, J. M., & Ashmore, D. W. (2023). Crevasse density, orientation and

temporal variability at Narsap Sermia, Greenland. *Journal of Glaciology*, *69*(277),

1125–1137. https://doi.org/10.1017/jog.2023.3

van der Veen, C. J. (1999). Crevasses on glaciers 1. *Polar Geography*, *23*(3), 213–245.

https://doi.org/10.1080/10889379909377677

---

## Author Comment (AC2)

This article provides a detailed review of how resistive stress is calculated for crevasse depth evaluation in various articles in the literature and show that they result in different crevasse depth predictions. As a part of the study, the authors considered both idealized cases and real cases of Antarctic ice shelves, especially those with shear margins. I really appreciated the systematic study in this article, given that my research has focused on exploring the crevasse models in the literature. I believe the article is of interest to the ice sheet modeling community and I will be happy to see it published in this journal. The article is well written and organized (and amazingly I could not find a single typo in text). However, I have several comments listed below, most requiring minor changes or clarifications but I have a couple major comments at the end related to stress evaluation using planar remote sensing data. While I have several articles of mine referenced in my review, I leave it up to the author's discretion to cite or not cite them in their paper.

-- Ravindra Duddu

We thank Dr. Ravindra Duddu for his assessment that this study systematically shows how assumptions in resistive stress calculations impact crevasse depth predictions. We respond to individual comments inline below.

**Detailed Comments:**

Line 51 - I suggest your say zero stress theory instead of Nye crevasse formulation. Originally, Nye (1957) did not include the effect of water pressure, but was later introduced by Jezek (1984).

Per this comment and similar from reviewer #1, we will change this verbiage to "zero stress approximation" throughout the manuscript.

Line 61 – If I remember correctly, Enderlin and Bartholomaus (2020) used observed surface strain rates to calculate stress in grounded glaciers. If the basal boundary is not free slip, then the stress variation with depth in grounded glaciers is not linear. Please clarify this point that the resistive stress is not a constant if boundary condition is not free slip.

That is correct. We would address this in the discussion section (5.3) where we consider this study and (Mottram & Benn, 2009). We provide specifics below in response to your comment about line 541. We will also note the impact of considering stress variation with depth for grounded ice in section 2.1 of the background where we discuss other factors that impact crevasse depth predictions like firn and temperature profile.

Line 81 – The term "stress differences" could be misunderstood. I think it is better to say, "viscous flow is driven by deviatoric stress, which is the component of Cauchy stress that

does not cause volume change during deformation; whereas brittle failure is governed by the Cauchy stress."

We have incorporated this recommendation as:

OLD: While the viscous flow of ice is driven by stress differences (deviatoric stresses), brittle failure comes from the full stress.

NEW: While the viscous flow of ice is driven by deviatoric stress, the component of the Cauchy stress that does not cause volume change during deformation, brittle failure is driven by the Cauchy stress itself.

Line 82 – Consider replacing the terms "biaxial" with "equi-biaxial" and "triaxial" with "equi-triaxial" in the paper, as you are referring to the cases when the stress magnitudes are equal. The terms biaxial and triaxial do not imply the stresses are the same in various directions.

Thank you for this suggestion for improving clarity; we have updated line 82 accordingly and will prepend the "equi" throughout the rest of the manuscript when indicating magnitudes are equal.

Line 90 – I do not agree with the statement "in areas with simple stress states, such as on an ice shelf or near an ice cliff …" Due to free surface effects near an ice cliff of a grounded glacier or floating ice shelf the stress state is not simple. Only in the far field do the stress becomes independent of the horizontal direction and one can use force balance calculation to determine resistive stress. Not sure what I am missing. Please clarify this point.

Thank you for pointing this out. What we were getting at is that stress states are more frequently assumed for ice cliffs (Bassis & Walker, 2011) and longitudinal extension dominated areas of shelves (Millstein et al., 2022). We agree it is incorrect to say that the stress state is truly simple and have removed the sentence as the above point isn't necessary in the paragraph. We have added, "In remote-sensing or field-measurements based workflows," to the following sentence as the introduction for calculating resistive stress from strain rates.

Line 94 and 98 – Change in ice rigidity due to firn layer are important, as you noted. Sorry for the self-promotion, but I would encourage you to read our recent papers (Gao et al., 2023; Clayton et al., 2024), which examine the influence of firn layer on crevasse propagation in glaciers and ice shelves. In Gao et al. (2023) we show that in ice shelves considering depth-varying density due to the firn layer changes the buoyancy depth and leads to deeper penetration. In Clayton et al., (2024) we consider both depth-varying Young's modulus and

density and derive analytical solutions and conduct analytical LEFM studies. We found that the inclusion of depth-dependent density influences the resistive stress and can thwart or promote deeper crevasse propagation depending on the glacier and ocean water heights, which is more nuanced than the description by van der Veen (1998a).

Gao, Y., Ghosh, G., Jiménez, S., & Duddu, R. (2023). A finite-element-based cohesive zone model of water-filled surface crevasse propagation in floating ice tongues. *Computing in Science & Engineering*, vol. 25, no. 3, pp. 8-16, May-June 2023, doi: 10.1109/MCSE.2023.3315661

Clayton, T., Duddu, R., Hageman, T., & Martínez-Pañeda, E. (2024). The influence of firn layer material properties on surface crevasse propagation in glaciers and ice shelves. *The Cryosphere*, *18*(12), 5573-5593.https://doi.org/10.5194/tc-18-5573-2024, 2024.

We appreciate references to recent literature that considered additional firn effects on crevasses. We have added both to the citation with van der Veen (1998a). We also updated the sentence to recognize the impact on stress from deformation as well as density.

OLD: For surface crevasses, firn density is an important consideration as it can significantly change the lithostatic pressure near the surface (van der Veen, 1998a).

NEW: For surface crevasses, firn properties are an important consideration as they can significantly change the lithostatic pressure and resistive stress near the surface (Clayton et al., 2024; Gao et al., 2023; van der Veen, 1998a).

Line 124 – You write height above buoyancy twice. Instead say: "where rho_pw is the density of the proglacial water (lake or ocean) and H_ab is the height above buoyancy defined as" and get rid of repetition.

Suggestion taken verbatim.

Line 130 – Perhaps you should mention that Eq. (4) is only valid in 1D.

We have updated the sentence before equation (4):

OLD: The resistive stress, $R_{xx}$, in Nick et al. (2010) is given as

NEW: The resistive stress, $R_{xx}$, in Nick et al. (2010) is the one-dimensional form and is given as

Line 136 – Better to say "… but considers the stress singularity at the crevasse tip …" Stress concentration is bounded and occurs around holes and U-shaped notches. At the sharp crack tip in LEFM theoretically there is a stress singularity.

Thank you for the correction; the suggestion was taken verbatim.

Line 175 – In Eq. (8), (9) and others, whenever the subscripts are not indices but rather descriptors like "eff" or "eef, planar" you must use \text{} or \mathrm{}. Only indices that are symbols taking numerical values are italicized.

We will update this throughout all the equations.

Line 197 – You can mention that tau_1 and tau_2 in Eq. (13) are invariants with coordinate transformation, whereas tau_flow dir is not in Eq. (12).

We have added the following sentence after the sentence that defined tau_1 and tau_2.

NEW: The principal stresses are invariants with coordinate transformation, while the flow direction stress is not.

Line 266 to 268 – Calculating surface and basal depths using different rigidities based on temperature differences is a bit ad hoc. To see how reasonable it is, a full Stokes FEM simulation could be conducted to obtain the stress field and then the depth where tensile stress becomes zero can be taken as the crevasse depth.

We appreciate this suggestion and, while we agree that this is an ad hoc assumption, we did not see other options as less ad hoc and went with the easiest thing to implement. We think future work could potentially develop damage fields based both on predictions of ice softening and crevasses to try to better understand the degree to which damage is accounting for each factor and would need better evaluation of ad hoc assumptions involved in basal crevasse heights particularly. For this work however, where the pattern of damage from crevasses alone is being tested via modeling as a small add on, we believe this to be beyond our scope.

Line 303 to 304 – In this discussion about inverting for damage, it can be noted that damage in the form of crevasses introduces anisotropy. In Huth et al. (2021) we show that when this anisotropy is considered we get more realistic rift propagation. A comment can be added to state that inversion for isotropic viscosity has the limitation attention that it applies the effect of crevasse damage equally in all directions.

Huth, A., Duddu, R., & Smith, B. (2021). A generalized interpolation material point method for shallow ice shelves. 2: Anisotropic nonlocal damage mechanics and rift propagation. *Journal of Advances in Modeling Earth Systems*, *13*(8), e2020MS002292. https://doi.org/10.1029/2020MS002292

We agree with this point made here and by reviewer #1. Isotropic damage is a limitation of our chosen ice sheet model, ISSM, and we will clarify and address this limitation with following addition at line 307 (end of the second to last paragraph):

NEW: Damage, both calculated from crevasse penetration and with inversion, is implemented assuming isotropy. The reduction in load bearing area from crevasses would be expected to be directional and anisotropic damage laws have been shown to better capture tabular iceberg calving (Huth et al., 2021). Our use of damage is to coarsely compare the bulk rheology implications of different stress calculations, so we move forward with this isotropic assumption.

Table 3 – You are calculating these at a point on the glacier by assuming that the strain rate is uniform. Is that right or did I misunderstand, please clarify.

That is correct. To add clarity in the text, we have added to the second sentence of the paragraph (line 315):

OLD: To compare the stress calculations in these flow types, the magnitudes of each strain rate component are held constant.

NEW: To compare the stress calculations in these idealized flow types, the same magnitude is used for each strain rate component ($\dot{\varepsilon}_{xx}$, $\dot{\varepsilon}_{yy}$ $\dot{\varepsilon}_{xy}$), which are assumed to be constant through thickness.

We have also added the following sentence to the end of the Table 3 caption:

NEW: Strain rate is assumed to be constant through thickness.

Line 333 – Please clarify the phrase "... but the ratio of depths between calculation will be identical" Does mean ratios taken column wise or row wise.

This sentence refers to Figure 4. We have added the sentence following the noted sentence.

NEW: For example, surface crevasse depth predictions for a strain rate state corresponding to -1 on the x-axis with Calc. B, C, and D will still yield a depth twice that of Calc. E and F.

(Per comment from reviewer #1, the calculation names will be updated to a code that gives information about the assumptions involved.)

Line 351 – Would be good to include the function plotted in Figure 4 in the Appendix, for the sake of reproducibility. Or maybe you can share the code used to generate these plots.

For publication, we would add this as a new section in the Jupyter notebook that presently reproduces the crevasse penetration plots. The new section would make the figure 4 plot as well as the minimum principal strain rate / maximum principal strain rate plots of Larsen B / PIG as recommended by reviewer #1.

Line 388 - Perhaps, my only major concern is that the evaluation of stress from observed strain rates in the region of a crack is physically meaningless. This has not been particularly mentioned in the paper. To elaborate, the observed strain rate in an area where there is a crack will be large and the stress evaluated using the Glen's law will be large, so you may rightly predict a full depth crevasse. However, the true Cauchy stress there will be zero because the ice rigidity becomes zero in an open crevasse. Therefore, it is important to point out that while one can use this approach, the evaluated stress is a trial stress (borrowing this term from plasticity) and not the true stress.

This comment is consistent with others from reviewer #1 regarding rifts in particular; we plan to better clarify this and would introduce then use the terminology "trial stress" because we agree it helps in explanation. Additionally, we plan to mask out the rifts based on a thickness and/or strain rate criterion in Figures 5 and 6 to reduce emphasis on these regions where a prediction of full crevasse penetration is accurate by self-guaranteed. We recognize that large crevasses that have not formed rifts are still subject to this.

Line 400 to 410 – The argument comparing Larsen B and Stancomb-Wills shelves are a bit difficult to understand. Also, why did you not include the cross-section thickness and crevasse penetration depth plots in Figure 8, just like in Figure 7. This would perhaps make it easy to follow the differences.

Cross-section thickness and crevasse penetration plots were not included because we did not find that we referenced them in the current explanation provided. We will add cross-section thickness and crevasse penetration plots back in. Additionally, we would rewrite a few sentences that try to discuss too much at once in this section.

Figure 9 could also be clearer if the crevasse depth plot like in Figure 7E was included. Also, by looking at the REMA, how are you able to tell whether crevasses are full depth or not. Please clarify this for the general reader.

A crevasse depth cross section like Figure 7E will be added to the figure. The argument is that the apparent discontinuous ice in 2018 is the result of complete crevasse penetration and the continuous appearance of ice in 2014 suggests crevasse penetration was partial except for the rifts. We would soften the claim that this guarantees there wasn't complete crevasse penetration prior to 2018, as it is possible that penetration was complete but had not yet led to the fragmentation in 2018 for some reason.

Line 443 – The statement "The assumption that the stress calculation that gives the best modeled velocity …" is fine but going to back to my previous comment this is just a trial stress whereas the true stress in a fully crevassed region is zero as the rigidity goes to zero, whereas the strain rate will be large.

Adding to the discussion before, this point is certainly fully true of rifts were any calculation would predict full penetration and the limitation on maximum damage will apply. We will point that out in this section.

In Figure 10A, I would recommend plotting the root mean squared of the velocity misfit rather than the average. The average would not be accurate measure. Also, if the nodal velocity misfit is the least with inversion, they why do we need to use calculation F using observed strain rates. Can we not just invert for damage and obtain damage and make estimates of crevasse depth.

We used the mean absolute error, if the concern is that positive and negative error were allowed to cancel. We will clarify this in the figure as well as its caption and discussion in text.

The first goal with the modeling work was to assess whether calc. E or F had a stronger tie to bulk rheology to recommend that calculation for applications where modeling could not or may not be performed (field measurements, remote sensing based workflows). A second point is that there are downsides to the inversion including non-uniqueness and being a catch all for temperature error, rheology error, and crevasses. The forward calculation of course will be affected by these error sources as well, but in a more traceable way.

We would add clarification (in sections 3.3 and 4.4) about the purpose of the no damage and inverse simulations as bookends to contextualize the forward calculation results as well as discussion about the strengths / weaknesses of crevasses from inversion versus forward calculation.

Line 462 to 465 – Please clarify what you mean by excess velocity. I am also confused by the comment "This may indicate that the damage in the spreading flow region …" Velocity is not a measure of strain rate or damage, instead the symmetric gradient of velocity could be used.

We are making the case that the high velocity misfit near the terminus is a result of over-prediction of crevasse penetration from Calc. E via overly increased strain rate approaching the terminus. To improve clarity, we have adjusted the sentence on line 461.

OLD: The calculation E correlation plot (Fig. 11A) shows that it predicts excess velocity for the fastest-moving ice near the terminus.

NEW: The modeled velocity correlation plot for damage from calculation E (Fig. 11A) shows that it predicts excess velocity for the fastest-moving ice near the terminus.

Figure 11 – Explain why in the 0 - 400 m/yr range the observed velocity is greater than the modeled velocity and why this happens in both cases A and C.

If this comment is in regard to the dense line of points that falls just bull the -20% error line, that is likely the blue patch (Figure 11B and 11D) to the right of the right shear margin. This may indicate the shear margin is too weak in both cases and is not pulling on the slow-moving ice enough. We will add that to the explanation in 4.4.1.

Section 5.2 – As I am reading, I feel like there is a distinction between studies using planar remote sensing data and ice sheet models that was not clearly identified. While you are right about the usage of resistive stress in the formulas with planar remote sensing data, with modeling studies one can evaluate the full stress and identify where it becomes zero in the domain and determine the crevasse depth directly based on zero stress theory. This is what is done I believe in Todd et al. (2018), where they use a full Stokes model that calculates velocity and pressure from mass and momentum balance. In Clayton et al. (2022), we use the momentum balance to calculate the elastic stress in a Maxwell viscoelastic solid and then determine the zero-stress based crevasse depth directly from the stress distribution.

Clayton, T., Duddu, R., Siegert, M., & Martinez-Paneda, E. (2022). A stress-based poro-damage phase field model for hydrofracturing of creeping glaciers and ice shelves. *Engineering Fracture Mechanics*, *272*, 108693.https://doi.org/10.1016/j.engfracmech.2022.108693

Thank you for this comment, we agree this point warrants more clarification. We concur with this distinction but would add the following complexity. In studies that implement the crevasse depth calving law in ice sheet models, approaches split between what could be described as applying the crevasse depth law as a calving parametrization or as a physical criterion. (Choi et al., 2018) and (Wilner et al., 2023), both SSA modeling studies of calving laws, do not find when the maximum principal stress (and assumed zero stress approx. crevasse tip) go to zero in the ice column, but instead us the equations from Nick et al., (2010) (equations 1 and 2 in this manuscript) with deviatoric stress component(s) subbed in for Rxx. In the Sun et al. (2017) damage law, it appears that the maximum principal deviatoric stress is used again leaning towards the parametrization version. In general, the framework we would discuss this distinction in would be:

- Field and remote sensing measurements of strain rate: a calculation of the types shown (e.g. A through F) in this study is unavoidable.
- Modeling studies:

- o Zero stress approximation as parametrization: will also assume the form of one of these calculations, though it may bypass aspects like effective strain rate.
- o Physical basis of zero stress approximation: bypasses the need for using one of our calculation versions.

A corresponding question would be whether to classify modeling studies using the "physical basis" approach like Todd et al. (2018) and In Clayton et al. (2022) as calculation F. Where SSA flow is a perfect assumption, these studies would be identical in our understanding. For inter-study comparison, this is perhaps useful so long as we are clear about the fact that modeling studies can bypass the calculation paths we lay out and handle more complex stress states if full stokes, visco-elastic, etc.

We would add a condensed discussion of the above framework in sections 5.2 (the classification table), 5.4 (discussion of CD law impact), and/or 5.5 (discussion of damage law impact).

Line 534 – The calculation F is recommended for use by the authors, which is reasonable if dealing with planar remote sensing data. With SSA models once can use resistive stress and the crevasse depth formula based on the approach of Sun et al. (2017) or simply once can calculate the depth varying stress using the pressure formula (Huth et al., 2023). Further, principal stress can be obtained from the eigenvalues of the 3 x 3 full stress matrix.

Huth, A., Duddu, R., Smith, B., & Sergienko, O. (2023). Simulating the processes controlling ice-shelf rift paths using damage mechanics. Journal of Glaciology, 69(278):1915-1928. https://doi.org/10.1017/jog.2023.71

Following the above discussion, we could rephrase the recommendation:

OLD: We recommend calculation F based on its physical basis and success in recreating ice sheet velocity patterns when implemented as damage.

NEW: For studies using calculating crevasse depths from observed strain rates, we recommend calculation F based on its physical basis and success in recreating ice sheet velocity patterns when implemented as damage. For studies implementing the crevasse depth calving law or damage laws based on the zero stress approximation, we recommend following the physical basis of the zero stress approximation (crevasse tips reach where the maximum principal stress from the Cauchy tensor reaches zero), which calculation F reproduces for the assumption of SSA flow.

Line 541 – Perhaps, I am repeating this statement. I believe the stress calculations are not valid in Mottram and Benn (2009) or Enderlin and Bartholomaus (2020) as they study

grounded glaciers, unless they are free slip at the base. For glaciers frozen to the bed the stress is not linear. In Jimenez and Duddu (2018) we used a cubic function to fit to the stress profile. I would encourage the authors to study grounded glaciers and the effect of boundary conditions and how this changes the Nye depth. This could be a quick study that could be added to this paper.

We agree with this point. We are curious how much error it would cause for dry surface crevasses that are shallow relative to ice thickness. For deep surface crevasses from meltwater, certainly this will be a major factor. We would assume SIA with varying ice thicknesses and calculate surface crevasse depths with the zero stress approximation, as recommended, to assess this. We would put this work in the supplement and note its findings as relevant to the Mottram and Benn (2009) and Enderlin and Bartholomaus (2020) studies in the discussion section here.

In the acknowledgements, only funding for the DEMs generation is mentioned, but it is not clear how the authors were funded to conduct this study.

Thank you for pointing this out. We will include our funding sources.

Bassis, J. N., & Walker, C. C. (2011). Upper and lower limits on the stability of calving

glaciers from the yield strength envelope of ice. *Proceedings of the Royal Society A:*

*Mathematical, Physical and Engineering Sciences*, *468*(2140), 913–931.

https://doi.org/10.1098/rspa.2011.0422

Choi, Y., Morlighem, M., Wood, M., & Bondzio, J. H. (2018). Comparison of four calving laws

to model Greenland outlet glaciers. *The Cryosphere*, *12*(12), 3735–3746.

https://doi.org/10.5194/tc-12-3735-2018

Clayton, T., Duddu, R., Hageman, T., & Martínez-Pañeda, E. (2024). The influence of firn

layer material properties on surface crevasse propagation in glaciers and ice

shelves. *The Cryosphere*, *18*(12), 5573–5593. https://doi.org/10.5194/tc-18-5573-

2024

Gao, Y., Ghosh, G., Jiménez, S., & Duddu, R. (2023). A Finite-Element-Based Cohesive Zone

Model of Water-Filled Surface Crevasse Propagation in Floating Ice Tongues.

*Computing in Science & Engineering*, *25*(3), 8–16.

https://doi.org/10.1109/MCSE.2023.3315661

Huth, A., Duddu, R., & Smith, B. (2021). A Generalized Interpolation Material Point Method

for Shallow Ice Shelves. 2: Anisotropic Nonlocal Damage Mechanics and Rift

Propagation. *Journal of Advances in Modeling Earth Systems*, *13*(8),

e2020MS002292. https://doi.org/10.1029/2020MS002292

Millstein, J. D., Minchew, B. M., & Pegler, S. S. (2022). Ice viscosity is more sensitive to

stress than commonly assumed. *Communications Earth & Environment*, *3*(1), 1–7.

https://doi.org/10.1038/s43247-022-00385-x

Mottram, R. H., & Benn, D. I. (2009). Testing crevasse-depth models: a field study at Breiðamerkurjökull, Iceland. *Journal of Glaciology*, *55*(192), 746–752. https://doi.org/10.3189/002214309789470905

Nick, F., Veen, C. J., Vieli, A., & Benn, D. (2010). A physically based calving model applied to marine outlet glaciers and implications for the glacier dynamics. *Journal of Glaciology*, *56*. https://doi.org/10.3189/002214310794457344

Sun, S., Cornford, S. L., Moore, J. C., Gladstone, R., & Zhao, L. (2017). Ice shelf fracture parameterization in an ice sheet model. *The Cryosphere*, *11*(6), 2543–2554. https://doi.org/10.5194/tc-11-2543-2017

van der Veen, C. J. (1998). Fracture mechanics approach to penetration of surface crevasses on glaciers. *Cold Regions Science and Technology*, *27*(1), 31–47. https://doi.org/10.1016/S0165-232X(97)00022-0

Wilner, J. A., Morlighem, M., & Cheng, G. (2023). Evaluation of four calving laws for Antarctic ice shelves. *The Cryosphere*, *17*(11), 4889–4901. https://doi.org/10.5194/tc-17-4889-2023

---

## Author Response (AR1)

**Reviewer #1**

**Review of Comprehensive Assessment of Stress Calculations for Crevasse Depths and Testing with Crevasse Penetration as Damage**

The authors make an important point that those in the field should be more careful about their definition of stress. The variety of stresses used in previous literature can lead to pronounced effects on the predicted crevasse depths, and subsequently damage and viscosity, which could potentially modify predictions of ice sheet flow and sea level rise.

That said, the authors miss relevant points in literature, and do not adequately address uncertainties in their modeling. As such, major revisions for this paper are suggested. As a starting minor point, I will refer to your "Nye's theory" as the "Zero Stress Approximation" throughout this review. First, following Benn et al. 2007's error and subsequently many others, the wrong paper is cited in Line 111 - Nye's back-of-the-envelope fracture depth calculation is equation 2 in his 1955 paper "Comments on Dr. Loewe's Letter and Notes on Crevasses". Second, Nye did not discuss basal crevasses, meltwater, or any other variation that has since been applied to the theory, so the version used in this study is not Nye's conception. Third, the term Zero Stress approximation already exists in the literature - Duddu et al., 2020; Huth et al., 2021; Coffey et al., 2024.

We appreciate the correction regarding Nye's 1955 and 1957 papers and have corrected this in the text. We have changed to "zero stress approximation" throughout the manuscript as recommended to be consistent with recent literature. We are encouraged that the reviewer concurs with the importance of showing how stress calculations in recent literature lead to different crevasse depths which impact parametrizations of damage and viscosity used in ice sheet modeling. We provide our updates corresponding the concerns about additional relevant literature and modeling uncertainty inline below.

**General Comments**

- 1. You should not include discussion with LEFM or papers that apply it unless you will do so properly. In its current form, this preprint does not adequately detail LEFM nor its limitations, summarized by the following two points. Overall, since you do not model LEFM, you should remove comparison of previous papers that use LEFM, as well as your description of the theory near the start of the paper. I recommend you focus on comparisons given what you have modeled, which is the Zero Stress Approximation (you call Nye's theory).
  - a. Mode I LEFM, as presented in van der Veen, 1998a,b from Tada's Handbook of Stress Intensity Factors, assumes plane strain (\epsilon\_{yy} = 0, with y the direction into or out of the page). Naturally, as strain rate is the time derivative of strain, this would make the deviatoric stress \tau\_{yy}=0. Hence,

- for Mode I LEFM, you must assume that the flow is 1D, and  $R_{xx} = 2 \times x^2 + 0$ . As such, it's inappropriate to discuss using stress states with 2HD (2 horizontal dimensions) because that goes against the assumptons used to calculate the SIFs (stress intensity factors) in the Tada Handbook.
- b. Second, there are other Modes of fracture for LEFM, specifically in-plane and out-of-plane shear, which are referred to as Modes II and III, respectively. You should not discuss Mode I LEFM as the failure mechanism in shear margins, which makes including Mode I LEFM papers confusing in your discussion of over-/under-predicting in shear margins.
  - We appreciate the constructive feedback regarding our consideration of LEFM. We believe that that our findings are relevant to the application of LEFM to crevasses based on our responses below to the two individual points. In a revised manuscript, we have provided a more thorough introduction to LEFM including Mode I,II,III and mixed mode fracture and studies assessing crevasse orientation.
- a. We appreciate the raising of this complexity and agree that it is an important broken assumption when comparing stress intensity factors to fracture toughness from plane strain tests. That said, we question whether, if LEFM is to be applied to all regions across ice shelves as has been done in Lai et al., 2020 and is likely to be done by future studies, applying a crack perpendicular stress that is known to be inaccurate is the best course of action. Put another way, when applying LEFM where tau\_yy~=0, neglecting the unknown effect of violating this assumption is unavoidable. Applying a stress perpendicular to the crevasse (s\_xx), which is the direction that seems reasonable to assume as most important, that is not the actual stress seems like an additional but more avoidable error. A minor additional point about this: our understanding is that in 2D LEFM derivations with plain strain, it is plain strain in an elastic material such that a stress parallel to the crevasse tip will occur (s\_zz in fracture literature, s\_yy in crevasse literature) (Anderson, 2005 section 2.10). The point remains that this assumption is violated because any value could occur when applying LEFM across glaciers and ice shelves versus the specific value controlled by Poisson's ratio.
- b. Our understanding of Mode I, II, and III crack orientations is that they are based on the loading direction rather than the deformation state. Mode I is defined as "where the principal load is applied normal to the crack plane" (Anderson, 2005 section 2.6). Given the observation of crevasses (admittedly surface crevasses) in shear margins aligning approximately 45 degrees from flow (Colgan et al., 2016; Van Wyk de Vries et al., 2023), which aligns with the maximum principal stress direction, we would consider these to be mode I dominated cracks. Mode III would be a crevasse aligned parallel to flow in a shear margin. There is certainly additional complexity (mixed mode) as crevasses in shear margins reorient with flow, which is less the case for

mode I crevasses in the center of flow. But for crevasse initialization in particular, we would argue Mode I is an appropriate starting place.

With these responses, we have maintained discussion of LEFM and LEFM-based studies in the manuscript. We have described the violation of the assumption noted (a) and that there is additional complexity in shear margins as discussed in (b) particularly as crevasses evolve with flow where we introduce LEFM (Section 2.4). We have also re-raised the noted violation (a) where we recommend Calculation F\_EF-SM-1 (5.1, 5.3, 5.6). Regardless of what LEFM studies choose to do with resistive stress, the difference will be significant and follow the pattern shown with the zero stress approximation.

2. The Zero Stress Approximation does not uphold horizontal force balance. This has been shown in Buck 2023 and discussed in Coffey et al., 2024. For isothermal ice, incorporating force balance as discussed in those two papers yields deeper crevasses, and reduces the calving stress threshold by a factor of 2. This is a significant omission that would alter the predicted crevasse depths maps and velocity misfits when using damage.

We appreciate the raising of this important point regarding the zero stress approximation and Buck's horizontal force balance model. Reviewing in particular Figure 6 of Buck (2023), we make the following notes:

- For the surface crevasse depth / basal crevasse height predictions, similar to LEFM, this model would increase the difference in basal crevasse that predicted from each stress calculation, because of the non-linear relationship between stress and crevasse size.
- 2. For the modeling component of the paper, inclusion of Buck's horizontal force balance model would likely have an impact like going from calc. F\_EF-SM-1 to calc. E\_EP-SM-1 for the Larsen B remnant. The deviation from the zero stress approximation grows with decreasing buttressing. As buttressing numbers tend to be lowest at the front, this would cause a relative weakening of the ice near the front relative to elsewhere. This is similar to the relative weakening toward the front from calc. E from omitting ice softening from the vertical strain rate.

With this, we have left the crevasse penetration plots in terms of the zero stress approximation. This is in part because it is not clear how the force balance model should be applied when longitudinal flow cannot be assumed (shear margins). We have noted the qualitative impact on table 3 and figure 4 as we did for LEFM.

We generated a crevasse penetration map with calc. F\_EF-SM-1 with the force balance model applied, and show the difference to calc. F\_EF-SM-1 with only the zero stress approximation. We also tested velocity misfit from damage with calc. F\_EF-SM-1 and force balance and compare to the original calc. F\_EF-SM-1 and calc.

E\_EF-SM-1 results. These results were included in supplement Section S7 and noted in Section 4.4.1 in the main text.

- 3. There are substantial omissions in addressing uncertainties in your data-model comparison.
  - a. From the modeling side, I have the following questions. They all tie to the point of inverse problems allowing for non-unique matches to data.
    - i. What is the uncertainty regarding rheology, such as the flow law exponent?
    - ii. Is there uncertainty in the dependence of your effective viscosity on temperature?
    - iii. What is your uncertainty in the vertical temperature profile?
    - iv. Is using a depth-averaged ice hardness equivalent to depth-varying ice hardness when computing fracture depths, or do these give possibly different results as discussed in Coffey et al., 2024?
    - v. Is SSA, a long-wavelength or large-scale continuum approximation of the momentum equation, still a good approximation when you have fractures or rifts?
    - vi. Why do you use isotropic damage mechanics when Huth et al., 2021 suggest using anisotropic damage?
    - vii. For the inversion, what is your cost function?
    - viii. If using damage is worse than inversion (Figure 10a), could this mean that your starting point for rheology with no damage (pink) is unreasonable?

Each of these uncertainties was briefly discussed in the methods section (3.3) consistent with the statements below. The purpose of modeling in this paper was primarily to assess whether calc. E\_EP-SM-1 or F\_EF-SM-1 yields a relative pattern of damage that is more consistent with observed velocity, as opposed to providing an estimate of crevasse sizes with fully assessed error bounds.

- i. As reviewed in Cuffey & Patterson (2010), rheology is anisotropic and path dependent as crystallographic preferred orientation and grain size change with deformation. Values of n from 2 to 5 have been found with 3 being used most frequently per their recommendation. Millstein et al. (2022) recently recommended n=4 as a better "average" representation. We assessed n=4 for the Larsen B remnant (Scar inlet) and PIG with the same modeling workflow as supplementary material (S6) and noted the findings in Sections 4.4.1 and 5.1.
- ii. We have clarified in the text that this uncertainty follows the above uncertainty, as microscale processes are temperature dependent and the function for flow factor will have error accordingly.

- iii. We have noted the lack of temperature constraints in the methods section.
- iv. No. We now reference (Coffey et al., 2024) in noting the difference in predictions caused.
- v. No continuum stress balance equations including full stokes will be a good representation of fractures and rifts. We now note that rifts are being treated as continuous.
- vi. This is a limitation of the ice sheet model selected, ISSM, and discussion of this limitation has been added.
- vii. We now state which inversion coefficients were used.
- viii. We were able to improve this inversion result by initializing with predicted crevasse penetration. We note this in section 4.4.
- b. From the observations side, I have the following questions.
  - i. Isn't the data returned from rifts inappropriate for use in your model because a) mélange may have different material properties, b) you may be computing strain rates from mélange velocity and spreading rather than glacial ice, c) the ice thickness is significantly decreased, greatly decreasing observed thickness H and increasing your crevasse depths e.g. R\_{xx} / \rho\_i g H to by default predict a full thickness fracture? One attempt to deal with this by Coffey et al., 2024 is masking H with an average of local unbroken ice thickness, but this does not fix the problem with strain rates. For a study explicitly on crevasse depths you should mask out rifts as your prediction of crevasse depths becomes non-causal.
  - ii. How do you compute the derivative of the velocity field to create strain rate maps? For example, the strain rate maps in Wearing et al., 2016 (thesis) vs Furst et al., 2016 are quite different. Wearing discusses the influence of various spatial filters it would be nice to see maps of strain rate that go into your crevasse depth maps, perhaps in an appendix.

**Responding to each point:**

- i. We agree on all subpoints and have masked the rifts from figures 5 and 6 with a minimum thickness (150m) to avoid readers taking information from these zones where the result is self-fulfilling.
- ii. We use second order accurate central differences (single-sided at boundaries) and no filtering. We have added Strain rate maps for the Scar Inlet and PIG shelves to the supplement (Fig. S1).
- 4. Since you discuss cliff failure, your paper is about errors of around a factor of 2 with the Zero Stress approximation, and you are asking authors to be careful about the confusion between resistive stress and deviatoric stress, is there anything more you would like to say about Bassis and Walker, 2011?

Reviewing Bassis & Walker (2011),  $S_{xx}$  is termed horizontal deviatoric stress when it is instead the horizontal resistive stress. Their Equation 2.2 is consistent with the definition of resistive stress and the resulting equation 2.4 appears to match the depth-averaged boundary condition for a marine ice cliff in terms of resistive stress.

5. Please use names for the stress calculations that have physical relevance and meaning. Calculation A-F gives the reader no insight into the differences. Please make this change in the text and especially in the figures.

We changed the following naming system to add physical relevance while maintaining brevity: E[X]-S[Y]-[Z] where [X] is assigned to 0 (no eff strain rate), P (planar eff strain rate), or F (full effective strain rate); [Y] is assigned to F (flow direction) or M (max prin direction); and [Z] is assigned to 0 (lateral deviatoric stress not used) or 1 (lateral deviatoric stress used).

This gave the following mapping from the old to the new system:

Calc.  $A = A_E0-SF-0$

Calc.  $B = B_E0-SM-0$

Calc.  $C = C_EP-SM-0$

Calc. D = D EF-SM-0

Calc. E = E EP-SM-1

Calc. F = F EF-SM-1

6. Following Table 2 and Figure 4: Can you compute, for a given ice shelf or idealized rectangular domain ice shelf, maps of the ratio of components of strain rate, e.g. minimum / maximum principal strain rate? This will give your audience a good idea of where these different choices of stress are most at play and would pair very nicely with your Figure 4 if you put them side by side. I think this would greatly strengthen your study.

We appreciate this suggestion and agree it would go a long way towards connecting figure 4 to the later crevasse penetration and crevasse penetration difference plots. We made the suggested plots for the Larsen B remnant (or Scar inlet) and PIG shelves and included them with figure 4. In conjunction with the recommendation from reviewer #2 to make code available for reproducing Figure 4, we added code to make these plots to the Jupyter notebook.

**Specific Comments and Technical Corrections**

1. There is no need to include surface meltwater in your figure 1 diagram. You do not use it, making it confusing for a fast read of your figures.

We removed meltwater from both panels of figure 1.

2. Lines 12-14: unnecessary 2 sentences. Either put those in the main text with specific citations or leave them out of the abstract. All you need to say in this upper abstract is that stress calculations vary greatly across studies and make cross-study validation challenging.

Sentences removed.

- 3. Introduction paragraph 1 is too large. Be more succinct or make 2 paragraphs.
  - Line 29: be more specific in tying back these fracture processes to grounding line flux, which is a glacial contribution to the rate of sea level rise.
     Added.
  - b. Line 31: define buttressing with citations.
    We have added the following definition with a sentence before this one.
    NEW: Ice shelves restrain upstream ice flow via buttressing, backstress from shear load transmitted to embayment walls or from compressive load caused by pinning points (Fürst et al., 2016; Gudmundsson, 2013; Schoof, 2007).
  - Line 36: I would not equate calving with shelf collapse. "Both can result in" rather than "the result can be the same"
     Made suggested change.
  - d. Line 38: New paragraph at Finally, maybe drop that word choice. We added the recommended split between processes that drive uncertainty in sea-level rise projections and how crevasse depths are used in parametrizations of those processes (and dropped "Finally").
  - e. Lines 38 to the end of the paragraph: reads as summarizing some previous work with no clear story arc, ending in surface energy balance which is never again mentioned in the paper. Decide if there is a message here or move this to when you discuss individual studies.
    - We dropped energy balance and ended with a stronger statement on importance of crevasse depths for these parametrizations of important processes.
- 4. Line 54: I would add Horizontal Force Balance (see main point 2).

We have modified and added to the start of that paragraph OLD: There are two primary methods for calculating crevasse depths from stress. The Nye crevasse formulation (Nye, 1957) assumes ice has no tensile strength and

that the presence of a crevasse does not modify the surrounding stress field. Linear elastic fracture mechanics (LEFM) ...

NEW: There are three primary methods for calculating crevasse depths from stress. The Zero Stress Approximation (Nye, 1955) assumes ice has no tensile strength and that the presence of a crevasse does not modify the surrounding stress field. The Horizontal Force Balance method (Buck, 2023) maintains the assumption that ice has no tensile strength but considers the impact of water pressure in basal crevasses on force balance. As basal crevasse height increases with stress according to the Zero Stress Approximation, so too does the force balance impact creating a crevasse size amplifying effect. Linear elastic fracture mechanics (LEFM) ...

We also discuss force balance as introduction section 2.3.

- 5. Line 56: LEFM "can recognize" ice strength, but it does not have to as you can choose zero fracture toughness e.g. for Mode I, K\_{Ic} = 0.
  - In an attempt to give a better one sentence opener for LEFM (because another fair point is that a "100KPa stress criterion" would also be recognizing ice strength, we made the following change.
  - OLD: Linear elastic fracture mechanics (LEFM), which was applied to crevasses by van der Veen (1998a, 1998b), recognizes ice strength and considers the stress-amplifying effect of crevasse geometry.
  - NEW: linear elastic fracture mechanics (LEFM), applied to crevasses by (Weertman, 1973) and many subsequent researchers, considers the stress-amplifying effect of crevasse geometry and allows laboratory measurements of a material's resistance to fracture to be used for predicting fracture in more complex stress states.
- 6. Paragraph of Line 75: please end with your main result at the end of the introduction so the audience knows where it is going, not just the broad methodology. We have added the following final sentence:
  - NEW: "We find that common assumptions made when calculating resistive stress from strain rates can lead to differing crevasse depths by a factor of two and that the most physically based calculation applied in an ice sheet model as damage best recreates observed velocity."
- 7. Line 81: ice deformation is set from the full stress tensor regardless of rheology. Also, if you're talking about ice shelves or SSA to start with, please begin with that instead so readers can follow your logic.
  - We are seeking to provide a distinction between the use of the deviatoric stress and the full or Cauchy stress and have made the following change.

OLD: While the viscous flow of ice is driven by stress differences (deviatoric stresses), brittle failure comes from the full stress.

NEW: While the viscous flow of ice is driven by deviatoric stress, the component of the Cauchy stress that does not cause volume change during deformation, brittle failure is driven by the Cauchy stress itself.

- 8. Line 82: provide a citation for ice not being able to flow in triaxial tension. Following a helpful point from review #2 we clarified this as "equi-triaxial" tension. If sigma\_1=sigma\_2=sigma\_3, all deviatoric stress terms are zero. We wanted to point out that an incoherency that would arise from using deviatoric stress for brittle failure is that ice would not be predicted to fail for that scenario.
- 9. Lines 82-3: Lithostatic pressure is essential to all glacier deformation. Lithostatic pressure is what creates the driving stress (e.g. \rho\_i g \partial\_x s) in SSA and is what drives vertical shear ice flow in SIA. Take away gravity as a body force and nothing drives glacier flow. It is often referred to as a viscous gravity current. Thank you for noting that our wording was confusing; we added a sentence pointing out the change in lithostatic pressure with distance causes flow.
- 10. Lines 89-90: Near an ice cliff (or ice front) there will likely be vertical shear effects. Not so simple.
  - Consistent with a similar comment from reviewer #2, we have removed this sentence and modified the following one for continuity.
- 11. Lines 92-4: Consider citing relevant literature: Gao et al., 2023 (firn), Coffey et al., 2024 (temperature), Meng et al., 2024 (poroelasticity).

  We have noted these effects with these citations.
- 12. Lines 99: meltwater in a surface crevasse. A pool of meltwater, or a small lake, will add a vertical force downwards on the ice surface (e.g. MacAyeal et al., 2015)

  Correction taken verbatim.
- 13. Lines 103-7 sentence: can be much more succinct. This can be visualized in the supplement of Buck and Lai 2021, or the Appendices of Coffey et al., 2024. Also, be more specific compressive stress vs lithostatic stress? I recommend lithostatic, unless you are talking about sources of buttressing providing compression. We made the section more succinct and described the increase in compressive stress with elevation from lithostatic pressure minus water pressure as "net compression" because we needed to distinguish between this and lithostatic pressure alone for surface crevasses.
- 14. Equation 4: Is this resistive stress only along-flow or crevasse-normal? Otherwise, you should include the second invariant of strain rate in your calculation (see Appendix A of Coffey et al., 2024). I realize you discuss more later on about

calculating resistive stress, but make a point of what that equation in Nick et al., 2010 is missing early on and what you want to change about it.

We have noted that the resistive stress here is the 1D form

OLD: The resistive stress,  $R_{xx}$ , in Nick et al. (2010) is given as...

NEW: The resistive stress,  $R_{\rm xx}$ , in Nick et al. (2010) is the one-dimensional form and is given as...

We foreshadowed by adding the following to the end of the paragraph:

NEW: Assessing three-dimensional implementations of equation 4 that consider effective strain rate (ice softening from multiple directions of deformation) and lateral stress is the primary focus of our work.

- 15. In case you strongly disagree and want to keep the LEFM portions of this paper,
  - a. Line 135: Say what boundary conditions are unphysical and what applications (ice shelves) must be changed.
  - b. Line 143: The resistive stress is not the same because of the plane strain assumption of the Mode I LEFM result in Tada.
     We rewrote this section consistent with our response to main point 1 addressing
    - (b). Discussion of whether SSA is appropriate for grounded ice crevasses (a) has moved to when we present SSA (section 2.5.1) and discussion of Enderlin & Bartholomaus (2020) and Mottram & Benn (2009) (section 5.3).
- 16. Section 2.4.1: In general, please define the whole Cauchy stress tensor, resistive stress tensor, the relation between deviatoric stress and strain rate, etc. Lead by example in being thorough with your stress definitions.
  - We have done so with a new section 2.5.1 and have introduced the need for the calculation path from strain rate to resistive stress for remote sensing based studies here as well.
- 17. Line 171: provide a citation, or argue that individuals in the field have done this (cite them) and state your opinion on the matter. If you write out the full expression from mass conservation with variable density, what is the relation between strain rates and density?
  - We cited Amaral et al. (2020) here, who neglected vertical strain rate. We also noted that neglected vertical strain rate (assuming it to be zero) will cause a density decrease according to new equation 21. We noted that prior to crevasses, density should not change, but that when crevasses are present it might.
- 18. Line 178: Move your chosen approximate momentum equation (SSA) up earlier when defining your stresses.
  - We are now discussing SSA earlier in the text (in our new section 2.5.1).

19. Equation 10: Move this up, and use another equality to show that the product of the first two terms is what you are calling viscosity.

We moved the equation for nye's generalization (now Equation 17) to directly after the 1d version of Glen's flow law (Equation 16). We dropped mention of viscosity throughout the paper as we do not need to use effective viscosity anywhere.

20. Lines 185-8: Choi et al., 2018 and Lai et al., 2020. You don't need a new sentence about the Lai et al., 2020 application.

Sentence about application removed.

21. Lines 195: can you provide a citation for where you get the jargon planar stress tensor?

Glaciological literature (e.g. van der Veen, 1999) will define tau\_1 and tau\_2 from the upper-left 2x2 matrix (x, y) as the maximum and minimum principal stresses. Recognizing the 3d state, tau\_zz could end up being the maximum or minimum principal stress in some cases. Maximum / minimum principal stress from the planar tensor was our idea for efficiently balancing these targets. Instead, we have now explained this difference between notation in glaciological and the true definition (tau\_1>tau\_2>tau\_3) then use tau\_1 and tau\_2 rather than introducing terminology that could cause confusion with the formal meaning of plane strain.

22. Line 203: cite SSA with neglecting vertical shear stresses.

We have noted SSA as the reason for neglecting bridging stress with citation (MacAyeal, 1989).

23. Equations 18 and 19: Do you mean at the crack tips? Otherwise \sigma\_{zz} should be a function of z.

Added "at the crack tip".

24. Line 215: You will not get the full stress as a function of depth unless you use \sigma\_{zz} (z). This is clear from the (z) component of Stokes flow, removing vertical shear stress terms.

We rewrote this to clarify we are solving for the surface crevasse depth and basal crevasse height where the full stress is zero.

OLD: Equations 18 and 19 can be substituted into Equation 17 to find the full stress as a function of depth ( $\sigma_n$  (z) in Equation 5) for surface and basal crevasses.

NEW: Equations 29 and 30 can be substituted into Equation 28 to find the full stress at the crack tip as a function of surface crevasse depth and basal crevasse height.

25. Lines 230, 526: Add Bassis and Walker to this list.

We originally chose not to include Bassis and Walker because they use the depthaveraged boundary condition rather than calculations from strain rates. As noted in main point 4, we do not think they used deviatoric stress rather than the resistive stress. 26. Line 253: shelf (typo).

Fixed! Thank you.

 $(B = A^{(-1/n)})$  following equation 16.

- 27. Section 3.1: Make these plots! It would be so useful! Even if they go in the appendix, they are the basis for how you understand the bizarre geometries of real ice shelves. I know you have Table 3, but following main point 6, it would be helpful. Thank you for this suggestion, we made these plots and added them to Fig. 4 in the main text.
- 28. Lines 267-8: Since the ocean is saltwater, the freezing point is roughly -2 C. Why do you use 0 instead of -2?
  - This was a carryover from old analysis; we updated to -2 C for all analysis.
- 29. Line 271: Write the rigidity function and say what you have used to interpolate temperature between the surface and the bed. This significantly affects crevasse height, see Lai et al., 2020 and Coffey et al., 2024.
  We added clarification here (constant temperature is assumed) and defined rigidity
- 30. Line 277: The theory you chose suggests that surface crevasses alone don't really matter for making a large damage variable and don't drive calving without water. I would be more forthcoming about why ice shelves are a natural environment to study crevasses (removing basal drag), and that basal crevasses are likely the driver of calving, as they have received far less attention in terms of number of papers. We foreshadowed why shelves are selected at the start of 3.2.1 as this would not yet be obvious. We added emphasis to why shelves are studied and that basal crevasses drive in 3.2.2.
- 31. Line 279: Isn't the Larsen B remnant multi-year landfast sea ice (Ochowat et al., 2023) instead of glacial ice? Would it have different fracture properties? It also collapsed from surface meltwater in 2022 (Ochowat et al., 2023) wouldn't this affect your modeling if the surface crevasses had meltwater in them, or if there were surface meltwater ponds again?
  - We are working with the southern end of the Larsen B that never collapsed. This remaining continuous shelf has been called both the Larsen B remnant (Borstad et al., 2016) and the Scar Inlet (Ochwat et al., 2024) . Scar inlet is less likely to cause confusion with the landfast ice in the embayment, so we switched so Scar Inlet throughout the manuscript
- 32. Paragraph starting with line 285: good logic! Well written. Thank you.
- 33. Paragraph 293: It is unclear what exactly you are doing with temperature. In 293, you say it is constant with depth. In 301, it is quadratic, with a 5-degree shin at 1/3rd of the ice thickness discussed in Line S19. There's a lot of discussion about tuning and

it's very unclear what the overall effect is - you warm bias temperature, and you tune temperature to match velocity with calculation F (shouldn't your damage calculations with calculation E by definition be worse?). Place some of this in S2 if you feel it is detailed.

We provided an updated short qualitative description in the main text and discussed further with a figure in supplement Section S4.

- 34. Line 321: for isothermal ice and the Zero Stress theory, you should be able to predict just how much larger the basal crevasses are than the surface crevasses by computing the ratio of basal to surface crevasses. I would recommend doing so.

  We noted the ratio with our temperature assumptions.
- 35. Table 3: Please make some of this nondimensional. I don't know how to contextualize these other than relative to each other but not a fraction of the ice thickness.
  - We agree fraction of ice thickness does not make sense as zero stress approximation crevasse depths / heights are independent of ice thickness for a given resistive stress and that producing arbitrary crevasse sizes is unfortunate. The only option we see is normalizing against calc. B\_E0-SM-0. Normalizing against a calculation that is not the most "true" seems misleading. Normalizing against calc. F\_EF-SM-1 however would recast the plot and obscure direction of crevasse size change across flow states. For this reason, we maintained as is with a reasonable value of strain rate so that predicted sizes are not unrealistic.
- 36. Figure 4: non-dimensionalize y-axis, change labels to be physically relevant. Same as above. We included our updated stress calculation labels.
- 37. Lines 358 & 390, Figures 5 & 9: as discussed in the main points 3.2.1, every theory will predict rifts if they exist in the data because of the reduced thickness and velocity anomaly. These should either not be included in your analyses or you should treat them carefully.
  - As described under main point 3.2.1, we masked out the rifts to better avoid drawing attention to non-meaningful results. We moved the cross section inf Fig. 7 to avoid the northern shear margin which appears to be partially rift.
- 38. Section 4.3: Is it valid to use the Zero Stress approximation for shear cracks in addition to tensile cracks?
  - Following onto our comments regarding the application of LEFM in shear areas, we would argue that cracks aligned ~45 degrees from the flow direction are tensile cracks in areas of shear deformation. Additional complexity would again apply as deformation reorients crevasses, but as a first pass we think zero stress is reasonable. We addressed in the updated introduction on LEFM (Section 2.4) and did not readdress here.

- 39. Line 442: "tuned ... across the domain" for calculation F? This is unclear.

  A single depth-averaged temperature is assumed across the domain and is tuned for minimum velocity misfit with the pattern of damage applied on top. We rewrote the temperature section to improve clarity.
- 40. Line 447, Figure 10: You should be clear about the point of the inversions. The way it is presented, if I want to match observations, I should just use the inversions, no need for calculation E or F. But I doubt that's what you want to say presumably, it is that you can't do better than the inversion, and the reader should measure your calculations (E & F) velocity misfit relative to no damage. You should state this more explicitly as it is unclear during a first fast read how to interpret your results.

The takeaways we want are:

- 1. Calculation F outperforms calculation E giving some evidence that calculation F has a stronger connection to real basal crevasse height patterns
- 2. Like you point out: Given the bookends of no damage and the best possible damage pattern for matching velocity (inversion), damage from crevasses far outperforms no damage and gets fairly close to inversions.

We rewrote paragraph 3 in section 4.4 to focus on these takeaways.

- 41. Lines 516-18: You can remove this example and just cite them as being unclear. We have done so, and made a clarification between modeling studies that use the physical basis of the zero stress approximation with directly calculated Cauchy stress and those that used a stress calculation in the text (section 5.2) and table 4.
- 42. Lines 522-3: With flow-direction versus maximum principal stress direction, did this alter the conclusions of this study? What is the order of magnitude of this distinction?
  - We included discussion of flow direction results (no crevasses in some shear margins zones) in section 5.1 and 6.
- 43. Line 545: see main point 1.

We reiterated the broken LEFM assumption here (section 5.3).

44. Can you add Wilner et al., 2023 to your table 4?

We added this. Because Wilner (testing the calving law in Pollard et al. (2015)) uses the divergence of velocity rather than resistive stress in 3D or its 2D version, we added a row.

We also added several studies reviewed since the initial submission (Hulbe et al., 2016; Scott et al., 2010) and classified studies that used the zero stress approximation's physical meaning in models (Clayton et al., 2022; Huth et al., 2021).

45. Section 5.6: see main point 1.1 and 1.2.

We noted the violation of plain strain here (1.1) here. We did not note mixed mode (1.2) with the rationale regarding shear crevasse formation from our updated LEFM introduction (section 2.4).

46. Line 590: the variation by a factor of 2 - is this more or less than confusing the deviatoric and resistive stress? Isn't this something that should be clear from the start of the study?

We have now clarified this here and earlier (at Table 3).

47. Line 590: The regions of difference between stress calculations on ice shelves is the major new finding and citing a figure to go along and show those differences would be very helpful in your figure 4 - see main point 6.

We have pointed back to the updated figure 4.

- 48. Lines 604-5: re physical basis for the Zero Stress approximation, see main point 2. We will include in the conclusion reiteration of how Buck's Force Balance Method, particularly in longitudinal flow, is truer to the criterion being no tensile strength.
- 49. Lines 607-8: cite someone or provide a supplemental figure for these points about convexity of the ice front.

We cited the result from Choi et al. (2018).

50. Table A2: might be useful to have equations in this table as well. Specifically for damage, defining resistive stress, etc.

We have added definitions for damage and resistive stress.

**Reviewer #2**

This article provides a detailed review of how resistive stress is calculated for crevasse depth evaluation in various articles in the literature and show that they result in different crevasse depth predictions. As a part of the study, the authors considered both idealized cases and real cases of Antarctic ice shelves, especially those with shear margins. I really appreciated the systematic study in this article, given that my research has focused on exploring the crevasse models in the literature. I believe the article is of interest to the ice sheet modeling community and I will be happy to see it published in this journal. The article is well written and organized (and amazingly I could not find a single typo in text). However, I have several comments listed below, most requiring minor changes or clarifications but I have a couple major comments at the end related to stress evaluation using planar remote sensing data. While I have several articles of mine referenced in my review, I leave it up to the author's discretion to cite or not cite them in their paper.

**-- Ravindra Duddu**

We thank Dr. Ravindra Duddu for his assessment that this study systematically shows how assumptions in resistive stress calculations impact crevasse depth predictions. We note our revisions based on individual comments inline below.

**Detailed Comments:**

Line 51 - I suggest your say zero stress theory instead of Nye crevasse formulation. Originally, Nye (1957) did not include the effect of water pressure, but was later introduced by Jezek (1984).

Per this comment and similar from reviewer #1, we have changed this throughout the manuscript.

Line 61 – If I remember correctly, Enderlin and Bartholomaus (2020) used observed surface strain rates to calculate stress in grounded glaciers. If the basal boundary is not free slip, then the stress variation with depth in grounded glaciers is not linear. Please clarify this point that the resistive stress is not a constant if boundary condition is not free slip.

That is correct. We have added discussion of where the shallow shelf approximation holds up for grounded ice to Section 2.5.1 and to where we discuss the impact of crevasse depth calculation on Enderlin & Bartholomaus (2020) in section 5.3.

Line 81 – The term "stress differences" could be misunderstood. I think it is better to say, "viscous flow is driven by deviatoric stress, which is the component of Cauchy stress that

does not cause volume change during deformation; whereas brittle failure is governed by the Cauchy stress."

We have incorporated this recommendation as:

OLD: While the viscous flow of ice is driven by stress differences (deviatoric stresses), brittle failure comes from the full stress.

NEW: While the viscous flow of ice is driven by deviatoric stress, the component of the Cauchy stress that does not cause volume change during deformation, brittle failure is driven by the Cauchy stress itself.

Line 82 – Consider replacing the terms "biaxial" with "equi-biaxial" and "triaxial" with "equi-triaxial" in the paper, as you are referring to the cases when the stress magnitudes are equal. The terms biaxial and triaxial do not imply the stresses are the same in various directions.

Thank you for this suggestion for improving clarity; we have updated line 82 accordingly and have prepended the "equi" throughout the rest of the manuscript when indicating magnitudes are equal.

Line 90 – I do not agree with the statement "in areas with simple stress states, such as on an ice shelf or near an ice cliff ..." Due to free surface effects near an ice cliff of a grounded glacier or floating ice shelf the stress state is not simple. Only in the far field do the stress becomes independent of the horizontal direction and one can use force balance calculation to determine resistive stress. Not sure what I am missing. Please clarify this point.

Thank you for pointing this out. What we were getting at is that stress states are more frequently assumed for ice cliffs (Bassis & Walker, 2011) and longitudinal extension dominated areas of shelves (Millstein et al., 2022). We agree it is incorrect to say that the stress state is truly simple and have removed the sentence as the above point isn't necessary in the paragraph. We have added, "In remote-sensing or field-measurements based workflows," to the following sentence as the introduction for calculating resistive stress from strain rates.

Line 94 and 98 – Change in ice rigidity due to firn layer are important, as you noted. Sorry for the self-promotion, but I would encourage you to read our recent papers (Gao et al., 2023; Clayton et al., 2024), which examine the influence of firn layer on crevasse propagation in glaciers and ice shelves. In Gao et al. (2023) we show that in ice shelves considering depth-varying density due to the firn layer changes the buoyancy depth and leads to deeper penetration. In Clayton et al., (2024) we consider both depth-varying Young's modulus and

density and derive analytical solutions and conduct analytical LEFM studies. We found that the inclusion of depth-dependent density influences the resistive stress and can thwart or promote deeper crevasse propagation depending on the glacier and ocean water heights, which is more nuanced than the description by van der Veen (1998a).

Gao, Y., Ghosh, G., Jiménez, S., & Duddu, R. (2023). A finite-element-based cohesive zone model of water-filled surface crevasse propagation in floating ice tongues. *Computing in Science & Engineering*, vol. 25, no. 3, pp. 8-16, May-June 2023, doi: 10.1109/MCSE.2023.3315661

Clayton, T., Duddu, R., Hageman, T., & Martínez-Pañeda, E. (2024). The influence of firn layer material properties on surface crevasse propagation in glaciers and ice shelves. *The Cryosphere*, *18*(12), 5573-5593.https://doi.org/10.5194/tc-18-5573-2024, 2024.

We appreciate references to recent literature that considered additional firn effects on crevasses. We have added both to the citation with van der Veen (1998a). We also updated the sentence to recognize the impact on stress from deformation as well as density.

OLD: For surface crevasses, firn density is an important consideration as it can significantly change the lithostatic pressure near the surface (van der Veen, 1998a).

NEW: For surface crevasses, firn properties are an important consideration as they can significantly change the lithostatic pressure and resistive stress near the surface (Clayton et al., 2024; Gao et al., 2023; van der Veen, 1998a).

Line 124 – You write height above buoyancy twice. Instead say: "where rho\_pw is the density of the proglacial water (lake or ocean) and H\_ab is the height above buoyancy defined as" and get rid of repetition.

Suggestion taken verbatim.

Line 130 – Perhaps you should mention that Eq. (4) is only valid in 1D.

We have updated the sentence before equation (4):

OLD: The resistive stress,  $R_{xx}$ , in Nick et al. (2010) is given as

NEW: The resistive stress,  $R_{\rm xx}$ , in Nick et al. (2010) is the one-dimensional form and is given as

Line 136 – Better to say "... but considers the stress singularity at the crevasse tip ..." Stress concentration is bounded and occurs around holes and U-shaped notches. At the sharp crack tip in LEFM theoretically there is a stress singularity.

Thank you for the correction; the suggestion was taken verbatim.

Line 175 – In Eq. (8), (9) and others, whenever the subscripts are not indices but rather descriptors like "eff" or "eef, planar" you must use \text{} or \mathrm{}. Only indices that are symbols taking numerical values are italicized.

We have updated this throughout the manuscript.

Line 197 – You can mention that tau\_1 and tau\_2 in Eq. (13) are invariants with coordinate transformation, whereas tau\_flow dir is not in Eq. (12).

We have added the following sentence after the sentence that defined tau\_1 and tau\_2.

NEW: The principal stresses are invariants with coordinate transformation, while the flow direction stress is not.

Line 266 to 268 – Calculating surface and basal depths using different rigidities based on temperature differences is a bit ad hoc. To see how reasonable it is, a full Stokes FEM simulation could be conducted to obtain the stress field and then the depth where tensile stress becomes zero can be taken as the crevasse depth.

We appreciate this suggestion and, while we agree that this is an ad hoc assumption, we did not see other options as less ad hoc and went with the easiest thing to implement. We think future work could potentially develop damage fields based both on predictions of ice softening and crevasses to try to better understand the degree to which damage is accounting for each factor and would need better evaluation of ad hoc assumptions involved in basal crevasse heights particularly. For this work however, where the pattern of damage from crevasses alone is being tested via modeling as a small add on, we believe this to be beyond our scope.

Line 303 to 304 – In this discussion about inverting for damage, it can be noted that damage in the form of crevasses introduces anisotropy. In Huth et al. (2021) we show that when this anisotropy is considered we get more realistic rift propagation. A comment can be added to state that inversion for isotropic viscosity has the limitation attention that it applies the effect of crevasse damage equally in all directions.

Huth, A., Duddu, R., & Smith, B. (2021). A generalized interpolation material point method for shallow ice shelves. 2: Anisotropic nonlocal damage mechanics and rift propagation. *Journal of Advances in Modeling Earth Systems*, *13*(8), e2020MS002292. https://doi.org/10.1029/2020MS002292

We agree with this point made here and by reviewer #1. Isotropic damage is a limitation of our chosen ice sheet model, ISSM, and we have clarified and addressed this limitation with following addition at the end of the second to last paragraph in Section 3.3 (line 423-429):

NEW: Damage, both calculated from crevasse penetration and with inversion, is implemented assuming isotropy. The reduction in load bearing area from crevasses would be expected to be directional and anisotropic damage laws have been shown to better capture tabular iceberg calving (Huth et al., 2021).

Table 3 – You are calculating these at a point on the glacier by assuming that the strain rate is uniform. Is that right or did I misunderstand, please clarify.

That is correct. To add clarity in the text, we have added to the second sentence of the paragraph (line 315):

OLD: To compare the stress calculations in these flow types, the magnitudes of each strain rate component are held constant.

NEW: To compare the stress calculations in these idealized flow types, the same magnitude is used for each strain rate component ( $\dot{\varepsilon}_{xx}$ ,  $\dot{\varepsilon}_{yy}$ ,  $\dot{\varepsilon}_{xy}$ ), which are assumed to be constant through thickness.

We have also added the following sentence to the end of the Table 3 caption:

NEW: Strain rate is assumed to be constant through thickness.

Line 333 – Please clarify the phrase "... but the ratio of depths between calculation will be identical" Does mean ratios taken column wise or row wise.

This sentence refers to Figure 4. We have added the sentence following the noted sentence.

NEW: For example, surface crevasse depth predictions for a strain rate state corresponding to -1 on the x-axis with Calc. B\_E0-SM-1, C\_EP-SM-0, and D\_EF-SM-0 will still yield a depth twice that of Calc. E\_EP-SM-1 and F\_EF\_SM-1.

Line 351 – Would be good to include the function plotted in Figure 4 in the Appendix, for the sake of reproducibility. Or maybe you can share the code used to generate these plots.

We have updated the Yupyter notebook linked under code availability to reproduce all plots in Fig. 4.

Line 388 - Perhaps, my only major concern is that the evaluation of stress from observed strain rates in the region of a crack is physically meaningless. This has not been particularly mentioned in the paper. To elaborate, the observed strain rate in an area where there is a crack will be large and the stress evaluated using the Glen's law will be large, so you may rightly predict a full depth crevasse. However, the true Cauchy stress there will be zero because the ice rigidity becomes zero in an open crevasse. Therefore, it is important to

point out that while one can use this approach, the evaluated stress is a trial stress (borrowing this term from plasticity) and not the true stress.

This comment is consistent with others from reviewer #1 regarding rifts in particular; we have now made note of this in Section 3.2.1 introducing the trial stress terminology. We also masked out the rifts based on a thickness criterion in Figures 5 and 6 to reduce emphasis on these regions where a prediction of full crevasse penetration is trivial.

Line 400 to 410 – The argument comparing Larsen B and Stancomb-Wills shelves are a bit difficult to understand. Also, why did you not include the cross-section thickness and crevasse penetration depth plots in Figure 8, just like in Figure 7. This would perhaps make it easy to follow the differences.

We added all plots to the Brunt/Stancomb-Wills figure but moved it to the supplement as Fig. S2. We did this as the Brunt/Stancom-Wills only really serves as a counterexample of what the velocity profile of a rift shear margin can look like. With this, we have also removed that paragraph that specifically considered the Brunt.

Figure 9 could also be clearer if the crevasse depth plot like in Figure 7E was included. Also, by looking at the REMA, how are you able to tell whether crevasses are full depth or not. Please clarify this for the general reader.

We have moved the plan view cross section plots from old Fig. 9 to Fig. 8 (replacing the Brunt/Stancomb-Wills figure). We have added the e2/e1 and crevasse penetration cross section plots (Fig. 7) to Fig. 9. We have removed the claim that crevasse penetration is not full in 2014 but suggest that the clear breakup in 2018 (if coming from when crevasse penetration becomes full) would suggest full crevasse penetration was not present in 2014.

Line 443 – The statement "The assumption that the stress calculation that gives the best modeled velocity ..." is fine but going to back to my previous comment this is just a trial stress whereas the true stress in a fully crevassed region is zero as the rigidity goes to zero, whereas the strain rate will be large.

Adding to the discussion before, this point is certainly fully true of rifts were any calculation would predict full penetration and the limitation on maximum damage will apply. We added that clarification to the methods section (3.3) but ultimately did not re-raise the caveat here.

In Figure 10A, I would recommend plotting the root mean squared of the velocity misfit rather than the average. The average would not be accurate measure. Also, if the nodal velocity misfit is the least with inversion, they why do we need to use calculation F using

observed strain rates. Can we not just invert for damage and obtain damage and make estimates of crevasse depth.

We used the mean absolute error, if the concern is that positive and negative error were allowed to cancel. We have now clarified this in the figure and throughout the text.

The first goal with the modeling work was to assess whether calc. E\_EP-SM-1 or F\_EF-SM-1 had a stronger tie to bulk rheology to recommend that calculation for applications where modeling could not or may not be performed (field measurements, remote sensing based workflows). A second point is that there are downsides to the inversion including non-uniqueness and being a catch all for temperature error, rheology error, and crevasses. The forward calculation of course will be affected by these error sources as well, but in a more traceable way.

We rewrote the second to last paragraph in section 4.4 to clarify that the primary goal was calc. E\_EP-SM-1 vs calc. F\_EF-SM-1 and noted the approach to the success of inversion as a minor subsequent point.

Line 462 to 465 – Please clarify what you mean by excess velocity. I am also confused by the comment "This may indicate that the damage in the spreading flow region ..." Velocity is not a measure of strain rate or damage, instead the symmetric gradient of velocity could be used.

We are making the case that the high velocity misfit near the terminus is a result of overprediction of crevasse penetration from Calc. E via overly increased strain rate approaching the terminus. To improve clarity, we have adjusted the sentence on line 461.

OLD: The calculation E correlation plot (Fig. 11A) shows that it predicts excess velocity for the fastest-moving ice near the terminus.

NEW: The modeled velocity correlation plot for damage from calculation E (Fig. 11A) shows that it predicts excess velocity for the fastest-moving ice near the terminus.

Figure 11 – Explain why in the 0 - 400 m/yr range the observed velocity is greater than the modeled velocity and why this happens in both cases A and C.

If this comment is in regard to the dense line of points that falls just bull the -20% error line, that is likely the blue patch (Fig. 11b and d) to the right of the right shear margin. This may indicate the shear margin is too weak in both cases and is not pulling on the slow-moving ice enough. We have now noted this region and added that explanation in the second paragraph of Section 4.4.1.

Section 5.2 – As I am reading, I feel like there is a distinction between studies using planar remote sensing data and ice sheet models that was not clearly identified. While you are right about the usage of resistive stress in the formulas with planar remote sensing data, with modeling studies one can evaluate the full stress and identify where it becomes zero in the domain and determine the crevasse depth directly based on zero stress theory. This is what is done I believe in Todd et al. (2018), where they use a full Stokes model that calculates velocity and pressure from mass and momentum balance. In Clayton et al. (2022), we use the momentum balance to calculate the elastic stress in a Maxwell viscoelastic solid and then determine the zero-stress based crevasse depth directly from the stress distribution.

Clayton, T., Duddu, R., Siegert, M., & Martinez-Paneda, E. (2022). A stress-based porodamage phase field model for hydrofracturing of creeping glaciers and ice shelves. *Engineering Fracture Mechanics*, *272*, 108693.https://doi.org/10.1016/j.engfracmech.2022.108693

Thank you for this comment, we agree this point warrants more clarification. We concur with this distinction but would add the following complexity. In studies that implement the crevasse depth calving law in ice sheet models, approaches split between what could be described as applying the crevasse depth law as a calving parametrization or as a physical criterion. (Choi et al., 2018) and (Wilner et al., 2023), both SSA modeling studies of calving laws, do not find when the maximum principal stress (and assumed zero stress approx. crevasse tip) go to zero in the ice column, but instead us the equations from Nick et al., (2010) (equations 1 and 2 in this manuscript) with deviatoric stress component(s) subbed in for Rxx. In the Sun et al. (2017) damage law, it appears that the maximum principal deviatoric stress is used again leaning towards the parametrization version. In general, the framework we would discuss this distinction in would be:

- Field and remote sensing measurements of strain rate: a calculation of the types shown (e.g. A through F) in this study is unavoidable.
- Modeling studies:
  - Zero stress approximation as parametrization: will also assume the form of one of these calculations, though it may bypass aspects like effective strain rate.
  - Physical basis of zero stress approximation: bypasses the need for using one of our calculation versions.

A corresponding question would be whether to classify modeling studies using the "physical basis" approach like Todd et al. (2018) and In Clayton et al. (2022) as calculation F. Where SSA flow is a perfect assumption, these studies would be identical in our

understanding. For inter-study comparison, this is perhaps useful so long as we are clear about the fact that modeling studies can bypass the calculation paths we lay out and handle more complex stress states if full stokes, visco-elastic, etc.

We have now added a condensed discussion of the above framework in sections 5.2 (the classification table). We included studies that directly calculated crevasse depths in the table but noted them with asterisks to distinguish from studies that calculate resistive stress as laid it in this study.

Line 534 – The calculation F is recommended for use by the authors, which is reasonable if dealing with planar remote sensing data. With SSA models once can use resistive stress and the crevasse depth formula based on the approach of Sun et al. (2017) or simply once can calculate the depth varying stress using the pressure formula (Huth et al., 2023). Further, principal stress can be obtained from the eigenvalues of the 3 x 3 full stress matrix.

Huth, A., Duddu, R., Smith, B., & Sergienko, O. (2023). Simulating the processes controlling ice-shelf rift paths using damage mechanics. Journal of Glaciology, 69(278):1915-1928. https://doi.org/10.1017/jog.2023.71

Following the above discussion, we could rephrase the recommendation:

OLD: We recommend calculation F based on its physical basis and success in recreating ice sheet velocity patterns when implemented as damage.

NEW: For studies using calculating crevasse depths from observed strain rates, we recommend calculation F based on its physical basis and success in recreating ice sheet velocity patterns when implemented as damage. For studies implementing the crevasse depth calving law or damage laws based on the zero stress approximation, we recommend following the physical basis of the zero stress approximation (crevasse tips reach where the maximum principal stress from the Cauchy tensor reaches zero), which calculation F reproduces for the assumption of SSA flow.

Line 541 – Perhaps, I am repeating this statement. I believe the stress calculations are not valid in Mottram and Benn (2009) or Enderlin and Bartholomaus (2020) as they study grounded glaciers, unless they are free slip at the base. For glaciers frozen to the bed the stress is not linear. In Jimenez and Duddu (2018) we used a cubic function to fit to the stress profile. I would encourage the authors to study grounded glaciers and the effect of boundary conditions and how this changes the Nye depth. This could be a quick study that could be added to this paper.

We agree with this point. We are curious how much error it would cause for dry surface crevasses that are shallow relative to ice thickness. For deep surface crevasses from

meltwater, certainly this will be a major factor. We began working on a study to assess this as recommended. Ultimately, we felt that for relatively shallow surface crevasses in tidewater glaciers, a case could be made from the literature that assuming constant velocity with depth is not likely to cause significant error (added to Sections 2.5.1 and 5.3). Developing conditions for when this assumption breaks down in terms of parameters like predicted crevasse depth, ice thickness, fraction of force balance taken by basal drag, and valley cross section aspect ratio would be valuable to develop approximate guard rails for this assumption, but we believe this to be out of scope as a supplement to this study.

In the acknowledgements, only funding for the DEMs generation is mentioned, but it is not clear how the authors were funded to conduct this study.

Thank you for pointing this out. We have now included our funding sources.

**References**

- Amaral, T., Bartholomaus, T. C., & Enderlin, E. M. (2020). Evaluation of Iceberg Calving

  Models Against Observations From Greenland Outlet Glaciers. *JOURNAL OF GEOPHYSICAL RESEARCH-EARTH SURFACE*, 125(6), e2019JF005444.

  https://doi.org/10.1029/2019JF005444
- Anderson, T. L. (2005). *Fracture mechanics: fundamentals and applications* (3rd ed.). Boca Raton, FL: Taylor & Francis.
- Bassis, J. N., & Walker, C. C. (2011). Upper and lower limits on the stability of calving glaciers from the yield strength envelope of ice. *Proceedings of the Royal Society A:*Mathematical, Physical and Engineering Sciences, 468(2140), 913–931.

  https://doi.org/10.1098/rspa.2011.0422
- Borstad, C., Khazendar, A., Scheuchl, B., Morlighem, M., Larour, E., & Rignot, E. (2016). A constitutive framework for predicting weakening and reduced buttressing of ice shelves based on observations of the progressive deterioration of the remnant Larsen B Ice Shelf. *Geophysical Research Letters*, 43(5), 2027–2035. https://doi.org/10.1002/2015GL067365
- Choi, Y., Morlighem, M., Wood, M., & Bondzio, J. H. (2018). Comparison of four calving laws to model Greenland outlet glaciers. *The Cryosphere*, *12*(12), 3735–3746. https://doi.org/10.5194/tc-12-3735-2018
- Clayton, T., Duddu, R., Siegert, M., & Martínez-Pañeda, E. (2022). A stress-based porodamage phase field model for hydrofracturing of creeping glaciers and ice shelves.

- Engineering Fracture Mechanics, 272, 108693. https://doi.org/10.1016/j.engfracmech.2022.108693
- Clayton, T., Duddu, R., Hageman, T., & Martínez-Pañeda, E. (2024). The influence of firn layer material properties on surface crevasse propagation in glaciers and ice shelves. *The Cryosphere*, *18*(12), 5573–5593. https://doi.org/10.5194/tc-18-5573-2024
- Coffey, N. B., Lai, C.-Y., Wang, Y., Buck, W. R., Surawy-Stepney, T., & Hogg, A. E. (2024).

  Theoretical stability of ice shelf basal crevasses with a vertical temperature profile. *Journal of Glaciology*, 70, e64. https://doi.org/10.1017/jog.2024.52
- Colgan, W., Rajaram, H., Abdalati, W., McCutchan, C., Mottram, R., Moussavi, M. S., & Grigsby, S. (2016). Glacier crevasses; observations, models, and mass balance implications. *Reviews of Geophysics (1985)*, *54*(1), 119–161. https://doi.org/10.1002/2015RG000504
- Cuffey, K. M., & Patterson, W. S. B. (2010). *The Physics of Glaciers* (4th ed.). Burlington, MA: Elsevier.
- Enderlin, E. M., & Bartholomaus, T. C. (2020). Sharp contrasts in observed and modeled crevasse patterns at Greenland's marine terminating glaciers. *The Cryosphere*, 14(11), 4121–4133. https://doi.org/10.5194/tc-14-4121-2020
- Fürst, J. J., Durand, G., Gillet-Chaulet, F., Tavard, L., Rankl, M., Braun, M., & Gagliardini, O. (2016). The safety band of Antarctic ice shelves. *Nature Climate Change*, 6(5), 479–482. https://doi.org/10.1038/nclimate2912

- Gao, Y., Ghosh, G., Jiménez, S., & Duddu, R. (2023). A Finite-Element-Based Cohesive Zone Model of Water-Filled Surface Crevasse Propagation in Floating Ice Tongues. Computing in Science & Engineering, 25(3), 8–16. https://doi.org/10.1109/MCSE.2023.3315661
- Gudmundsson, G. H. (2013). Ice-shelf buttressing and the stability of marine ice sheets.

  The Cryosphere, 7(2), 647–655. https://doi.org/10.5194/tc-7-647-2013
- Hulbe, C. L., Klinger, M., Masterson, M., Catania, G., Cruikshank, K., & Bugni, A. (2016).
  Tidal bending and strand cracks at the Kamb Ice Stream grounding line, West
  Antarctica. *Journal of Glaciology*, 62(235), 816–824.
  https://doi.org/10.1017/jog.2016.74
- Huth, A., Duddu, R., & Smith, B. (2021). A Generalized Interpolation Material Point Method for Shallow Ice Shelves. 2: Anisotropic Nonlocal Damage Mechanics and Rift
   Propagation. *Journal of Advances in Modeling Earth Systems*, 13(8),
   e2020MS002292. https://doi.org/10.1029/2020MS002292
- MacAyeal, D. R. (1989). Large-scale ice flow over a viscous basal sediment: Theory and application to ice stream B, Antarctica. *Journal of Geophysical Research: Solid Earth*, 94(B4), 4071–4087. https://doi.org/10.1029/JB094iB04p04071
- Millstein, J. D., Minchew, B. M., & Pegler, S. S. (2022). Ice viscosity is more sensitive to stress than commonly assumed. *Communications Earth & Environment*, *3*(1), 1–7. https://doi.org/10.1038/s43247-022-00385-x

- Mottram, R. H., & Benn, D. I. (2009). Testing crevasse-depth models: a field study at Breiðamerkurjökull, Iceland. *Journal of Glaciology*, *55*(192), 746–752. https://doi.org/10.3189/002214309789470905
- Nick, F., Veen, C. J., Vieli, A., & Benn, D. (2010). A physically based calving model applied to marine outlet glaciers and implications for the glacier dynamics. *Journal of Glaciology*, 56. https://doi.org/10.3189/002214310794457344
- Nye, J. F. (1955). Comments on Dr. Loewe's Letter and Notes on Crevasses. *Journal of Glaciology*, 2(17), 512–514. https://doi.org/10.3189/S0022143000032652
- Nye, J. F. (1957). The Distribution of Stress and Velocity in Glaciers and Ice-Sheets.

  Proceedings of the Royal Society of London. Series A, Mathematical and Physical Sciences, 239(1216), 113–133.
- Ochwat, N. E., Scambos, T. A., Banwell, A. F., Anderson, R. S., Maclennan, M. L., Picard, G., et al. (2024). Triggers of the 2022 Larsen B multi-year landfast sea ice breakout and initial glacier response. *The Cryosphere*, *18*(4), 1709–1731.

  https://doi.org/10.5194/tc-18-1709-2024
- Pollard, D., DeConto, R. M., & Alley, R. B. (2015). Potential Antarctic Ice Sheet retreat driven by hydrofracturing and ice cliff failure. *Earth and Planetary Science Letters*, *412*, 112–121. https://doi.org/10.1016/j.epsl.2014.12.035
- Roger Buck, W. (2023). The role of fresh water in driving ice shelf crevassing, rifting and calving. *Earth and Planetary Science Letters*, *624*, 118444. https://doi.org/10.1016/j.epsl.2023.118444

- Schoof, C. (2007). Ice sheet grounding line dynamics: Steady states, stability, and hysteresis. *Journal of Geophysical Research: Earth Surface*, *112*(F3). https://doi.org/10.1029/2006JF000664
- Scott, J. B. T., Smith, A. M., Bingham, R. G., & Vaughan, D. G. (2010). Crevasses triggered on Pine Island Glacier, West Antarctica, by drilling through an exceptional melt layer.

  Annals of Glaciology, 51(55), 65–70. https://doi.org/10.3189/172756410791392763
- Sun, S., Cornford, S. L., Moore, J. C., Gladstone, R., & Zhao, L. (2017). Ice shelf fracture parameterization in an ice sheet model. *The Cryosphere*, *11*(6), 2543–2554. https://doi.org/10.5194/tc-11-2543-2017
- Van Wyk de Vries, M., Lea, J. M., & Ashmore, D. W. (2023). Crevasse density, orientation and temporal variability at Narsap Sermia, Greenland. *Journal of Glaciology*, 69(277), 1125–1137. https://doi.org/10.1017/jog.2023.3
- van der Veen, C. J. (1998). Fracture mechanics approach to penetration of surface crevasses on glaciers. *Cold Regions Science and Technology*, *27*(1), 31–47. https://doi.org/10.1016/S0165-232X(97)00022-0
- van der Veen, C. J. (1999). Crevasses on glaciers 1. *Polar Geography*, *23*(3), 213–245. https://doi.org/10.1080/10889379909377677
- Weertman, J. (1973). Can a water-filled crevasse reach the bottom surface of a glacier?

  International Association of Scientific Hydrology--Association Internationale

  d'Hydrologie Scientifique (International Union of Geodesy and Geophysics--Union

  de Geodesie et de Geophysique Internationale), Publication. [Louvain], 95, 139–145.

Wilner, J. A., Morlighem, M., & Cheng, G. (2023). Evaluation of four calving laws for Antarctic ice shelves. *The Cryosphere*, *17*(11), 4889–4901.

https://doi.org/10.5194/tc-17-4889-2023

---

## Referee Report (RR1)

**Dear Editor,**

Reynolds and others have made significant improvements to their manuscript since the last round of edits. At this stage, I do not have major comments, but a substantial quantity of minor comments that I feel would improve the readability of the manuscript, as well as a few clarifying comments, that I will list below. The larger comments tend to have sub-points a), b), etc.

Note that all line references for the main text come from the difference file unless otherwise specified.

- In going through the response to reviewers, as well as using control F, I see there is a
  discrepancy between what lines are added to the difference file (presumably red and
  blue) and what is in the original pdf <a href="https://egusphere.copernicus.org/preprints/2024/egusphere-2024-2424/egusphere-2024-2424/egusphere-2024-2424/egusphere-2024-2424/egusphere-2024-2424/egusphere-2024-2424/egusphere-2024-2424/egusphere-2024-2424/egusphere-2024-2424/egusphere-2024-2424/egusphere-2024-2424/egusphere-2024-2424/egusphere-2024-2424/egusphere-2024-2424/egusphere-2024-2424/egusphere-2024-2424/egusphere-2024-2424/egusphere-2024-2424/egusphere-2024-2424/egusphere-2024-2424/egusphere-2024-2424/egusphere-2024-2424/egusphere-2024-2424/egusphere-2024-2424/egusphere-2024-2424/egusphere-2024-2424/egusphere-2024-2424/egusphere-2024-2424/egusphere-2024-2424/egusphere-2024-2424/egusphere-2024-2424/egusphere-2024-2424/egusphere-2024-2424/egusphere-2024-2424/egusphere-2024-2424/egusphere-2024-2424/egusphere-2024-2424/egusphere-2024-2424/egusphere-2024-2424/egusphere-2024-2424/egusphere-2024-2424/egusphere-2024-2424/egusphere-2024-2424/egusphere-2024-2424/egusphere-2024-2424/egusphere-2024-2424/egusphere-2024-2424/egusphere-2024-2424/egusphere-2024-2424/egusphere-2024-2424/egusphere-2024-2424/egusphere-2024-2424/egusphere-2024-2424/egusphere-2024-2424/egusphere-2024-2424/egusphere-2024-2424/egusphere-2024-2424/egusphere-2024-2424/egusphere-2024-2424/egusphere-2024-2424/egusphere-2024-2424/egusphere-2024-2424/egusphere-2024-2424/egusphere-2024-2424/egusphere-2024-2424/egusphere-2024-2424/egusphere-2024-2424/egusphere-2024-2424/egusphere-2024-2424/egusphere-2024-2424/egusphere-2024-2424/egusphere-2024-2424/egusphere-2024-2424/egusphere-2024-2424/egusphere-2024-2424/egusphere-2024-2424/egusphere-2024-2424/egusphere-2024-2424/egusphere-2024-2424/egusphere-2024-2424/egusphere-2024-2424/egusphere-2024-2424/egusphere-2024/egusphere-2024/egusphere-2024/egusphere-2024/egusphere-2024/egusphere-2024/egusphere-2024/egusphere-2024/egu
- 2. Line 64: My understanding is that it would be the impact of crevasses on force balance, not just water pressure in a basal crevasse. It seems that any new boundary condition, whether hydrostatic water or air, might modify the force balance in that framework.
  - a. Line 170: same point.
- 3. Lines 215-221: First, I am glad that you mention this important point. However, I found this text quite confusing, and found it moderately confusing after reading it with the response to reviewers and the mentioned textbook. I would recommend suggesting that you put in a discussion similar to what you responded to reviewers with. The first assumption that tau\_yy is approximately 0 is roughly upheld in various parts if tau\_yy is viscous. The second part, about the influence of young's modulus, is considering the stress state after fractures open, which would imply that validating crevasse depth theories requires the stress state incipient for fracture, not after fractures develop. LEFM (and force balance) begin to think about the stress state after fractures open, which I believe is a point that Anderson was discussing. Either truncate after the added sentence in line 215, or enhance the clarity of this paragraph.
- 4. Table 1: Here, and many places in this study, I do not feel that there is any consideration of how data availability, measurement methods in the field, etc. could contribute to the choice of using equation 23 or 24 (in general lacking a \tau\_2). It may be worth discussing whether data availability constraints influenced authors' choice of these formulations. Please provide some consideration of this as a possible circumstance.
  - a. Another place to discuss this is in section 5.2.
- 5. Lines 417-419: Perhaps consider the work of Surawy-Stepney and others in 2023, who show (and cite in their work) the growing evidence for crevasses and velocity change having a correlation. I think that a point could be made about coupling between flow and fracture: zero stress is fully uncoupled; damage is one-way coupled, and there are currently no theories that are two-way coupled (fractures and velocity co-evolve with equations that are simultaneously solved). I believe this may be a relevant limitation of

all crevasse-depth equations that are inserted into flow laws through damage, as defined in Borstad's work.

- a. Lines 426-428: If I understand correctly, as discussed in the last point, the zero stress approximation and LEFM assume that fractures have no impact on the viscous component of the stress field, and are an uncoupled byproduct of an englacial stress distribution. If so, I think this is a relevant point to make.
- 6. Lines 454-456: There are certainly problems with using SSA while considering fracturing, but you may want to add that it is among the tools at present to try to assess the validity of crevasse depth predictions, and include in your next paragraph.
- 7. Line 476-478: You didn't say if it's Mean Squared Error (I assume). Additionally, I always recommend including all relevant equations, for example the damage model you use and cost function, to be written out somewhere in the paper or supplement, so that a reader does not have to switch papers.
- 8. Lines 481-491: In general, if something is mentioned in your paper, please give a brief explanation or takeaway-message. In this paragraph, you mention that you study other flow-law exponents, and that you study force balance. On result robustness given rheological uncertainty, later in the paper, you mentioned that your main result regarding using planar effective strain rate versus full remains with different flow law exponents. If this is the main point, I would include it again here. Similarly, your supplement shows that the force balance approach has mixed results, with better and worse nodal velocity misfit on two different ice shelves. These points would be useful to set expectations for your readers.
- 9. Figure 4:
  - a. Make a statement somewhere about why A isn't included, even if you find it obvious.
  - b. In lines 518-522, you give beautiful explanations of the different x-axis points of 4e. If possible, it would be fantastic to put a small, <5 word version of these limits as text on the plot, so that readers can see "ice tongue", "1HD flow", "pure shear", etc.
- 10. Line 698-699: Please elaborate on why inverting with no damage but viscosity prefactor tuning can produce the best results, and what that implies for the reliability of your results. Specifically, it would be nice to understand as a reader,
  - a. why the bulk temperature can have such a large effect, and
  - b. why readers should retain confidence in your results working for the "right reasons" given the approximations/limitations discussed in the introduction, when misfit minimization may suggest closer agreement to observations but for the "wrong reasons" (unphysical temperature tuning).

**11. Section 5**

a. Please use numbers or bullet points for your recommendations. Further, provide the evidence in your paper that supports each claim. That will be very clear for everyone, and say what sets your recommendations (nodal velocity misfit, or unphysical crevasse depth prediction, etc.). One idea could be to make it chronological with sections of the paper, which gets to the point of why there are only two stress calculations studied with nodal velocity misfit in the main text.

- i. Example: Line 680: Because EF performs better than EP across all metrics, including the final metric of nodal velocity difference, I think it is the most well-defended point of your paper, as it is stated as the primary takeaway in Line 695. This could be the first/last point, with the corresponding figures that defend this claim.
- b. You should also note in this section that you analyzed results based on the zero stress approximation only in the main text, and all the results that depend on this approximation.
- 12. Lines 875 and 830: both discuss the physical basis of the zero stress approximation. Could you elaborate on what you mean by this? Do you imply that other crevasse theories are not physical, or that studies have not implemented the zero stress approximation properly?
  - a. For example, if the implication is that other crevasse theories are not physical, I would consider the following. In a simple width-averaged ice sheet model, with basal crevasses below surface crevasses, a zero stress crack depth would not result in calving, as the stress required would be twice that of the ice front.
- 13. Supplement S7: I do not feel the authors gave an adequate description of the results. There is a focus on the velocity misfit at the ice front on Scar Inlet, but no speculation as to why force balance appears to do better than the other two inversion results. Please elaborate on this point.
- 14. Supplement Line 131: Please include contours on your plots of where force balance is applied versus the shear margin areas in which it is not. The same goes with the complete shear margin failure result in the other two calculations, described in lines 134-136.
- 15. Speculation in lines 139-145: There are two pieces of this argument that I would like to question.
  - a. First, if I understand correctly, the stress increase in the unbroken ligament (at depth in the ice between crevasse tips) within force balance is not necessarily equal to increasing the "local" stress field one would measure at the ice *surface* with remote sensing products O(kms) away from the cracks. Another theory with a stress increase in the unbroken ligament is LEFM, as the stress at a crack tip would theoretically be infinite, and fall off with radial distance into the unbroken ligament. As such, please consider if you would make the same argument with respect to LEFM, which does indeed indicate that fractures modify the elastic component of the stress field.
  - b. Second, let us suppose that there is indeed a damage feedback mechanism, where fractures influence the viscous flow of ice which in turn influence more fracturing, so on and so forth. In reality, we have observations of stress fields some time after crevasse fields have formed, often with very large strain rates in these fractured areas (e.g. your figure S1b). Additionally, our simple crack depth theories all (zero stress, LEFM, force balance) assume either an initial unfractured state or that cracks don't modify the background stress field (zero stress). If this is the case, I would think that we don't have the correct data to validate our theories in this paper, where you'd want the time-dependent stress

fields that lead to crevasse formation as in Surawy-Stepney and others, 2023. In sum, I think it is a slippery slope to suggest that one of these simple theories would be invalid due to a damage feedback mechanism apparent in the remote sensing data, as it invokes further questions about the existence of damage feedbacks and the well-posedness of the problem you are studying.

**16. Typos**

- a. Line 192: typo: Mode I is load.
- b. Line 196: typo: however, crevasse typically
- c. Line 827: typo: using calculating
- 17. Grammar, rephrasing, and potential citations
  - a. Line 35: I personally think of pinning points as another source of buttressing, but if there is literature that does not call it as such, it is fine to exclude from your definition of buttressing.
  - b. Line 65: This sentence is grammatically correct but rather dense; you might consider rephrasing for clarity. I feel that it can be improved or removed.
  - c. Line 309-310: Consider rewriting this sentence for grammatical correctness and clarity.
  - d. Lines 98-99: consider rewriting this sentence or turning it into two sentences.
  - e. Introductory paragraph with lines 50-60: while the final sentence is removed, I still do not consider this paragraph to have a conclusive end. Put overly simplistically, the paragraph could flow as: "A commonality ... is the importance of ... crevasses. A damage feedback, calving, cliff failure, hydrofracture, ..., all depend upon the modeling of crevasses (cite many papers). And yet, in the simplest theoretical cases, there remains disagreement upon one physical theory for predicting crevasse depths." This would tie it to the next paragraph.
  - f. Line 221: I believe there is reason to truncate this sentence, as it's difficult to quantify the magnitude of this error, particularly given that ice sheets/shelves are often modeled purely viscously.
  - g. Line 330-331: Please write this out in a more detailed manner. If this was the case, where the maximum or minimum principal strain rate was in the vertical direction and not the horizontal plane, what would happen? E.g., when would this lead to horizontal plane fracturing?
    - And second point here, please note that your maximum and minimum principal stresses are considering only the horizontal plane for the entirety of your study. Another place to say this could be lines 358-359.
  - h. Line 409: include the details provided in the response to reviewers, such as second order central finite differencing without filtering, etc.
  - i. Line 95: Cite the paper from which you are using a damage model in ISSM in this sentence.
  - j. Line 803: Some modeling studies ... -> please cite directly which ones you are referring to.
  - k. Supplement: various citations missing with Error! Resource not found.

**References**

- Borstad, C. P., A. Khazendar, E. Larour, M. Morlighem, E. Rignot, M. P. Schodlok, and H. Seroussi (2012), A damage mechanics assessment of the Larsen B ice shelf prior to collapse: Toward a physically-based calving law, *Geophys. Res. Lett.*, 39, L18502, doi:10.1029/2012GL053317.
- Surawy-Stepney, T., Hogg, A.E., Cornford, S.L. et al. Episodic dynamic change linked to damage on the Thwaites Glacier Ice Tongue. Nat. Geosci. 16, 37–43 (2023). <a href="https://doi.org/10.1038/s41561-022-01097-9">https://doi.org/10.1038/s41561-022-01097-9</a>

---

## Author Response (AR2)

Dear Editor,

Reynolds and others have made significant improvements to their manuscript since the last round of edits. At this stage, I do not have major comments, but a substantial quantity of minor comments that I feel would improve the readability of the manuscript, as well as a few clarifying comments, that I will list below. The larger comments tend to have sub-points a), b), etc.

We appreciate the commendation that the manuscript is significantly improved as well as the additional recommendations and corrections below, which we respond to in line.

Note that all line references for the main text come from the difference file unless otherwise specified.

In going through the response to reviewers, as well as using control F, I see there is a
discrepancy between what lines are added to the difference file (presumably red and
blue) and what is in the original pdf -

https://egusphere.copernicus.org/preprints/2024/egusphere-2024-2424/egusphere-2024-2424/egusphere-2024-2424/egusphere-2024-2424/egusphere-2024-2424/egusphere-2024-2424/egusphere-2024-2424/egusphere-2024-2424/egusphere-2024-2424/egusphere-2024-2424/egusphere-2024-2424/egusphere-2024-2424/egusphere-2024-2424/egusphere-2024-2424/egusphere-2024-2424/egusphere-2024-2424/egusphere-2024-2424/egusphere-2024-2424/egusphere-2024-2424/egusphere-2024-2424/egusphere-2024-2424/egusphere-2024-2424/egusphere-2024-2424/egusphere-2024-2424/egusphere-2024-2424/egusphere-2024-2424/egusphere-2024-2424/egusphere-2024-2424/egusphere-2024-2424/egusphere-2024-2424/egusphere-2024-2424/egusphere-2024-2424/egusphere-2024-2424/egusphere-2024-2424/egusphere-2024-2424/egusphere-2024-2424/egusphere-2024-2424/egusphere-2024-2424/egusphere-2024-2424/egusphere-2024-2424/egusphere-2024-2424/egusphere-2024-2424/egusphere-2024-2424/egusphere-2024-2424/egusphere-2024-2424/egusphere-2024-2424/egusphere-2024-2424/egusphere-2024-2424/egusphere-2024-2424/egusphere-2024-2424/egusphere-2024-2424/egusphere-2024-2424/egusphere-2024-2424/egusphere-2024-2424/egusphere-2024-2424/egusphere-2024-2424/egusphere-2024-2424/egusphere-2024-2424/egusphere-2024-2424/egusphere-2024-2424/egusphere-2024-2424/egusphere-2024-2424/egusphere-2024-2424/egusphere-2024-2424/egusphere-2024-2424/egusphere-2024-2424/egusphere-2024-2424/egusphere-2024-2424/egusphere-2024-2424/egusphere-2024-2424/egusphere-2024-2424/egusphere-2024-2424/egusphere-2024-2424/egusphere-2024-2424/egusphere-2024-2424/egusphere-2024-2424/egusphere-2024-2424/egusphere-2024-2424/egusphere-2024-2424/egusphere-2024-2424/egusphere-2024-2424/egusphere-2024-2424/egusphere-2024-2424/egusphere-2024-2424/egusphere-2024-2424/egusphere-2024-2424/egusphere-2024-2424/egusphere-2024-2424/egusphere-2024-2424/egusphere-2024-2424/egusphere-2024-2424/egusphere-2024-2424/egusphere-2024-2424/egusphere-2024-2424/egusphere-2024-2424/egusphere-2024-2424/egusphere-2024-2424/egusphere-2024-2424/egusphere-2024-2424/egusphere-2024-242

To our knowledge and intent, "track changes" was used for the entire editing process between the first preprint and the first revised submission. The one exception to this we are aware of is added citations which are controlled by the automated citation manager and appear black (removed citations are tracked). We generally added the citations recommended by both reviewers and confirmed this in the responses to the first round of revisions. If there are other mismatches, we apologize for the error but would need examples to address the concern.

- 2. Line 64: My understanding is that it would be the impact of crevasses on force balance, not just water pressure in a basal crevasse. It seems that any new boundary condition, whether hydrostatic water or air, might modify the force balance in that framework.
  - a. Line 170: same point.

Thank you for this correction. We have updated old line 64 as:

OLD: Second, the horizontal force balance method (Buck, 2023) maintains the assumption that ice has no tensile strength but considers the impact water pressure in basal crevasses on force balance.

NEW: Second, the horizontal force balance method (Buck, 2023) maintains the assumption that ice has no tensile strength but considers **the impact of reduced ice thickness from surface and basal crevasses**, **air or water pressure in surface crevasses**, **and** water pressure in basal crevasses on force balance.

We have updated old line 170 as:

OLD: As surface crevasse depth and basal crevasse height increase, force is carried by a smaller cross section of ice (here termed the ligament), and basal water pressure adds force that must be counteracted by additional force from ice deformation.

NEW: As surface crevasse depth and basal crevasse height increase, force is carried by a smaller cross section of ice (here termed the ligament), and basal water pressure as well as air or meltwater pressure in surface crevasses adds force that must be counteracted by additional force from ice deformation.

(**bold** indicates the added text)

3. Lines 215-221: First, I am glad that you mention this important point. However, I found this text quite confusing, and found it moderately confusing after reading it with the response to reviewers and the mentioned textbook. I would recommend suggesting that you put in a discussion similar to what you responded to reviewers with. The first assumption - that tau\_yy is approximately 0 - is roughly upheld in various parts if tau\_yy is viscous. The second part, about the influence of young's modulus, is considering the stress state after fractures open, which would imply that validating crevasse depth theories requires the stress state incipient for fracture, not after fractures develop. LEFM (and force balance) begin to think about the stress state after fractures open, which I believe is a point that Anderson was discussing. Either truncate after the added sentence in line 215, or enhance the clarity of this paragraph.

Our understanding: Rate of crevasse formation relevant. If tau\_yy is 0 and crevasse formation is rapid, then, the resulting yy direction stress should be controlled by ice elastic response through poisson ratio and youngs modulus. If tau\_yy is not 0, the plane strain assumption is violated as originally noted. If the crevasse formation is slower, then the yy direction stress could relax to whatever the background state. In this case, even if the background is 0, the yy direction stress will not match the stress corresponding to a linear elastic response. But in this case LEFM may not apply anyway as raised in the conclusion of Jiménez & Duddu (2018).

Updated text 215 and on aimed at raising the issue while remaining more general.

NEW: For an elastic material in plane strain, the formation of a fracture causes a stress running parallel to the crack tip due to Poisson ratio. The additional plasticity from this stress increases the tendency to fracture (Anderson, 2005 Section 2.10). This state may be represented in glacial ice if crevasse formation is rapid and there is no far-field crevasse parallel stress. The latter assumption will be violated in some regions when applying LEFM to all strain rate states across ice sheet surfaces. Impacts of crevasse formation timescales are considered in Jiménez and Duddu (2018), Lipovsky (2020), and Clayton et al. (2024).

We have removed the discussion of the test sample, although the main tenant of LEFM is the ability to compare a crack tip state in the structure (or glacier) of concern to a laboratory test crack tip state thereby avoiding the difficulty of understanding the actual failure process at the crack tip.

- 4. Table 1: Here, and many places in this study, I do not feel that there is any consideration of how data availability, measurement methods in the field, etc. could contribute to the choice of using equation 23 or 24 (in general lacking a \tau\_2). It may be worth discussing whether data availability constraints influenced authors' choice of these formulations. Please provide some consideration of this as a possible circumstance.
  - a. Another place to discuss this is in section 5.2.

We have added discussion of this in the beginning of section 5.2. We also added a note in section 5.3 that measurement with stakes may not give crevasse parallel stress, but that so longs as tau\_2 >= 0, the effect is minimal as can be seen and referenced in Fig. 4e.

5. Lines 417-419: Perhaps consider the work of Surawy-Stepney and others in 2023, who show (and cite in their work) the growing evidence for crevasses and velocity change having a correlation. I think that a point could be made about coupling between flow and fracture: zero stress is fully uncoupled; damage is one-way coupled, and there are currently no theories that are two-way coupled (fractures and velocity co-evolve with equations that are simultaneously solved). I believe this may be a relevant limitation of all crevasse-depth equations that are inserted into flow laws through damage, as defined in Borstad's work.

We have cited Surawy-Stepney after the first clause.

a. Lines 426-428: If I understand correctly, as discussed in the last point, the zero stress approximation and LEFM assume that fractures have no impact on the viscous component of the stress field, and are an uncoupled byproduct of an englacial stress distribution. If so, I think this is a relevant point to make.

Our understanding is that LEFM equates the viscous stress field with an elastic stress field and considers the increase in stress on a reducing cross section. That said, to avoid a potential falsehood, we have noted that zero stress would neglect this and have left LEFM unmentioned.

6. Lines 454-456: There are certainly problems with using SSA while considering fracturing, but you may want to add that it is among the tools at present to try to assess the validity of crevasse depth predictions, and include in your next paragraph.

We have added the following note after old lines 454-456.

OLD: ISSM is run with the shallow shelf approximation (MacAyeal, 1989).

NEW: ISSM is run with the shallow shelf approximation (MacAyeal, 1989), which cannot represent individual fractures but can be used to study the bulk rheology impact of crevasses.

7. Line 476-478: You didn't say if it's Mean Squared Error (I assume). Additionally, I always recommend including all relevant equations, for example the damage model you use and cost function, to be written out somewhere in the paper or supplement, so that a reader does not have to switch papers.

We have noted that the inversion uses mean squared error and have added the cost function equation to our supplement (Section S9).

8. Lines 481-491: In general, if something is mentioned in your paper, please give a brief explanation or takeaway-message. In this paragraph, you mention that you study other flow-law exponents, and that you study force balance. On result robustness given rheological uncertainty, later in the paper, you mentioned that your main result regarding using planar effective strain rate versus full remains with different flow law exponents. If this is the main point, I would include it again here. Similarly, your supplement shows that the force balance approach has mixed results, with better and worse nodal velocity misfit on two different ice shelves. These points would be useful to set expectations for your readers.

We have added clauses to the sentences mentioning n=4 and force balance that note the main takeaways.

**9. Figure 4:**

a. Make a statement somewhere about why A isn't included, even if you find it obvious.

We added a parenthetical in the caption that calc. A is the one not included based on it not using maximum principal direction stresses.

b. In lines 518-522, you give beautiful explanations of the different x-axis points of 4e. If possible, it would be fantastic to put a small, <5 word version of these limits as text on the plot, so that readers can see "ice tongue", "1HD flow", "pure shear", etc.

Thank you. We have added names of these points and an example of where they occur to the figure.

10. Line 698-699: Please elaborate on why inverting with no damage but viscosity prefactor tuning can produce the best results, and what that implies for the reliability of your results. Specifically, it would be nice to understand as a reader,

We have made some small verbiage changes to improve clarity of why we think it makes sense that inversions can do best – namely including all factors in bulk rigidity (spatial temperature, flow law error, crevasses).

a. why the bulk temperature can have such a large effect, and

**Added discussion:**

NEW: Bulk temperature is a strong tuning factor because of ice rheology's high sensitivity to temperature; however, we did not need to tune bulk temperature outside of reasonable values. The tuned depth-averaged temperatures are close to the surface temperatures, which is not unreasonable because of the advection of cold ice as can be seen from borehole measurements at the Fimbul and Amery ice shelves (Humbert, 2010; Wang et al., 2022). The tuned temperatures for the Scar Inlet and Pine Island Glacier ice shelves are discussed in Section 4.4.1 and 4.4.2.

b. why readers should retain confidence in your results working for the "right reasons" given the approximations/limitations discussed in the introduction, when misfit minimization may suggest closer agreement to observations but for the "wrong reasons" (unphysical temperature tuning).

The temperature tuning is unphysical in that it's being tuned not directly modeled. We believe the resulting temperature to be reasonable enough based on the added text above.

**11. Section 5**

- a. Please use numbers or bullet points for your recommendations. Further, provide the evidence in your paper that supports each claim. That will be very clear for everyone, and say what sets your recommendations (nodal velocity misfit, or unphysical crevasse depth prediction, etc.). One idea could be to make it chronological with sections of the paper, which gets to the point of why there are only two stress calculations studied with nodal velocity misfit in the main text.
  - i. Example: Line 680: Because EF performs better than EP across all metrics, including the final metric of nodal velocity difference, I think it is the most well-defended point of your paper, as it is stated as the primary takeaway in Line 695. This could be the first/last point, with the corresponding figures that defend this claim.

Thank you for this suggestion for improving clarity and the strength of our recommendations. We have re-written and re-ordered section 5.1 to better follow the order in the paper: 6 calculations -> remove flow dir based on missing shear margin crevasses -> remove 2d Rxx based on over-predicting shear margin crevasses -> calc F over calc E based on

modeling. We did not number recommendations as the recommendation is simply use calc. F with some caveats noted.

b. You should also note in this section that you analyzed results based on the zero stress approximation only in the main text, and all the results that depend on this approximation.

Added to the second to last sentence of 5.1.

- 12. Lines 875 and 830: both discuss the physical basis of the zero stress approximation. Could you elaborate on what you mean by this? Do you imply that other crevasse theories are not physical, or that studies have not implemented the zero stress approximation properly?
  - a. For example, if the implication is that other crevasse theories are not physical, I would consider the following. In a simple width-averaged ice sheet model, with basal crevasses below surface crevasses, a zero stress crack depth would not result in calving, as the stress required would be twice that of the ice front.

We did not intend any comparison / claims about the physical basis of zero stress vs force balance vs LEFM. Our meaning is more consistent with the latter statement, "studies have not implemented the zero stress approximation properly." We also do not claim that studies have implemented LEFM improperly, but we do raise the point that these stress calculations will yield large differences in LEFM workflows that more or less follow the differences that would occur for the zero stress approximation.

Our claim is that the zero stress approximation seems to work best in accordance with its physical meaning (crevasse ends where the max principal full stress term reaches zero (noted in old 830-831)) rather than any of the other forms that show up in literature.

To clarify this, we have switched to "mathematical consistency" above old line 830, have changed "physical basis" to "physical meaning" in old line 830 and old line 875, and have changed from "physical basis" to "physical consistency" in the final paragraph of the conclusion. We hope this clarifies that we are not claiming the zero stress approximation has the strongest physical basis but that calculating stress as calc. F is consistent with continuum mechanics for an incompressible, isotropic fluid (which of course ice is not).

13. Supplement S7: I do not feel the authors gave an adequate description of the results.

There is a focus on the velocity misfit at the ice front on Scar Inlet, but no speculation as

to why force balance appears to do better than the other two inversion results. Please elaborate on this point.

We have added a small final paragraph in the section discussing the results from Pine Island Glacier ice shelf.

14. Supplement Line 131: Please include contours on your plots of where force balance is applied versus the shear margin areas in which it is not. The same goes with the complete shear margin failure result in the other two calculations, described in lines 134-136.

We did not manually delineate shear margins but used flow direction buttressing rather than maximum principal direction buttressing. This gives the desired effect (removes shear margins) without manual delineation because the flow and maximum principal stress directions are similar in the center of flow but misaligned in shear margins. We therefore cannot easily include the suggested contours and have left these figures unchanged. However, to improve clarity we added the following sentence as second to last in the first paragraph of the force balance method section (S7):

"The flow direction and maximum principal directions are similar in the center of flow but diverge in shear margins."

- 15. Speculation in lines 139-145: There are two pieces of this argument that I would like to question.
  - a. First, if I understand correctly, the stress increase in the unbroken ligament (at depth in the ice between crevasse tips) within force balance is not necessarily equal to increasing the "local" stress field one would measure at the ice *surface* with remote sensing products O(kms) away from the cracks. Another theory with a stress increase in the unbroken ligament is LEFM, as the stress at a crack tip would theoretically be infinite, and fall off with radial distance into the unbroken ligament. As such, please consider if you would make the same argument with respect to LEFM, which does indeed indicate that fractures modify the elastic component of the stress field.

We concur with this point and elaborate below. Stating it as "double counting" was too strong as we agree that the remote-sensing-measured strain rate is unlikely to be amplified to the strain rate in the ligament. On the other hand, if it is higher, because of the ligament's increased strain rate and some surrounding region, then a "partial double counting" will occur which is our new verbiage.

b. Second, let us suppose that there is indeed a damage feedback mechanism, where fractures influence the viscous flow of ice which in turn influence more fracturing, so on and so forth. In reality, we have observations of stress fields some time after crevasse fields have formed, often with very large strain rates in these fractured areas (e.g. your figure S1b). Additionally, our simple crack depth theories all (zero stress, LEFM, force balance) assume either an initial unfractured state or that cracks don't modify the background stress field (zero stress). If this is the case, I would think that we don't have the correct data to validate our theories in this paper, where you'd want the time-dependent stress fields that lead to crevasse formation as in Surawy-Stepney and others, 2023. In sum, I think it is a slippery slope to suggest that one of these simple theories would be invalid due to a damage feedback mechanism apparent in the remote sensing data, as it invokes further questions about the existence of damage feedbacks and the well-posedness of the problem you are studying.

Again our full, conceptual response is below. Here we'll be clear that we did not intend to claim force balance (or one of the other two) is invalid. For calculating what crevasse sizes would form starting with unbroken ice, we are convinced that force balance is more appropriate than the zero stress approximation (though this matters significantly only when crevasses become large and is an aside to the purpose of this manuscript). Our findings do leave us with a potential problem: force balance applied in this workflow, which makes two temperature assumptions that would tend to decrease crevasse size (-2C for the full basal crevasse and -2C for calculation of buttressing number), predicts complete crevasse penetration and thus a large speedup of the front. We provide our theory for why this occurred.

**To respond fully, we can consider two end members:**

- 1) The presence of crevasses has no effect on the remote-sensing-measured strain rate: in this case, force balance would be most appropriate.
- 2) The presence of crevasses causes an increase in measured strain rate corresponding to the seracs between carrying no load: This is the assumption of damage as applied by Sun et al. (2017). In this case, even though the size of the crevasses is amplified by the effects force balance considers, those effects have fully modified the measured strain rate such that the zero stress approximation just yields the thickness of ice that would be in tension. "If ice were continuously load bearing here, it would be in tension so crevasse is predicted." This is consistent with the trial stress idea introduced in the manuscript following reviewer #2's comments.
- We think reality falls between those end members meaning that (if ice has no strength were the perfect failure criterion), the zero stress would somewhat underpredict and force balance would somewhat overpredict when working from remote-sensing products in crevassed regions.
- With this explanation, we have included recognition that the remote-sensing strain rate is likely to be increased due to including the elevated strain rate at and around crevasses in each "pixel." And called the effect a partial double

counting and added more verbiage clarifying this is our understanding of a possible explanation.

**16. Typos**

a. Line 192: typo: Mode I is load.

b. Line 196: typo: however, crevasse typically

c. Line 827: typo: using calculating

All corrected. Thank you!

**17. Grammar, rephrasing, and potential citations**

a. Line 35: I personally think of pinning points as another source of buttressing, but if there is literature that does not call it as such, it is fine to exclude from your definition of buttressing.

Our intent was to include pinning points within the definition of buttressing but we were not clear. Rewritten as:

OLD: Ice shelves restrain upstream ice flow via buttressing, backstress from shear load transmitted to embayment walls, or from compressive loading caused by pinning points (Fürst et al., 2016; Gudmundsson, 2013; Schoof, 2007).

NEW: Ice shelves restrain upstream ice flow via buttressing, backstress which comes from shear load transmitted to embayment walls or compressive loading caused by pinning points (Fürst et al., 2016; Gudmundsson, 2013; Schoof, 2007).

b. Line 65: This sentence is grammatically correct but rather dense; you might consider rephrasing for clarity. I feel that it can be improved or removed.

Sentence removed.

c. Line 309-310: Consider rewriting this sentence for grammatical correctness and clarity.

**Rewritten.**

OLD: This is incorrect when considering stress before crevasse formation but could potentially apply once crevasses have formed violating incompressibility.

NEW: Neglecting vertical strain rate is incorrect when considering stress prior to crevasse formation, but it could be argued that incompressibility no longer applies once crevasses exist.

d. Lines 98-99: consider rewriting this sentence or turning it into two sentences.

Split into two sentences:

OLD: While the viscous flow of ice is driven by deviatoric, the component of the Cauchy stress that does not cause volume change during deformation, brittle failure is driven by the Cauchy stress itself.

NEW: The viscous flow of ice is driven by deviatoric stress, the component of the Cauchy stress that does not cause volume change during deformation. Brittle failure is driven by the Cauchy stress itself.

e. Introductory paragraph with lines 50-60: while the final sentence is removed, I still do not consider this paragraph to have a conclusive end. Put overly simplistically, the paragraph could flow as: "A commonality ... is the importance of ... crevasses. A damage feedback, calving, cliff failure, hydrofracture, ..., all depend upon the modeling of crevasses (cite many papers). And yet, in the simplest theoretical cases, there remains disagreement upon one physical theory for predicting crevasse depths." This would tie it to the next paragraph.

We have added to the end of that paragraph:

NEW: With this need for crevasse depths in modeling these processes, researchers have proposed several physical theories for making crevasse depth predictions.

And replaced the sentence at the start of the next paragraph:

OLD: There are three primary methods for calculating crevasse depths from stress.

NEW: The three primary theories for crevasse depth predictions vary in their assumptions about ice's strength and the effect of a crevasse the surrounding stress field.

f. Line 221: I believe there is reason to truncate this sentence, as it's difficult to quantify the magnitude of this error, particularly given that ice sheets/shelves are often modeled purely viscously.

While we agree quantifying this error would be challenging, we do not believe that to be grounds to neglect raising this violation that is guaranteed to occur when plane strain LEFM equations are used. We have added "in some regions" to allow the possibility that it's a small effect, as we (and to our knowledge, the field) do not know.

g. Line 330-331: Please write this out in a more detailed manner. If this was the case, where the maximum or minimum principal strain rate was in the vertical

direction and not the horizontal plane, what would happen? E.g., when would this lead to horizontal plane fracturing?

i. And second point here, please note that your maximum and minimum principal stresses are considering only the horizontal plane for the entirety of your study. Another place to say this could be lines 358-359.

In old 330-331 we updated to say that the vertical will "often" be the max or min, as it would be the min for longitudinal extension. We also added sentences explaining when vertical will be max or min and noted that in our study we always mean max and min of the surface terms.

NEW after old 330-331: If the surface terms compressive, the vertical stress will be the true maximum principal stress and horizontal plane fracturing would be predicted. When the surface terms are tensile, the vertical term will be compressive and would be the minimum principal deviatoric stress. Throughout this study, maximum and minimum principal deviatoric stresses,  $\tau_1$  and  $\tau_2$  will always refer to surface components. `

NEW after old 358-359: ...again considering only the surface terms.

h. Line 409: include the details provided in the response to reviewers, such as second order central finite differencing without filtering, etc.

Updated noting 2nd order and no filtering.

i. Line 95: Cite the paper from which you are using a damage model in ISSM in this sentence.

We have added citations of ISSM and an ISSM damage application here (Borstad et al., 2012; Larour et al., 2012).

j. Line 803: Some modeling studies ... -> please cite directly which ones you are referring to.

Cited: (Choi et al., 2018; Pollard et al., 2015; Sun et al., 2017; Wilner et al., 2023)

k. Supplement: various citations missing with Error! Resource not found.

Thank you. We have fixed these broken references.

- Borstad, C. P., A. Khazendar, E. Larour, M. Morlighem, E. Rignot, M. P. Schodlok, and H. Seroussi (2012), A damage mechanics assessment of the Larsen B ice shelf prior to collapse: Toward a physically-based calving law, *Geophys. Res. Lett.*, 39, L18502, doi:10.1029/2012GL053317.
- 2. Surawy-Stepney, T., Hogg, A.E., Cornford, S.L. et al. Episodic dynamic change linked to damage on the Thwaites Glacier Ice Tongue. Nat. Geosci. 16, 37–43 (2023).

https://doi.org/10.1038/s41561-022-01097-9

**References for reply:**

- Borstad, C. P., Khazendar, A., Larour, E., Morlighem, M., Rignot, E., Schodlok, M. P., & Seroussi, H. (2012). A damage mechanics assessment of the Larsen B ice shelf prior to collapse: Toward a physically-based calving law. *Geophysical Research Letters*, 39(18). https://doi.org/10.1029/2012GL053317
- Choi, Y., Morlighem, M., Wood, M., & Bondzio, J. H. (2018). Comparison of four calving laws to model Greenland outlet glaciers. *The Cryosphere*, *12*(12), 3735–3746. https://doi.org/10.5194/tc-12-3735-2018
- Humbert, A. (2010). The temperature regime of Fimbulisen, Antarctica. *Annals of Glaciology*, *51*(55), 56–64. https://doi.org/10.3189/172756410791392673
- Jiménez, S., & Duddu, R. (2018). On the evaluation of the stress intensity factor in calving models using linear elastic fracture mechanics. *Journal of Glaciology*, *64*(247), 759–770. https://doi.org/10.1017/jog.2018.64
- Larour, E., Seroussi, H., Morlighem, M., & Rignot, E. (2012). Continental scale, high order, high spatial resolution, ice sheet modeling using the Ice Sheet System Model (ISSM). *Journal of Geophysical Research: Earth Surface*, *117*(F1). https://doi.org/10.1029/2011JF002140
- MacAyeal, D. R. (1989). Large-scale ice flow over a viscous basal sediment: Theory and application to ice stream B, Antarctica. *Journal of Geophysical Research: Solid Earth*, 94(B4), 4071–4087. https://doi.org/10.1029/JB094iB04p04071
- Pollard, D., DeConto, R. M., & Alley, R. B. (2015). Potential Antarctic Ice Sheet retreat driven by hydrofracturing and ice cliff failure. *Earth and Planetary Science Letters*, *412*, 112–121. https://doi.org/10.1016/j.epsl.2014.12.035
- Roger Buck, W. (2023). The role of fresh water in driving ice shelf crevassing, rifting and calving. *Earth and Planetary Science Letters*, *624*, 118444. https://doi.org/10.1016/j.epsl.2023.118444

- Sun, S., Cornford, S. L., Moore, J. C., Gladstone, R., & Zhao, L. (2017). Ice shelf fracture parameterization in an ice sheet model. *The Cryosphere*, *11*(6), 2543–2554. https://doi.org/10.5194/tc-11-2543-2017
- Wang, Y., Zhao, C., Gladstone, R., Galton-Fenzi, B., & Warner, R. (2022). Thermal structure of the Amery Ice Shelf from borehole observations and simulations. *The Cryosphere*, *16*(4), 1221–1245. https://doi.org/10.5194/tc-16-1221-2022
- Wilner, J. A., Morlighem, M., & Cheng, G. (2023). Evaluation of four calving laws for Antarctic ice shelves. *The Cryosphere*, *17*(11), 4889–4901. https://doi.org/10.5194/tc-17-4889-2023

We again thank Dr. Duddu for his help in being as clear and precise with fracture definitions as possible. Our replies are in-line below.

I commend the authors for their comprehensive responses to both reviewer questions. Except for one minor comment / correction below about fracture modes that needs revision, the article can be accepted for publication.

**Comment:**

In response to reviewer 1, comment 1, the authors modified the discussion of mode I, II and III fracture. A key difference between mode II and mode III is not just it is simply inplane and out-of-plane shear but also how the crack opens, in mode II the crack opens through sliding whereas in mode II the crack opens through tearing. One can have mode II (crack sliding) fracture in-plane or out-of-plane, after all the coordinate directions are simply constructs and depending on whether we consider a 2D flow-line or 2D shallow shelf, the out-of-plane direction is different. In shear zones of ice shelves, the 45 degree in-plane crack is mode I dominated (i.e. crack opening is along the maximum tensile stress direction) but can have a mix of mode I and mode II, where the mode II crack will be parallel to the flow line. In contrast, mode III requires forces out of the ice shelf plane in the vertical direction, for example, non-uniform ocean swell could cause tearing at the tip of a rift or tearing could also occur as shear margins due to buoyancy forces. As such a mode III crack will still be perpendicular to the flow lines as the tearing will occurs out-of-plane of the ice shelf. Mode III cannot occur in-plane of the ice shelf because the ice shelf length and width are much larger than the ice shelf thickness.

Based on this understanding, I feel that the discussion on Lines 174 to 180 must be revised as follows:

-- "Mode I involves opening of the crevasse walls wide apart, Mode II involves sliding of the crevasse walls, as in a strike-slip fault, and Mode III involves tearing, for example, due to the rising of the surface on one side of the crevasse while the other side's surface lowers, and

(van der Veen, 1998a)."

-- "This tendency holds in shear margins, where crevasses form approximately 45-degrees

from flow as Mode I crevasses, whereas Mode II fractures would strike parallel to the flow direction."

Thank you for this important correction. We have taken the suggestion verbatim but have changed the citation to van der veen (1999) which provides an explanation in terms more like this verbiage.

Also, another minor correction is that authors say plain strain multiple times in their response, instead of plane strain. In the manuscript it is correctly typed as plane strain.

Thank you for the double check on this, we have double checked that the usage is correct and consistent in the manuscript (and principle vs principal).